# Are EEG Foundation Models Worth It? Comparative Evaluation with Traditional Decoders in Diverse BCI Tasks

**Liuyin Yang**[1][*][†], **Qiang Sun**[1][*], **Ang Li**[2], **Marc M. Van Hulle**[1][†]
[1]Laboratory for Neuro- & Psychophysiology, Department of Neurosciences, KU Leuven
[2]Shanghai Advanced Research Institute, Chinese Academy of Sciences
Code & Models: https://github.com/LiuyinYang1101/STEEGFormer

## Abstract

Foundation models have recently emerged as a promising approach for learning generalizable EEG representations for brain–computer interfaces (BCIs). Yet, their true advantages over traditional methods—particularly classical non-neural approaches—remain unclear. In this work, we present a comprehensive benchmark of state-of-the-art EEG foundation models, evaluated across diverse datasets, decoding tasks, and six evaluation protocols, with rigorous statistical testing. We introduce spatiotemporal EEGFormer (*ST-EEGFormer*), a simple yet effective Vision Transformer (ViT)-based baseline, pre-trained solely with masked autoencoding (MAE) on over 8M EEG segments. Our results show that while fine-tuned foundation models perform well in data-rich, population-level settings, they often fail to significantly outperform compact neural networks or even classical non-neural decoders in data-scarce scenarios. Furthermore, linear probing remains consistently weak, and performance varies greatly across downstream tasks, with no clear scaling law observed among neural network decoders. These findings expose a substantial gap between pre-training and downstream fine-tuning, often diminishing the benefits of complex pre-training tasks. We further identify hidden architectural factors that affect performance and emphasize the need for transparent, statistically rigorous evaluation. Overall, this study calls for community-wide efforts to construct large-scale EEG datasets and for fair, reproducible benchmarks to advance EEG foundation models.

## 1 Introduction

Electroencephalography (EEG) is a non-invasive brain recording technique widely used in Brain-Computer Interface (BCI) research to improve the quality of life for patients and to augment the capabilities of healthy individuals. Various EEG paradigms have been explored to advance BCI development. For example, motor imagery (MI)—the mental simulation of physical movement—can be employed to control exoskeletons (Soekadar et al., 2016; Choi et al., 2020), navigate real or virtual environments (Choi & Cichocki, 2008; Tsui et al., 2011; Yang & Van Hulle, 2023), or facilitate rehabilitation (Baniqued et al., 2021; Liao et al., 2023). Event-related potentials (ERPs), such as the P300 response—a positive deflection elicited by infrequent events—have been leveraged for user attention decoding and smart home applications (Holzner et al., 2009; Masud et al., 2017). Visual-evoked potentials (VEPs), including steady-state visual-evoked potentials (SSVEPs), are amplitude changes in EEG elicited by visual stimuli, enabling high-speed spelling devices (Wittevrongel & Van Hulle, 2017; Nakanishi et al., 2018a; Xing et al., 2018).

Despite the rapid progress of deep learning in BCI, which has led to state-of-the-art performance across many tasks, variations in experimental paradigms, hardware setups, and limited dataset sizes often necessitate training separate models for each task (Murad & Rahimi, 2025). A promising solution is the use of EEG foundation models—models pre-trained on large-scale EEG datasets to learn generalizable representations that can be adapted to a variety of downstream tasks. In recent years, numerous EEG foundation models have reported strong performance across diverse BCI

---

[*]Equal contribution.
[†]Correspondence to: liuyin.yang@kuleuven.be & marc.vanhulle@kuleuven.be

tasks. However, many studies lack comparisons to classical EEG decoding methods and typically evaluate performance on only one or two protocols, such as leave-one-out zero-shot or population decoding—often without statistical testing. While existing work demonstrates the feasibility of applying foundation models to EEG decoding, the practical benefits remain uncertain, especially given the substantial computational and time costs associated with pre-training and fine-tuning.

**Contributions:** We present a comprehensive benchmark of EEG foundation models, introducing a six-dimensional evaluation framework that encompasses seven classification tasks and two regression tasks. Our extensive experiments enable statistically rigorous comparisons across decoders and evaluation protocols. We systematically compare foundation models against both classical non-neural and neural network-based EEG decoders across diverse paradigms. Additionally, we introduce *ST-EEGFormer*—a simple, transparent Vision Transformer-based (Dosovitskiy et al., 2021) foundation model pre-trained solely with MAE (He et al., 2022) on more than 8 million raw EEG segments—serving as a strong baseline. Our key findings are as follows:

1. **Linear probing remains weak and task-dependent.** Across foundation models, linear probing generally yields suboptimal results, with performance fluctuating significantly across different downstream tasks. This highlights the limited robustness and inconsistent generalization of current pre-trained representations.
2. **Foundation models are not universally better.** While fine-tuned foundation models perform well in data-rich, population-level settings, they often fail to significantly outperform compact neural networks or even classical non-neural methods in data-scarce or subject-specific settings.
3. **Scaling does not guarantee success.** Although larger foundation models have more capacity, they do not reliably outperform smaller neural decoders, particularly on complex and data-limited tasks such as motor imagery or inner speech.
4. **Simple pre-training can be effective.** Challenging the prevailing view from LaBraM (Jiang et al., 2024) that MAE on raw EEG is ineffective, we demonstrate that direct MAE pre-training on raw signals can in fact produce top-performing models.

Our findings reveal critical gaps between common pre-training assumptions and empirical decoding capabilities, underscoring the strength of simple, often-overlooked classical baselines. They also highlight the urgent need for a large-scale, standardized EEG dataset—analogous to ImageNet (Deng et al., 2009)—for both pre-training and downstream evaluation. Such a resource is essential to explore scaling behavior, rather than overfitting for marginal gains on limited data. To support transparent benchmarking and future research, we release all code and models.[1]

## 2 RELATED WORK

Recent years have seen a surge of EEG foundation models leveraging large-scale self-supervised pre-training to learn transferable neural representations (Kostas et al., 2021; Yang et al., 2023; Wang et al., 2024; Jiang et al., 2024; Wang et al., 2025). While these approaches claim improved generalization, they are typically benchmarked under limited evaluation protocols and seldom compared against classic non-neural network decoders, which often remain competitive in practical BCI tasks (Chevallier et al., 2024). For example, classical methods like CSP/FBCSP (Ramoser et al., 2000; Ang et al., 2008), Riemannian classifiers (Congedo et al., 2017), FBCCA (Chen et al., 2015), and TRCA (Nakanishi et al., 2018b) continue to provide robust baselines for their respective tasks, often outperforming compact neural network models (Schirrmeister et al., 2017; Lawhern et al., 2018) in data-limited scenarios. However, systematic comparisons across paradigms and with rigorous statistical testing are still lacking. To address these gaps, we provide the first comprehensive benchmark spanning foundation models, classic neural networks, and non-neural decoders across diverse classification and regression tasks. A detailed review of prior EEG foundation, neural, and non-neural methods is provided in Appendix C.

## 3 METHODS

### 3.1 EVALUATION PROTOCOLS

In this paper, we propose six evaluation protocols that provide a comprehensive view of model generalization, transferability, and practical utility in real-world BCI settings, as demonstrated and

---

[1] Pre-trained model weights and the corresponding implementation are publicly available at: https://github.com/LiuyinYang1101/STEEGFormer

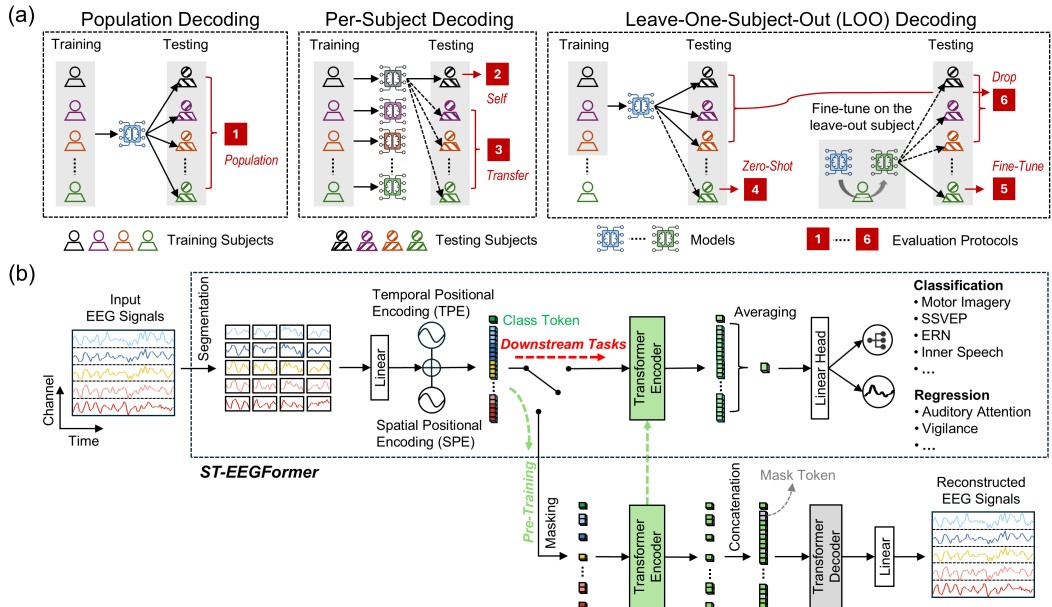

Figure 1: **(a) Graphical representation of the six evaluation protocols**. In population decoding, data from all subjects are pooled to train a single model, which is then tested on each subject individually, yielding the **(1) Population** performance. In per-subject decoding, a separate model is trained for each subject and tested both on itself—giving **(2) Per-Subject (Self)** performance—and on other subjects for **(3) Per-Subject (Transfer)** performance. In leave-one-subject-out (LOO) decoding, a population model is trained on all subjects except the LOO subject. This model is evaluated on all subjects, with its performance on the LOO subject representing **(4) LOO Zero-Shot** performance. After fine-tuning the model on the LOO subject, it is again tested on all subjects. Performance on the LOO subject after fine-tuning gives **(5) LOO Fine-Tune** performance, while changes in population subjects' performance before and after fine-tuning quantify the generalization capability, denoted as **(6) LOO Drop**. **(b) Graphical representation of the proposed *ST-EEGFormer***. During pre-training, the input EEG data are divided into segments along spatial and temporal dimensions. Each segment is tokenized through a linear projection layer, with each token receiving its corresponding temporal positional encoding (TPE) and spatial positional encoding (SPE). After randomly masking 75% of all tokens, the encoder processes the remaining unmasked tokens. The mask tokens, with their added temporal and spatial positional embeddings, are then concatenated with the encoder output to form a full set of tokens. This full set of tokens is input to a small decoder comprising a transformer followed by a linear projection layer, which reconstructs the original EEG signal. Once the model is pre-trained, only the encoder is utilized as the ST-EEGFormer model for fine-tuning on a downstream dataset.

explained in Figure 1 (a). **Population Decoding** quantifies a model's ability to capture global neural patterns that generalize across individuals, valuable to study population-level features and to applications where a single universal model is desired. **Per-Subject (Self)** reflects the traditional BCI approach, assessing performance when the model is tailored for individual users (also known as subject-dependent or subject-specific models). **Per-Subject (Transfer)** and **LOO Zero-Shot** both interrogate the transferability of learned representations: the former asks how well a subject-specific model works on another individual, while the latter tests whether a model trained on a cohort can be directly deployed to a novel subject without further adaptation. **LOO Fine-Tune** examines how effectively a population model can be adapted to a new subject through limited subject-specific training. Finally, **LOO Drop** evaluates the extent to which model fine-tuning on a new subject erodes prior knowledge—akin to catastrophic forgetting—highlighting the balance between subject adaptation and preservation of generalized population knowledge. Together, these dimensions offer a comprehensive benchmark for EEG foundation models.

## 3.2 Downstream Benchmark Tasks

We selected seven classification tasks and two regression tasks to comprehensively evaluate EEG foundation models across diverse EEG paradigms with varying complexity levels. The classification tasks include: an error-related negativity (ERN) classification task (Error-ERN) (Kueper et al., 2024); a three-class Alzheimer's Disease classification task distinguishing Alzheimer's, frontotemporal dementia, and healthy subjects (Alzheimer's) (Miltiadous et al., 2023); a four-class inner

speech classification task (Inner Speech) (Nieto et al., 2022); a four-class motor imagery task from the classic BCI Competition IV 2a dataset (BCI-IV-2A) (Tangermann et al., 2012); a seven-class upper limb motor execution task (Motor-Execution) and a seven-class upper limb motor imagery task (Motor-Imagination), both derived from the same dataset (Ofner et al., 2017); and a challenging 40-target binocular steady-state visual evoked potential (SSVEP) classification task (Binocular-SSVEP) (Yike et al., 2024). All selected datasets have been rarely explored in existing literature and were absent from the pre-training data of the cited foundation models, with the exception of BCI-IV-2A. We included this well-known motor imagery benchmark specifically to serve as a recognizable baseline, acknowledging its extensive use in prior foundation model benchmarks and its inclusion in the pre-training of our ST-EEGFormer. Additionally, the selected tasks vary significantly in complexity, ranging from the relatively simple binary ERN task to the intricate 40-target binocular SSVEP task, characterized by complex inter-modulation components arising from binocular swap stimulation (Yan et al., 2011; Sun et al., 2024). For classification tasks, top-1 accuracy (Acc1), top-2 accuracy (Acc2), balanced accuracy (BAcc), area under the curve (AUC), and Cohen's kappa coefficient (Kappa) were used as evaluation metrics for model performance. In addition to the seven tasks described above, we also benchmarked two widely used datasets—FACED (9-class emotion recognition) (Chen et al., 2023a) and TUEV (6-class EEG event classification) (Obeid & Picone, 2016)—under the conventional cross-subject zero-shot evaluation protocol.

For regression, we evaluate on two datasets. The DTU auditory attention decoding dataset targets reconstruction of the attended speech envelope from EEG recordings (Fuglsang et al., 2018). SEED-VIG (Zheng & Lu, 2017) predicts vigilance level in a virtual, monotonous driving task. Model performance is assessed using mean squared error (MSE) and Pearson correlation coefficient ($R$).

All benchmarked datasets underwent minimal and standardized pre-processing steps. EEG signals were band-pass filtered between 0.1–128 Hz and downsampled to 256 Hz if the original sampling rate is higher. During model training, epochs were further resampled to match each foundation model's native sampling rate. For most classification datasets, we applied a 5-fold cross-validation scheme within each subject. Exceptions are the BCI-IV-2A and ERN datasets, for which we strictly followed the original competition-style train/validation splits. For TUEV and FACED, we followed the conventional cross-subject split approach, where 80% subjects were used as the training subjects, and the remaining 20% as the test subjects. For the two regression datasets, we used the first 80% of each subject's recording as the training set and the remaining 20% as the test set. Detailed dataset descriptions and train/test split strategies are provided in Appendix D.

### 3.3 BENCHMARK MODELS

#### 3.3.1 FOUNDATION MODELS

For all benchmark datasets, we evaluated a range of EEG foundation models, including BENDR (Kostas et al., 2021), BIOT (Yang et al., 2023), LaBraM (Jiang et al., 2024), EEGPT (Wang et al., 2024), and CBraMod (Wang et al., 2025). In addition, we pre-trained our proposed model, ST-EEGFormer, using a straightforward MAE strategy on raw EEG signals to provide a transparent baseline (Figure 1 (b)). Detailed pre-training procedures for ST-EEGFormer are described in Appendix E, while additional intermediate benchmark experiments validating its effectiveness are presented in Appendix E.9. For all foundation models, we systematically evaluated both linear probing and fine-tuning, with implementation details provided in Appendix F.1.

#### 3.3.2 CLASSIC NEURAL NETWORK MODELS

We benchmarked two well-established convolutional neural network (CNN)-based EEG decoders: DeepConvNet (Schirrmeister et al., 2017) and EEGNet (Lawhern et al., 2018). Additionally, we included more recent, transformer-based architectures, EEG Conformer (Song et al., 2023) and CT-Net (Zhao et al., 2024), which have demonstrated state-of-the-art performance yet remain computationally simpler compared to larger foundation models. Implementation details for these models can be found in Appendix F.2.

#### 3.3.3 CLASSIC NON-NEURAL NETWORK MODELS

Depending on the specific downstream task, we benchmarked the most widely used classical models accordingly. For movement and speech classification tasks, we included CSP- (Ramoser et al., 2000) and FBCSP- (Ang et al., 2008) based pipelines (CSP-LDA, CSP-SVM, FBCSP-LDA, and FBCSP-

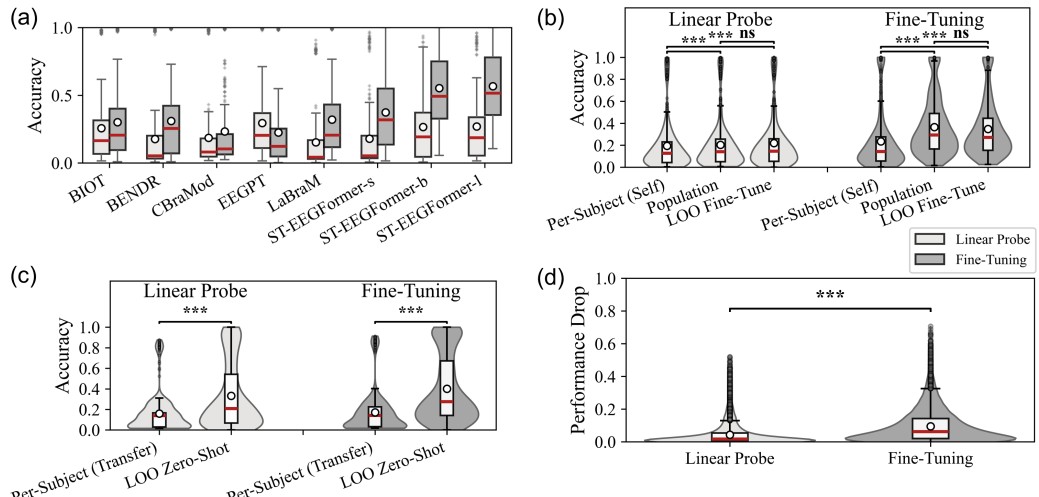

Figure 2: Comparison of linear-probing and fine-tuning foundation models across evaluation protocols. **(a) Per-model performance.** Classification accuracy for each foundation model, aggregated across all subjects and downstream tasks, under the Population, Per-Subject (Self), and LOO Fine-Tune protocols. **(b) Within-subject performance.** Comparison of all foundation models (pooled) across Per-Subject (Self), Population, and LOO Fine-Tune protocols. **(c) Cross-subject performance.** Comparison for Per-Subject (Transfer) and LOO Zero-Shot protocols. **(d) Generalization performance.** Accuracy drop on population subjects after subject-specific fine-tuning, comparing linear probing and full fine-tuning. In all panels, light grey indicates linear probing and dark gray indicates fine-tuning. Box plots show median (red line), mean (white dot), and interquartile range; violin plots illustrate the full data distribution. Asterisks in (b)-(d) indicate statistically significant differences (Wilcoxon signed-rank test, Bonferroni-corrected; ***: $p < 0.001$, ns: not significant).

SVM), as well as Riemannian geometry-based classifiers, including Minimum Distance to Mean (MDM) (Barachant et al., 2012b), Fisher Geodesic MDM (FgMDM) (Barachant et al., 2012b), and tangent space mapping (TS) with ElasticNet (TS-ElasticNet) (Corsi et al., 2022). For Alzheimer's diagnosis, we included four decoding pipelines used in the dataset paper of (Miltiadous et al., 2023), which extract Relative Band Power (RBP) features as input for Random Forest (RBP-RF), SVM (RBP-SVM), k-Nearest Neighbors (RBP-kNN), and LightGBM (RBP-LightGBM) classifiers. For ERN detection, we included xDAWN-LDA (Rivet et al., 2009), xDAWNCov-MDM (Barachant, 2014), xDAWNCov-TS-SVM (Chevallier et al., 2018), ERPCov-MDM (Barachant & Congedo, 2014), and DCPM (Xiao et al., 2020). For SSVEP target recognition, we included the two decoding models FBCCA and TRCA used in the dataset paper (Yike et al., 2024). Implementation details for these models can also be found in Appendix F.3.

## 4 RESULTS

In this section, we present findings addressing our key research questions. For detailed results per dataset, please refer to Appendix G.

### 4.1 DO FOUNDATION MODELS LEARN ROBUST REPRESENTATIONS AFTER PRE-TRAINING?

Figure 2 summarizes the performance of foundation models under both linear probing and fine-tuning across all subjects and downstream classification tasks. The corresponding summary table is listed in Appendix Tables G.1 and G.2. In Figure 2 (a), fine-tuning generally yields higher accuracy than linear probing for all foundation models, with the exception of EEGPT. Figure 2 (b) compares within-subject evaluation protocols, showing that classic per-subject training results in significantly lower accuracy than both population-level and LOO fine-tuned models, while no significant difference is observed between the latter two. Figure 2 (c) highlights transfer performance, demonstrating that population-trained models achieve substantially better cross-subject generalization (LOO Zero-Shot) than models trained individually per subject (Per-Subject (Transfer)), for both linear probing and fine-tuning. Finally, Figure 2 (d) illustrates that subject-specific fine-tuning leads to a larger accuracy drop (generalization drop) on the population subjects compared to linear probing, suggesting a tendency for fine-tuned models to "forget" information learned from the broader population (i.e., increased catastrophic forgetting).

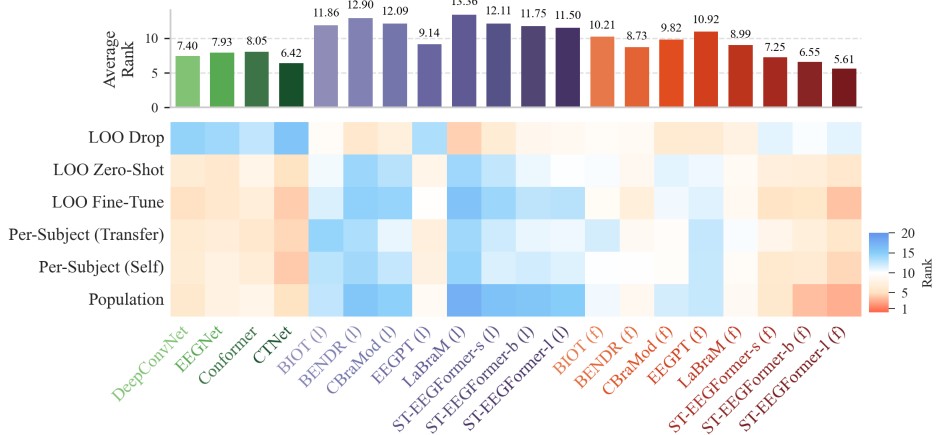

Figure 3: **Comparison of classic neural network (NN) decoders and foundation models using aggregated ranks across six evaluation protocols.** The bar plot on top shows the average rank (lower is better) across metric, subject, dataset, and protocol. Green bars denote classic NN models, while purple and red indicate linear-probed (l) and fine-tuned (f) foundation models, respectively. The heatmap at the bottom shows the rank (lower in red is better) averaged across metric, subject, and dataset per protocol.

## 4.2 DO FOUNDATION MODELS OUTPERFORM CLASSIC NN-BASED DECODERS?

Figure 3 provides a comparison between classic neural network (NN) decoders and foundation models across six distinct evaluation protocols, using aggregated rank as the performance metric. Per-dataset result can be obtained in Appendix G.3. The results reveal that, while certain linear-probed foundation models perform competitively in specific evaluation contexts, classic NN models—particularly CTNet—consistently outperform all linear-probed foundation models across most evaluation protocols, except for the LOO Drop protocol. Notably, among fine-tuned foundation models, only the largest fully fine-tuned foundation architectures (i.e., ST-EEGFormer-l) achieve performance levels equal to or surpassing those of the best-performing classic NN models. This underscores that despite their promise, foundation models do not inherently outperform well-established, classic neural network approaches.

## 4.3 DO FOUNDATION MODELS OUTPERFORM CLASSIC NON-NN-BASED DECODERS?

Figure 4 compares the best-performing decoders from each group across six evaluation protocols. A detailed per-dataset result can be found in Appendix G.4. Overall, fine-tuned foundation models—particularly the largest variant, ST-EEGFormer-l—consistently achieve the highest or comparable top performance. In contrast, classic neural and non-neural decoders show more variable results depending on the evaluation setting. Linear-probed foundation models generally underperform across all protocols. Detailed observations are as follows:

- **Population** Fine-tuned foundation models and ST-EEGFormer-l achieve the highest accuracy with no significant difference between them. Both significantly outperform classical NN and non-NN decoders, and classical NN decoders in turn outperform linear-probed foundation models.
- **Per-Subject (Self)** ST-EEGFormer-l shows a notable performance advantage, but with no statistically significant differences between the classic non-NN decoders.
- **Per-Subject (Transfer)** ST-EEGFormer-l is statistically better than every other group, whereas pairwise comparisons among the remaining models reveal no significant differences.
- **LOO Zero-Shot** Classic non-NN decoders yield the lowest mean accuracy. Classic NN decoders, fine-tuned foundation models, and ST-EEGFormer-l perform similarly, with no significant differences among them.
- **LOO Fine-Tune** Classic NN decoders perform comparably to both fine-tuned foundation models and ST-EEGFormer-l, and all three significantly outperform linear-probed foundation models.
- **LOO Drop** Classic NN decoders exhibit the most pronounced generalization drop after subject-specific fine-tuning, performing significantly worse than all foundation model groups.

## 4.4 DO EEG CLASSIFICATION MODELS TRANSFER TO REGRESSION?

Table 1 summarizes the regression performance of all neural network–based models on DTU and SEED-VIG under the LOO Zero-Shot protocol; full per-model results are provided in Ap-

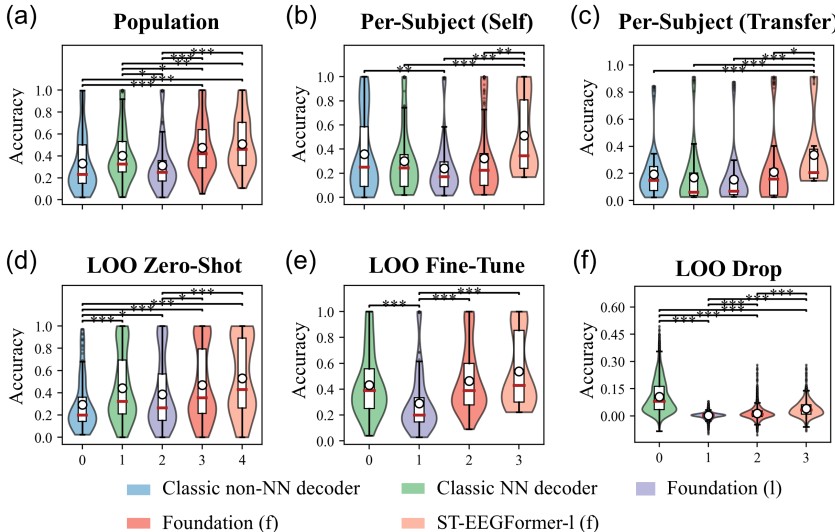

Figure 4: **Comparison of best-performing models across decoder groups and evaluation protocols.** Violin plots show the distribution of accuracy scores for the best model within each decoder group—classic non-NN decoders (blue), classic NN decoders (green), linear-probed foundation models (purple), fine-tuned foundation models (red), and the large ST-EEGFormer-l (orange)—under each evaluation protocol: **(a) Population, (b) Per-Subject (Self), (c) Per-Subject (Transfer), (d) LOO Zero-Shot, (e) LOO Fine-Tune, and (f) LOO Drop.** LOO Fine-Tune and LOO Drop are not applicable for classic non-NN decoders. Accuracy is used instead of rank to highlight meaningful performance differences. For each group, the best-performing model may vary across datasets. Statistical significance is assessed using permutation testing ($n_{resamples} = 50,000$) with Bonferroni correction (***: $p < 0.001$, **: $p < 0.01$, *: $p < 0.05$). Box plots within violins indicate the median (red line), mean (white dot), and interquartile range.

pendix G.1.10 and Appendix G.1.11. For DTU, the task is to predict a single continuous target from 1-s EEG epochs, evaluated with mean squared error (MSE) and Pearson correlation ($R$). For SEED-VIG, the task is to predict vigilance level from 5-s EEG epochs. On DTU, the best models are CTNet and the linear-probed EEGPT, both reaching $R \approx 0.05$. On SEED-VIG, EEGNet and DeepConvNet exceed $R > 0.45$.

Figure 5 further compares model families. On DTU, classic decoders outperform fine-tuned foundation models, and linear-probed foundation models also outperform fine-tuned foundation models; however, the top model from each family does not differ significantly from the others. On SEED-VIG, although the average performance of classic models exceeds that of foundation-model variants, the difference is not statistically significant.

### 4.5 CAN WE OBSERVE ANY SCALING LAW IN EEG CLASSIFICATION TASKS?

Figure 6 illustrates the relationship between NN-based EEG decoder size and both classification performance and training time. Panel (a) shows that, although there is a slight upward trend in normalized accuracy with increasing model size, the poor logarithmic fit suggests that a clear scaling law does not exist for downstream EEG classification tasks. In contrast, Panel (b) demonstrates that training time per EEG epoch grows exponentially with model size, achieving a reasonably good fit ($R^2 = 0.60$), indicating that computational cost scales much faster than accuracy improvements.

## 5 DISCUSSION

**Fine-Tuning vs. Linear Probing** Fine-tuning enables foundation models—particularly larger variants like ST-EEGFormer-l—to achieve strong performance in population-level decoding. In contrast, linear probing consistently yields poor performance across all models and evaluation protocols (except for LOO Drop), as shown in Figure 2 and Figure 3. This suggests that current pre-training strategies do not produce EEG representations that are sufficiently generalizable and discriminative across a broad range of BCI tasks. Supporting evidence is provided in Appendix G.5, where attention-weight visualizations reveal that the regions of interest change substantially after fine-tuning, underscoring the dependence of learned representations on task-specific adaptation. However, this does not suggest that foundation models fail to learn any useful representations. For

Table 1: Average LOO Zero-Shot performance across all subjects on DTU and SEED-VIG. The best and second-best MSE and Pearson scores are in bold, with the highest one surrounded by a box.

| Model | DTU MSE | DTU Pearson | SEED-VIG MSE | SEED-VIG Pearson |
|---|---|---|---|---|
| DeepConvnet | 0.999 ±0.020 | 0.039 ±0.033 | 0.055 ±0.035 | **0.452** ± **0.352** |
| EEGNet | 0.994 ±0.018 | 0.042 ±0.029 | 0.095 ±0.077 | 0.471 ± 0.296 |
| Conformer | 1.024 ±0.028 | 0.017 ±0.025 | 0.056 ±0.060 | 0.398 ±0.338 |
| CTNet | **0.993** ± **0.017** | 0.048 ± 0.032 | 0.065 ±0.052 | 0.435 ±0.313 |
| BIOT (f) | 1.315 ±0.052 | 0.006 ±0.015 | 0.065 ±0.060 | 0.360 ±0.342 |
| BIOT (l) | 1.000 ±0.019 | -0.000 ±0.018 | 0.056 ±0.042 | 0.440 ±0.283 |
| BENDR (f) | 1.616 ±0.055 | 0.010 ±0.021 | 0.050 ± 0.037 | 0.347 ±0.330 |
| BENDR (l) | 0.995 ±0.018 | 0.039 ±0.019 | 0.104 ±0.077 | 0.037 ±0.141 |
| CBraMod (f) | 1.431 ±0.065 | 0.005 ±0.024 | 0.062 ±0.067 | 0.408 ±0.361 |
| CBraMod (l) | 0.996 ±0.018 | 0.039 ±0.024 | 2.951 ±2.877 | 0.232 ±0.262 |
| EEGPT (f) | 0.994 ±0.017 | 0.043 ±0.032 | **0.051** ± **0.040** | 0.413 ±0.343 |
| EEGPT (l) | 0.994 ±0.017 | **0.047** ± **0.033** | 0.063 ±0.059 | 0.443 ±0.332 |
| LaBraM (f) | 1.638 ±0.055 | 0.011 ±0.013 | 0.057 ±0.061 | 0.421 ±0.319 |
| LaBraM (l) | 0.993 ±0.018 | 0.024 ±0.021 | 0.061 ±0.057 | 0.418 ±0.305 |
| ST-EEGformer-s (f) | 1.237 ±0.039 | 0.010 ±0.020 | 0.055 ±0.061 | 0.441 ±0.370 |
| ST-EEGformer-s (l) | 0.993 ± 0.019 | 0.016 ±0.031 | 0.064 ±0.062 | 0.372 ±0.367 |

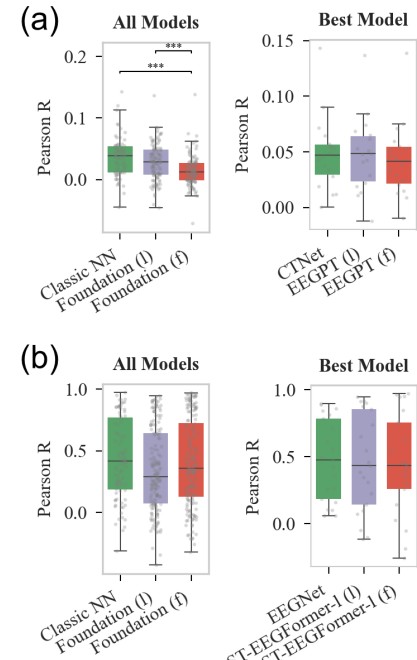

Figure 5: **LOO Zero-Shot regression performance on (a) DTU and (b) SEED-VIG.** Left panels compare classic NN decoders, linear-probed foundation models, and fine-tuned foundation models; right panels show the best model from each family. Group differences use the Mann–Whitney U test; paired top-model comparisons use the Wilcoxon signed-rank test (Bonferroni corrected, ***: $p < 0.001$).

instance, in ERN detection, all linear-probed foundation models perform relatively well, suggesting that the utility of pre-trained features may be highly task-dependent. These results imply that the effectiveness of pre-training can vary significantly across EEG paradigms, potentially due to fundamental differences in the underlying neural representations. Moreover, the strong performance of our simple yet top-performing ST-EEGFormer model indicates that complex pre-training objectives may not yield substantial downstream benefits, especially when models are fine-tuned. Fine-tuning appears to overwrite or adjust much of what is learned during pre-training, thereby narrowing the performance gap between models with simple versus sophisticated pre-training tasks.

## 5.1 STRENGTHS AND LIMITATIONS OF EEG FOUNDATION MODELS

**Data Sensitivity and Generalization Gaps** Foundation models perform best when enough training data is available, such as under population decoding, but their advantage diminishes in low-data settings like per-subject decoding. In these scenarios, classic neural networks (CTNet) remain competitive. Non-neural decoders can also achieve performance comparable to foundation models in certain evaluation protocols (e.g., Per-Subject (Self)), although they perform statistically worse than NN-based models in LOO Zero-Shot. Interestingly, while foundation models often achieve higher mean accuracy than classic NN models in LOO Zero-Shot and LOO Fine-Tune settings (Figure 4 (d-e)), these differences are not statistically significant. These findings highlight the value of simpler baselines, which are frequently overlooked in current foundation model research.

**On the Transferability of EEG Classification Models to Regression** Our regression case study underscores limits to the transferability of EEG foundation model representations. On the conventional SEED-VIG dataset, fine-tuned foundation models are competitive and often rank highest overall (Figure G.12); however, under specific protocols—such as LOO Zero-Shot (Table 1)—classic neural networks still produce the best results. In contrast, on the more challenging DTU auditory dataset—where foundation models have not previously been benchmarked—classic decoders out-

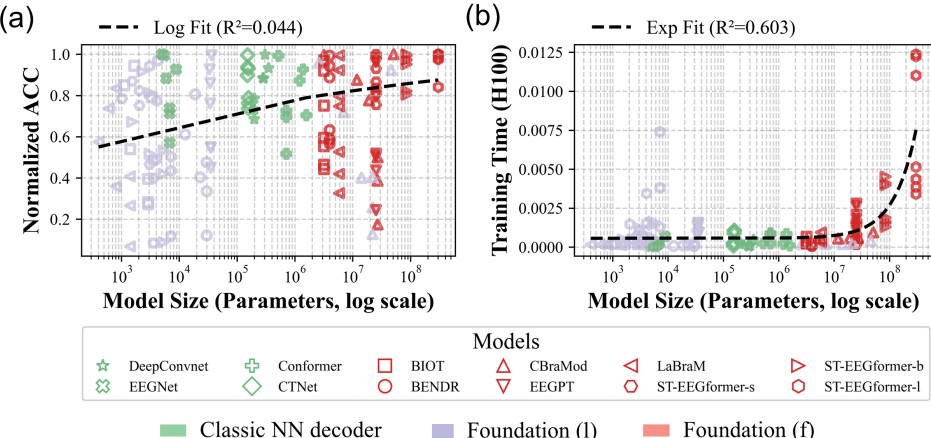

Figure 6: **Cross-model scaling trends in NN-based EEG decoders. (a) Normalized accuracy versus total trainable parameters.** For each dataset, accuracies are rescaled so the dataset's best model equals 1. A pooled logarithmic fit $a \ln(\ln x) + b$ is applied; the coefficient of determination ($R^2$) is reported above the panel. **(b) Training time (S) per EEG epoch versus total trainable parameters,** computed on a single NVIDIA H100 as $\mathrm{Time\ per\ Batch}/\mathrm{Batch\ Size}$. An exponential fit $a\,e^{bx} + c$ is applied to the pooled data, with $R^2$ shown above the panel. Each point denotes a distinct model (unique marker); colors indicate the three model families.

perform both fine-tuned and linear-probed foundation models. Taken together, these findings suggest that feature representations for regression are not universally robust across tasks and protocols.

**Scaling Behavior and Task Dependence** Figure 6 (a) indicates that a simple "bigger-is-better" rule does not generally hold across EEG decoders. Within a single architecture, however, we do observe conventional scaling: for ST-EEGFormer, the large variant outperforms the base and small variants (Figure 2). When we widen the comparison to all neural networks in the benchmark, large foundation models often deliver performance that is merely comparable to classic, smaller NN decoders. We report this intentionally as a benchmark-level finding rather than a tuning artifact: if a large foundation model only matches a compact classic model, then within-family scaling gains have limited practical value—especially given the substantial computational and training costs of large models. A primary factor is downstream data scarcity: most downstream BCI datasets include fewer than 50 subjects, which constrains the benefits of large foundation models. This echoes results in related modalities, where an increasing number of training data leads to substantial performance gains for sEMG decoding (cf., Figure 2: over 40% error rate when training with less than 50 subjects, whereas below 10% error rate when scaling up the training subjects to more than 6,000) (Kaifosh et al., 2025). Task dependence further compounds the picture. Easier paradigms (e.g., ERN) reach near-ceiling accuracy ($\approx 99.9\%$) even with small models, leaving little headroom for scaling; harder paradigms (e.g., inner speech) show minimal improvement regardless of model size. Collectively, these observations highlight fundamental limits in EEG decoding: foundation models can help, but they are not a universal remedy in data-scarce regimes. Progress will require not only architectural advances and task-aware objectives, but also substantially larger and more diverse datasets, alongside a deeper understanding of the theoretical and physiological limits of scalp EEG.

## 5.2 HIDDEN IMPLEMENTATION FACTORS

Foundation model performance depends not only on pre-training but also on downstream design and training choices that are often underreported.

**Head Capacity under Linear Probing** EEGPT and CBraMod employ multi-layer classification heads even in settings described as linear probing, effectively increasing capacity relative to approaches that freeze the backbone and attach a single linear layer. This likely contributes to EEGPT's strong linear-probing performance and illustrates how the term "linear probing" can conceal substantial variation in head complexity. In Appendix H.1, we show that more complex heads can significantly improve linear-probing accuracy.

**Token Fusion Strategy** The way token features are aggregated before the head is likewise critical. CBraMod feeds all tokens into its head; LaBraM supports either class-token or full-token fusion; ST-EEGFormer adopts a ViT-style average-token fusion by default. These choices alter the effective receptive field and the information delivered to the head. As demonstrated in Appendix H.1, simple fusion coupled with a single linear layer typically underperforms more expressive heads (e.g., multi-

layer or full-token designs), even when backbone and data are held fixed. Additional overlooked factors are discussed in Appendix C.4.

**Takeaway** Seemingly minor implementation details, such as head depth/width, fusion scheme, can induce substantial performance gaps and complicate cross-model comparisons. We therefore recommend that future foundation model studies (i) *explicitly specify* head architecture and fusion strategy for every setting, and (ii) adopt a *reporting checklist* to standardize linear-probing vs. fine-tuning protocols and ensure fair, reproducible benchmarks.

### 5.3 NEED FOR FAIR AND REPRODUCIBLE BENCHMARKING

Current evaluation practices often rely on selective downstream tasks and evaluation protocols, enabling overly optimistic claims. Statistical testing is frequently absent or insufficiently emphasized. To truly assess progress, foundation models must be compared against strong classic baselines across diverse tasks and protocols, with rigorous significance testing.

### 5.4 CALL FOR COMMUNITY-WIDE COLLABORATION

The current landscape of EEG foundation models remains fragmented: models employ different pre-training strategies, evaluation protocols, and reporting practices, making direct comparisons difficult and limiting transparency. At the same time, the small size and limited diversity of downstream EEG datasets constrain large-scale benchmarking and analyses of generalization. To advance the field meaningfully, we call for a coordinated community effort to: 1) Develop and share large-scale EEG datasets suitable for both pre-training and standardized evaluation. 2) Establish common evaluation protocols and strong, consistent baselines to enable fair and reproducible comparisons. Without such shared resources and benchmarking standards, progress in EEG foundation models risks being incremental, domain-specific, and unlikely to generalize to new subjects, tasks, or datasets.

## 6 CONCLUSION

This study presents a comprehensive benchmark of EEG foundation models across diverse tasks, evaluation protocols, and model types. In addition, we introduce ST-EEGFormer, a simple yet effective foundation model based on the Vision Transformer architecture and pre-trained solely using masked autoencoding. Our results demonstrate that while foundation models can offer clear advantages in data-rich settings, they are not universally superior—particularly in data-scarce scenarios such as per-subject decoding. In such cases, classic neural and non-neural models remain strong contenders and should not be underestimated. Moreover, the generally poor performance of linear probing and the sensitivity to downstream implementation details highlight the need for greater transparency and standardization in evaluation. Moving forward, progress in EEG foundation modeling will depend on community-wide efforts to establish large-scale datasets and adopt fair, statistically rigorous benchmarking practices.

ACKNOWLEDGMENTS

*L.Y. is supported by the Research Foundation – Flanders (FWO) grant 1S65622N.
*Q.S. is supported by the China Scholarship Council (no. 202206050022).
*A.L. is supported by the China Scholarship Council (no. 202404910389).
*M.M.V.H. is supported by research grants received from Horizon Europe's Marie Sklodowska-Curie Action (grant agreement No. 101118964), Horizon 2020 research and innovation programme under grant agreement No. 857375, the special research fund of the KU Leuven (C24/18/098), the Belgian Fund for Scientific Research – Flanders (G0A4321N, G0C1522N, G031426N), and the Hercules Foundation (AKUL 043).
*The resources and services used in this work were provided by the VSC (Flemish Supercomputer Center), funded by the Research Foundation - Flanders (FWO) and the Flemish Government.

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

APPENDIX

## A CLARIFICATION ON THE USE OF LARGE LANGUAGE MODELS (LLMs)

In this work, Large Language Models (LLMs) (GPT-4o and GPT-5) were used as auxiliary tools to support writing and implementation. For manuscript preparation, LLMs were used for grammar checking and text refinement only. A typical prompt example is:
"Refine the following paragraph to improve clarity and conciseness while preserving technical meaning."
For experiments, LLMs were occasionally consulted to assist in coding tasks such as generating a specific function using Python. Importantly, all code produced with LLM assistance was carefully reviewed, validated, and tested by the authors to ensure correctness. Thus, the scientific contributions, experimental design, and data analyses presented in this work are entirely the responsibility of the authors.

## B REPRODUCIBILITY

We've open-sourced the project on GitHub, with detailed documentation and usage examples to help the community adopt it.[2] The release includes:

- **Pre-training framework:** source code and pre-trained model weights for the proposed ST-EEGFormer.
- **Downstream benchmarks:** training and evaluation code for both foundation models and classic neural networks.
- **Classical baselines:** implementations of non-neural methods (e.g., FBCSP, TRCA, CCA, Riemannian classifiers) used in this benchmark.
- **Utilities:** example SLURM scripts for job submission, reflecting our HPC-based experimental setup, to help others reproduce large-scale training efficiently.

## C RELATED WORK

### C.1 EEG FOUNDATION MODELS

EEG foundation models have attracted significant attention as they provide generalizable EEG representations transferable across various tasks and datasets. BENDR (Kostas et al., 2021) was among the earliest to apply masked pre-training and contrastive learning to EEG data, utilizing a transformer backbone to learn from large-scale unlabeled datasets. Building upon this approach, BIOT (Yang et al., 2023) and LaBraM (Jiang et al., 2024) adopted similar transformer architectures combined with extensive masked pre-training strategies. Specifically, BIOT introduced flexible channel tokenization to enhance cross-dataset transferability, while LaBraM proposed "neural tokens" and argued that straightforward masked autoencoding on raw EEG data failed to converge effectively, thereby encouraging further exploration into more sophisticated pre-training objectives instead of a simple raw-signal reconstruction task. More recently, CBraMod (Wang et al., 2025) advanced LaBraM's paradigm by employing a criss-cross transformer architecture alongside local and global masked reconstruction losses, significantly enhancing the transferability of EEG feature representations. Similarly, EEGPT (Wang et al., 2024) adds spatio-temporal representation alignment, constructing a self-supervised task on EEG representations with high SNR and rich semantic information instead of raw signals. A detailed comparison of these foundation models is presented in Appendix Table C.1

Despite methodological differences, most existing EEG foundation models share the underlying assumption that large-scale, self-supervised pre-training leads to better generalization in downstream tasks. However, these models are rarely evaluated against traditional non-neural baselines and are typically tested using limited evaluation protocols—such as population decoding or leave-one-out

---

[2]Pre-trained model weights and the corresponding implementation are publicly available at:
`https://github.com/LiuyinYang1101/STEEGFormer`

zero-shot decoding. As a result, prior studies often report isolated high-performing numbers without statistical verification, leaving the true advantages of foundation models largely unsubstantiated.

## C.2 CLASSIC NEURAL NETWORK MODELS

Parallel to the advancements in foundation models, the BCI community has actively developed compact neural architectures specifically tailored for EEG decoding. Early deep-learning models such as DeepConvNet (Schirrmeister et al., 2017) and EEGNet (Lawhern et al., 2018) demonstrated that convolutional architectures can effectively capture spatio-temporal EEG features with relatively low complexity. More recently, hybrid designs like EEG Conformer (Song et al., 2023) and CTNet (Zhao et al., 2024)—which combine convolutional modules with Transformer-based attention—have achieved state-of-the-art performance on diverse tasks while maintaining significantly lower computational demands than foundation models. Due to the inherent coupling of EEG signals with the specific tasks performed by participants, many existing neural network architectures are tailored to incorporate explicit task-specific priors (or knowledge). For instance, in tasks involving emotion, fatigue, or Alzheimer's disease etc., where cross-brain-region information transfer is prevalent, graph-based architectures have proven effective in modeling this underlying functional connectivity (Zhong et al., 2020; Ding et al., 2022; Klepl et al., 2024). Conversely, for SSVEP detection, where the brain response to the stimulation is relatively well-defined, stimulation priors are incorporated into the neural network architecture design, leading to improved decoding performance (Li et al., 2020; Zhang et al., 2022; Deng et al., 2024). These developments highlight that well-designed compact models can capture rich EEG representations across a variety of BCI tasks, often matching or exceeding the performance of larger models while remaining more efficient and interpretable.

## C.3 CLASSIC NON-NEURAL NETWORK MODELS

Classic machine learning models—those not based on neural networks—remain the most widely used for EEG decoding, though they are often overlooked in recent foundation model research. According to a recent comprehensive study on EEG decoder reproducibility, classic approaches often exhibit superior performance, even outperforming neural networks on various BCI tasks (Chevallier et al., 2024). For instance, Common Spatial Patterns (CSP) (Ramoser et al., 2000), filter bank CSP (FBCSP) (Ang et al., 2008), and Riemannian geometry-based classifiers (Congedo et al., 2017), continue to demonstrate competitiveness in various motor imagery BCI applications. In the context of SSVEP decoding, traditional methods such as filter bank Canonical Correlation Analysis (FBCCA) (Chen et al., 2015), Task-Related Component Analysis (TRCA) (Nakanishi et al., 2018b), and Spatiotemporal Beamforming (Wittevrongel & Van Hulle, 2017) remain dominant due to their robustness and high efficiency with limited training data. For event-related potential (ERP) classification, xDAWN is widely used as a pre-processing step for improving signal-to-noise ratio (Rivet et al., 2009). Additionally, classic machine learning techniques employing carefully designed EEG features and explainable classifiers such as Linear Discriminant Analysis (LDA), Random Forest (RF), and Support Vector Machines (SVM) are still popular in scenarios characterized by limited data availability and a preference for interpretability, such as Alzheimer's diagnosis (Miltiadous et al., 2023).

## C.4 METHODOLOGICAL GAPS AND IMPLEMENTATION ISSUES IN PRIOR EEG FOUNDATION MODELS

While Table C.1 outlines the key methodological differences across EEG foundation models, several important implementation details have been largely overlooked in prior work. For instance, as discussed in Section 5.2, the choice of token fusion strategy can substantially affect downstream performance, yet it is rarely examined systematically. During our benchmarking, a careful review of publicly available code revealed additional overlooked factors and inconsistencies in implementation, which may partly explain discrepancies in reported results. We summarize and discuss these issues below to highlight the importance of transparent and reproducible evaluation practices.

Beyond the differences summarized in Table C.1, several methodological details warrant emphasis. EEGPT employs an encoder that restricts attention to tokens within the same time step, treating tokens from different time steps as independent. This design prevents the model from explicitly

capturing temporal dependencies. Furthermore, EEGPT discards all patch tokens after the encoder, relying solely on the [CLS] token for both pre-training and downstream tasks. In downstream applications, an additional convolutional layer is inserted before the encoder to remap input channels, which may inadvertently diminish the role of channel embeddings.

For CBraMod, positional information is introduced through ACPE embeddings, constructed via two-dimensional convolutions over the spatio-temporal neighborhood of each patch. However, the model was pre-trained exclusively on a fixed 19-channel dataset. Consequently, when applied to datasets with different channel configurations, it requires fine-tuning to learn new channel embeddings and representations. This limitation is reflected in their reported results: as shown in Table 4 of the original paper, the downstream performance differences between the pre-trained model and the model trained directly on downstream tasks without pre-training are very marginal, suggesting limited transferability of the learned representations.

BIOT, in contrast, adopts a bipolar montage during pre-training rather than a standard single-channel montage. As a result, downstream datasets must either be remapped into a consistent bipolar montage or supplemented with a convolutional layer before the encoder to automatically learn this mapping.

Finally, we also identified potential implementation issues in the official EEGPT codebase that were not documented in the paper. Specifically, the authors employed the `torcheeg` library to extract EEG segments from datasets such as TSU, M3CV, and SEED. By default, this library generates 1-second epochs, which were subsequently interpolated to match the longer durations reported in the paper (e.g., 4 s for TSU and 10 s for SEED).

## C.5 GAPS AND MOTIVATION

Despite recent advances, current EEG foundation model studies remain limited in scope. Most focus narrowly on tasks like leave-one-out zero-shot decoding, often overlooking real-world scenarios such as per-subject evaluation and omitting comparisons with classical non-neural methods. Statistical testing is rarely performed, and regression tasks are largely unexplored. To address these gaps, we present a comprehensive, statistically rigorous benchmark across diverse classification and regression tasks, comparing foundation models against both classical and compact neural decoders. Notably, we show that top performance can be achieved with a simple MAE-pre-trained model, challenging the need for overly complex pre-training strategies.

Table C.1: **Comparison of large-scale EEG foundation models.** Columns detail: pre-training datasets, pre-training method, total model size, downstream datasets, task type, downstream optimization strategy, evaluation protocol coverage (using our six-dimension taxonomy), and baseline/benchmark comparison models.

| Model | pre-training Dataset Size | pre-training Method | Model Size | Downstream Datasets | Downstream Task Type | Downstream Optimization | Evaluation protocols | Comparison Models |
|---|---|---|---|---|---|---|---|---|
| BENDR (2021) | TUEG (Obeid & Picone, 2016): ~10k subjects, ~1.5TB raw EEG | Masked segment prediction with contrastive loss (wav2vec2-inspired) | ~3M (Encoder) | Multiple BCI/EEG: P300, SMR MI, Sleep (SSC) | Classification (ERP, MI, ERN, sleep stage) | Fine-tuning, linear probing | Population decoding, LOO zero-shot | None |
| BIOT (2023) | ~5M sleep EEG segments (SHHS) (Zhang et al., 2018), ~5M resting EEG (PREST) | Tokenization + channel/patch masking; contrastive prediction between original and perturbed tokens | 3.2M | CHB-MIT (Tran et al., 2022), IIIC (Ge et al., 2021), TUAB (Lopez et al., 2015), TUEV (Harati et al., 2015) | Classification (seizure, abnormal EEG) | Fine-tuning | Population decoding, LOO zero-shot | Classic NN models |
| LaBraM (2024) | ~20 public/proprietary EEG datasets; ~2,534h | VQ neural tokenizer (Fourier spectrum pred.) + masked token prediction (Transformer) | 5.8M/46M/369M (base/large/huge) | TUAB (Lopez et al., 2015), TUEV (Harati et al., 2015), SEED-V (Liu et al., 2022), MoBI (gait) (He et al., 2018) | Classification (seizure, abnormal EEG, emotion), Regression (gait) | Fine-tuning, linear probing | Population decoding, LOO zero-shot | Classic NN models, BIOT |
| EEGPT (2024) | 5 public datasets (~700k+ trials/segments) | Dual self-supervised: spatio-temporal alignment + mask reconstruction | 4.7M/25M (tiny/large) | TUAB (Lopez et al., 2015), TUEV (Harati et al., 2015), BCIC-IV 2A/2B (Tangermann et al., 2012), ERN (Margaux et al., 2012), PhysioNet (Goldberger et al., 2000), Sleep-EDFx (Kemp et al., 2000) | Classification (seizure, abnormal EEG, emotion, MI, Sleep) | Linear probing (frozen encoder), small transformer on top for long seqs | Population decoding, LOO zero-shot | Classic NN models, BENDR, BIOT, LaBraM |
| CBraMod (2025) | TUEG latest (Obeid & Picone, 2016): 14,987 subjects, ~27,062 h | Patch-based masked reconstruction; criss-cross transformer (spatial & temporal attn.); dual-branch (time/freq) encoder; ACPE | 4M | 12 datasets: FACED (Chen et al., 2023b), SEED-V (Liu et al., 2022), PhysioNet (Goldberger et al., 2000), SHU-MI (Ma et al., 2022), ISRUC (Khalighi et al., 2016), CHB-MIT (Tran et al., 2022), Imagined speech (Jeong et al., 2022), Mumtaz2016 (Mumtaz, 2016), SEED-VIG (Min et al., 2017), MentalArithmetic (Zyma et al., 2019), TUEV (Harati et al., 2015), TUAB (Lopez et al., 2015) | Classification (emotion, MI, event, abnormal, seizure, ERP, SSVEP, sleep), Regression (fatigue detection) | Fine-tuning | Population decoding, LOO zero-shot | Classic NN models, BIOT; LaBraM |

## D    BENCHMARK DATASETS

In this section, we introduce all downstream benchmark datasets used in this study and their pre-processing and data-split strategies.

### D.1    BENCHMARKED DOWNSTREAM DATASETS

#### D.1.1    BCI COMPETITION IV-2A (4-CLASS MOTOR IMAGERY)

The **BCI Competition IV-2a** dataset (Tangermann et al., 2012) is a widely used benchmark for motor imagery (MI) classification. It contains EEG recordings from *9 subjects* performing four distinct imagined movements: left hand, right hand, both feet, and tongue. Each subject participated in two sessions on separate days, each consisting of 288 trials (576 trials in total per subject). EEG was recorded from 22 electrodes (international 10–20 system) at 250 Hz, with each trial comprising a 4-second MI period following a visual cue. In the original competition design, the first session is designated as the training set and the second as the test set. This dataset's well-controlled protocol and multi-class setting (4 classes) have made it a canonical benchmark for MI decoding. In our study, we strictly follow the original competition split: models are trained on the first session and evaluated on the second.

#### D.1.2    UPPER-LIMB MOTOR EXECUTION/IMAGERY DATASET (7-CLASS MOTOR EXECUTION/IMAGERY)

The **Upper-Limb Motor Execution/Imagery** dataset (Ofner et al., 2017) contains EEG recordings from *15 healthy subjects* performing both executed and imagined upper-limb movements. Each subject completed two sessions on separate days: one with actual **motor execution (ME)** and one with **motor imagery (MI)** of the same tasks. In both conditions, subjects performed six distinct sustained movements of the right arm (elbow flexion/extension, forearm supination/pronation, and hand open/close), plus a rest condition, yielding seven classes in total. Tasks were visually cued, and subjects either executed the movement (ME) or vividly imagined it (MI) for several seconds per trial. Each session comprised 10 runs of 42 trials, resulting in 60 trials per class (420 trials per session). EEG was recorded from 61 electrodes (motor coverage) at 512 Hz. This dataset enables multi-class decoding across overt and imagined movements, while also supporting analysis of execution–imagery differences. In this study, we apply a 5-fold cross-validation strategy within each subject and modality.

#### D.1.3    INNER SPEECH EEG DATASET (4-CLASS INNER SPEECH)

The **Thinking Out Loud** dataset (Nieto et al., 2022) is an open-access benchmark for inner speech classification. It contains EEG recordings from *10 native Spanish speakers* instructed to silently imagine saying four command words ("arriba, abajo, izquierda, derecha" meaning up, down, left, right). For comparison, the same participants also performed overt speech (speaking the words aloud) and a visual imagery control task, though here we focus only on the inner speech condition. Each subject completed multiple runs across three sessions, yielding about 200 trials per word for inner speech. EEG was acquired using a 136-channel system (128 scalp electrodes plus 8 EOG/EMG channels) at 1024 Hz. During inner speech trials, participants were asked to repeatedly imagine pronouncing the target word in their own voice while avoiding overt articulation. This dataset provides a four-class classification challenge in the inner speech paradigm, offering a benchmark for developing BCIs aimed at natural, speech-based communication. In this study, we adopt a 5-fold cross-validation strategy within subjects to train and evaluate models.

#### D.1.4    ERROR-RELATED EEG DATASET (2-CLASS ERN CLASSIFICATION)

The **Error-Related EEG Dataset** (Kueper et al., 2024) captures brain responses to unexpected movement errors during human–robot interaction. *Eight subjects* wore an active robotic orthosis on the right arm, which guided elbow flexion/extension movements. In ~20% of trials, the orthosis briefly (250 ms) moved in the opposite direction before returning to the correct trajectory, inducing a detectable error. Subjects remained passive but reported errors by squeezing a ball with the left hand. Each subject performed 10 runs of 30 trials (15 flexion, 15 extension), with 6 error trials

per run, resulting in $\sim$300 total trials and 60 error events per subject. EEG was recorded from 64 scalp channels (extended 10–20 montage, Brain Products LiveAmp) at 500 Hz, along with 8 EMG channels on arm muscles. Impedances were kept below 5 kΩ. The dataset is designed to study **Error-Related Potentials (ErrPs)**, providing a binary classification task (error vs. correct movement). In this study, we use 1-s EEG epochs following the error onset and adopt the same train-test split strategy as the original work for evaluation.

### D.1.5  Binocular Dual-Frequency SSVEP Dataset (40-class binocular SSVEP)

The **Binocular SSVEP Dataset** (Yike et al., 2024) introduces a novel paradigm where distinct flickering stimuli are presented separately to the left and right eyes using a polarized light system. The benchmarked subset corresponds to the *binocular-swap* experiment, involving *35 healthy subjects*. A total of 40 visual targets were defined by different binocular frequency combinations. In each trial, participants fixated on a single target for 2 s, with each target repeated 5 times. EEG was recorded with a 64-channel Neuroscan Quik-Cap, following the international 10–20 electrode placement system. This dataset provides a large-scale multi-class SSVEP benchmark under binocular stimulation. In this study, we adopt a 5-fold cross-validation strategy, ensuring that held-out test trials are never used during training to avoid data leakage.

### D.1.6  Alzheimer's Diagnosis EEG Dataset (3-class classification)

The **Alzheimer's EEG Dataset** (Miltiadous et al., 2023) provides resting-state EEG recordings for studying **Alzheimer's disease (AD)** and **Frontotemporal dementia (FTD)**. It includes *88 elderly subjects*: 36 with probable AD, 23 with FTD, and 29 cognitively healthy age-matched controls. EEGs were collected during routine clinical assessments, with each recording consisting of 12–14 minutes of eyes-closed resting-state activity. Signals were acquired from 19 scalp electrodes (10–20 system) at 500 Hz and are shared in BIDS format with preprocessing and accompanying metadata, including Mini-Mental State Exam (MMSE) scores. This dataset supports both binary (e.g., AD vs. Control) and multi-class classification (AD vs. FTD vs. Control), offering a valuable resource for developing machine learning models for early **dementia diagnosis** from non-invasive EEG. In this study, we adopt a leave-one-subject-out split, reflecting the clinical goal of diagnosing an unseen patient.

### D.1.7  FACED (9-class emotion recognition)

The **Finer-grained Affective Computing EEG Dataset (FACED)** (Chen et al., 2023a) is a large-scale benchmark for multi-class EEG-based emotion recognition. It contains EEG recordings from *123 healthy subjects* who watched 28 emotion-eliciting video clips spanning **nine discrete emotion categories**: amusement, inspiration, joy, tenderness; anger, fear, disgust, sadness; and neutral. After each clip, participants provided self-reported ratings on eight target emotions plus valence, arousal, liking, and familiarity. EEG was recorded from 32 electrodes (10-20 system) at 250 or 1000 Hz and subsequently standardized to 250 Hz; for each trial, the last 30 s of the video were retained and pre-processed, and both raw and feature-level (DE/PSD) representations are released. In this study, we treat FACED as a **9-class emotion recognition** benchmark and adopt a conventional cross-subject zero-shot protocol on the processed $250Hz$ data, where models are trained on 80% of subjects and evaluated on the remaining 20% test subjects.

### D.1.8  TUH EEG Events Corpus (TUEV, 6-class EEG events)

The **TUH EEG Events Corpus (TUEV)** (Obeid & Picone, 2016) is a clinically collected benchmark derived from the Temple University Hospital EEG (TUEG) database. It consists of short EEG segments extracted from routine clinical recordings and labeled as one of **six event types**: spike and sharp wave (SPSW), generalized periodic epileptiform discharges (GPED), periodic lateralized epileptiform discharges (PLED), eye movement (EYEM), artifact (ARTF), and background (BCKG). In common benchmark settings, THE EEG signals are recorded at $250Hz$ with 23 channels. The corpus has become a standard dataset for automatic detection of epileptic discharges and general EEG event classification in clinical environments. In this study, we formulate TUEV as a **6-class EEG event classification** task and follow the conventional cross-subject protocol provided

Table D.1: Summary of benchmarked EEG datasets used in this study.

| Dataset | #Subj. | Classes / Target | EEG Ch. | $f_s$ (Hz) | Trial Dur. | Trials / Subj. | Task Type |
|---|---|---|---|---|---|---|---|
| BCI-IV-2A | 9 | 4 classes (L/R hand, feet, tongue) | 22 | 250 | 4 s | 576 | Motor Imagery |
| Upper-limb ME/MI | 15 | 7 classes (6 arm movements + rest) | 61 | 512 | 3 s | 420 | Motor Exec./Imagery |
| Inner Speech | 10 | 4 classes (silent words) | 128 | 1024 | 3 s | ∼200/word | Inner Speech |
| Error-related (ErN) | 8 | 2 classes (error vs. correct) | 64 | 500 | 1 s epoch | ∼300 | Error Monitoring |
| Binocular SSVEP | 35 | 40 targets (binocular freq. combos) | 64 | 250 | 2 s | 200 | SSVEP |
| Alzheimer's/FTD/HC | 88 | 3 classes (AD, FTD, HC) | 19 | 500 | 12–14 min | 1 session | Clinical Diagnosis |
| FACED | 123 | 9 classes (discrete emotions) | 32 | 250 | 30 s | 28 | Emotion Recognition |
| TUEV (Events) | 290 | 6 classes (SPSW, GPED, PLED, EYEM, ARTF, BCKG) | 23 | 250 | variable (event segments) | EEG Event Detection | |
| DTU Cocktail Party | 18 | Regression (speech envelope) | 64 | 512 | ∼50 s | 6 trials (∼30 min) | Auditory Attention |
| SEED-VIG | 23 | Regression (vigilance level) | 17 | 200 | 5 s | ∼750 | Vigilance |

by the dataset, training on the official training partition and evaluating on the held-out evaluation partition without subject overlap.

### D.1.9 AUDITORY ATTENTION (DTU "COCKTAIL PARTY" DATASET, REGRESSION)

The **DTU "Cocktail Party" Dataset** (Fuglsang et al., 2018) is a benchmark for **auditory attention decoding (AAD)**, formulated here as a regression task. It contains EEG recordings from *18 subjects* listening to continuous speech in a dual-speaker setting. In each ∼50 s trial, two concurrent speech streams (one male, one female, presented from different spatial locations) were played, and subjects were instructed to **attend to one speaker** while ignoring the other. Some baseline trials featured only a single speaker. EEG was recorded at 512 Hz with a 64-channel BioSemi system (plus EOG), and the speech waveforms of both speakers were simultaneously recorded and temporally aligned with the EEG. Each subject contributed ∼30 min of data across six trials with varying attention conditions.

Unlike categorical BCI datasets, this dataset provides a **continuous regression target**: the temporal envelope of the attended speech. Models are evaluated by reconstructing the attended envelope from EEG and comparing it against the true attended vs. unattended audio streams. This paradigm captures realistic neural tracking of continuous stimuli and assesses a model's ability to decode selective attention in naturalistic listening environments. In this study, we regress EEG to the attended auditory envelope, using the first 80% of each subject's recording for training and the remaining 20% for testing.

### D.1.10 SEED-VIG (VIGILANCE ESTIMATION)

The **SEED-VIG** dataset (Zheng & Lu, 2017) is a multimodal benchmark for **driver vigilance estimation**. It was collected from *23 participants* performing a ∼2 h sustained simulated driving task on a monotonous four-lane highway designed to induce fatigue. EEG and forehead EOG were recorded with a Neuroscan system; for EEG, 17 scalp electrodes were placed according to the international 10–20 system, and signals were downsampled to 200 Hz. Continuous vigilance labels in the range [0, 1] were derived from PERCLOS (percentage of eyelid closure). SEED-VIG has become a standard benchmark for EEG-based drowsiness and vigilance estimation in automotive safety research. In this study, we use SEED-VIG as a **regression** benchmark.

To ensure a comprehensive evaluation of EEG decoders, we benchmark across datasets spanning a broad range of paradigms, including motor imagery/execution, inner speech, error monitoring, visual (SSVEP), auditory attention, and clinical diagnosis. This diversity captures both laboratory and real-world BCI scenarios, testing models under varying cognitive tasks, electrode montages, and recording conditions. A summary of all benchmarked datasets is provided in Table D.1.

### D.2 DATA PRE-PROCESSING

We apply minimal and standardized pre-processing across all benchmarked datasets to ensure comparability while preserving raw signal characteristics. EEG signals are band-pass filtered between 0.1–128 Hz and notch filtered at the power-line frequency using the `mne.filter` module. All datasets are downsampled to a baseline rate of 256 Hz, chosen to align with EEGPT (which natively operates at 256 Hz). For foundation models requiring lower sampling rates, additional resampling is applied at data-fetch time using `mne.resample`, followed by their own normalization schemes. A summary of preprocessing steps is provided in Table D.2. Note that for the Binocular SSVEP dataset, recordings originally sampled at 250 Hz are kept at 250 Hz, only being upsampled to 256 Hz

Table D.2: Default preprocessing pipeline applied across all datasets.

| Step | Description |
|---|---|
| Band-pass filtering | 0.1–128 Hz using `mne.filter` |
| Notch filtering | Power-line frequency (50/60 Hz) using `mne.filter` |
| Downsampling | 256 Hz (baseline rate) |
| Resampling | Model-specific rate via `mne.resample` |

Table E.1: Details of ST-EEG-MAE variants, all with an EEG segment (patch) size of 16 samples and a mask ratio of 0.75.

| Model | Encoder layers | Encoder embed size | Encoder MLP size | Encoder heads | Decoder layers | Decoder embed size | Decoder MLP size | Decoder heads | Params |
|---|---|---|---|---|---|---|---|---|---|
| small | 8 | 512 | 2048 | 8 | 4 | 384 | 1536 | 16 | 32.7M |
| base | 12 | 768 | 3072 | 12 | 8 | 512 | 2048 | 16 | 110.9M |
| large | 24 | 1024 | 4096 | 16 | 8 | 512 | 2048 | 16 | 328.4M |

when training EEGPT. Additionally, the Binocular SSVEP and DTU datasets undergo task-specific segmentation, described below.

**Binocular SSVEP** To increase the number of training examples and to evaluate asynchronous classification, we segment each 2-s trial into overlapping windows. A sliding window of 1 s with a 0.1 s step size is used, yielding 11 segments per trial. With 40 targets and 5 repetitions each, this results in $11 \times 40 \times 5 = 2200$ samples. In each fold, 1760 samples are used for training and 440 for testing. The first segment of each trial (stimulus onset) is considered a synchronous trial (40 in total), while the remaining 400 segments serve as asynchronous trials.

**DTU (Auditory Attention)** For regression benchmarking, we adopt a simple formulation: given a 1-s EEG segment, the model predicts one sample of the attended speech envelope. Recordings are segmented into 3-s windows with a 0.1-s step size. Within each window, the model is trained to reconstruct the envelope using the abovementioned 1-s sliding window approach. The training objective combines mean squared error (MSE) and Pearson correlation loss between the predicted and true envelopes.

# E ST-EEGFOMER

In this section, we introduce the proposed ST-EEGFormer, including its model architecture, pre-training task, implementation details, and results.

## E.1 MODEL ARCHITECTURE

The proposed ST-EEGFomer is based on the ViT architecture (Dosovitskiy et al., 2021), pre-trained using MAE (He et al., 2022). The pre-training task involves reconstructing the original EEG inputs from masked tokens, as illustrated in Figure 1 (b). During pre-training, raw EEG signals are divided into spatial and temporal segments, which are tokenized through a linear projection layer. Each token is augmented with temporal and spatial positional embeddings to preserve structural information. A random masking strategy is applied, with 75% of tokens hidden, and the encoder processes only the remaining visible tokens. To enable reconstruction, the masked tokens—together with their positional embeddings—are concatenated with the encoder output to form the full token sequence, which is then passed through a lightweight decoder to reconstruct the original EEG signal. After pre-training, only the encoder is retained and used as the ST-EEGFormer backbone for fine-tuning on downstream tasks. Three different ST-EEGFormers (small, base, and large models) were pre-trained in this study. The base and large models have the same architecture as the base, large models proposed in the ViT implementation, while the corresponding decoders have the same architecture as in the MAE implementation. Details about the benchmarked model can be found in table E.1.

## E.2    PRE-TRAINING DATASETS

The MAE reconstruction task is conducted on 11 public datasets. These datasets include:

*1) EEG-MI-BCI (Cho et al., 2017)*: This dataset contains 52 subjects performing 2-class imagined left and right-hand movements, recorded with 64 EEG channels using a Biosemi ActiveTwo system. It includes approximately 5,000 trials of 3-second motor imagery (MI) data per class.

*2) HGD (Schirrmeister et al., 2017)*: The High Gamma Dataset comprises 20 subjects performing 4-second trials of executed movements with four classes (left hand, right hand, both feet, and rest). The data were recorded with 128 high-density EEG caps (WaveGuard Original, ANT, Enschede, NL) and sampled at 5 kHz using a NeurOne amplifier (Mega Electronics Ltd, Kuopio, FI). It includes roughly 3,000 trials per class.

*3) BCI-Comp-IV2a (Tangermann et al., 2012)*: This dataset includes nine subjects performing four-second trials of four classes (imagined left hand, right hand, feet, and tongue movements), recorded with 22-electrode EEG caps. It contains a training set and a test set from two separate sessions, each with roughly 600 trials per class.

*4) BCI-Comp-IV2b (Tangermann et al., 2012)*: This dataset consists of nine subjects performing 2-class imagined left and right-hand movements, recorded with three EEG channels. It contains a training set of approximately 1,800 trials per class and a separate test set of approximately 1400 trials per class.

*5) Large-MI-Classic (Kaya et al., 2018)*: This dataset comprises 13 subjects performing 1-second trials of six classes (imagined left hand, right hand, left foot, right foot, tongue, and rest). The data were recorded with 19-channel EEG caps plus 2 ground lead channels (Electro-Cap International, USA) and were mostly sampled at 200 Hz, with some recordings sampled at 1000 Hz using the EEG-1200 system. In total, it includes approximately 50,000 trials (different classes have an unequal number of trials).

*6) Large-MI-5F (Kaya et al., 2018)*: From the same study as 5) but different experiments, this dataset comprises 13 subjects performing 1-second trials of five classes of finger movements (imagined thumb, index, middle, ring, pinkie). In total, it includes around 18000 trials.

*7) P300 (Won et al., 2022)*: This dataset consists of 55 participants performing a P300 speller experiment and 50 participants viewing a rapid serial visual representation (RSVP). In total, it includes 99000 training P300 trials and 277200 test trials.

*8) SSVEP (Liu et al., 2020)*: This dataset consists of 70 participants performing cue-guided SSVEP target-selecting experiments, comprising 40 flickering stimuli ranging between 8 Hz to 15.8 Hz with an interval of 0.2 Hz. For each target, it contains 20 trials of 5-s stimulation data.

*9) Online MI BCI Classification (Stieger et al., 2021)*: This dataset contains 600 hours of 62-channel EEG recordings, sampled at 1000 $Hz$, collected during online and continuous BCI control from 62 healthy adults, spanning multiple sessions across different days. The BCI paradigm involves imagining left, right, and both hand movements (opening and closing), as well as a resting state condition, to control a virtual cursor. The provided data consists of epoched trials of varying lengths, structured with a 2-second inter-trial interval, followed by a 2-second target presentation. The task imagination phase varies in length, with a maximum duration of up to 6.04 seconds, followed by a 1-second post-trial interval.

*10) KUL Auditory Decoding Dataset (Bollens et al., 2023)*: This dataset consists of 64-channel EEG recordings from 85 young participants, each exposed to 90–150 minutes of continuous natural speech. Data were acquired with a BioSemi 64-channel system at a sampling rate of 1024 Hz,

providing a large-scale resource for auditory attention and speech decoding research.

*11) **SEED-V Emotion EEG Dataset** (Liu et al., 2022)*: This dataset provides multimodal EEG and eye-tracking data for emotion recognition. It includes recordings from 16 subjects who participated in three sessions, each watching 15 movie clips spanning five emotional categories: happy, sad, fear, disgust, and neutral (3 clips per emotion per session, totaling 45 trials per subject). EEG was acquired using a 62-channel NeuroScan system (10–20 layout) at 1000 Hz (typically downsampled to 200 Hz), while eye movement was captured with SMI tracking glasses. Each trial includes stimulus presentation followed by a rest/self-assessment period. SEED-V stands out as a rich resource for emotion decoding from EEG–eye multimodal data, offering both raw signals and precomputed differential entropy features across standard frequency bands.

The dataset selection followed two main criteria: (i) size and quality, as highlighted in a recent review (Gwon et al., 2023), and (ii) benchmark relevance. Specifically, the MI datasets were chosen for their robustness and widespread use, with Datasets 3 and 4 serving as classic benchmarks in motor imagery research. The P300 dataset was included as it represents one of the largest publicly available collections for event-related potential decoding. Similarly, the SSVEP dataset was selected due to its established role as a standard benchmark in visual BCI studies. Finally, the auditory decoding dataset was incorporated given its scale and its unique position as the largest EEG resource in the auditory domain. In addition, we integrated an **in-house EEG dataset** covering multiple paradigms (e.g., MI and SSVEP) with diverse channel configurations. This enriched the model's exposure to a wide range of electrode montages, ultimately enabling it to learn from **142 unique EEG channels** and improving its adaptability to future datasets.

### E.3   DATA PREPROCESSING

All datasets underwent minimal preprocessing to ensure comparability while preserving raw signal characteristics. Specifically, power-line noise was removed when present using the `mne.filter.notch_filter()` function (Python 3.8.19, MNE 1.6.1). Next, a band-pass filter between 0.1–64 Hz was applied to all channels via `mne.filter.filter_data()` with a windowed FIR design (`fir_design='firwin'`). The signals were then downsampled to 128 Hz using `mne.filter.resample()` and finally standardized to zero mean and unit variance per channel.

### E.4   DATA SEGMENTATIONS FOR PRE-TRAINING

The benchmark datasets differ in whether they provide continuous EEG recordings or only task-related epochs. To unify pre-training data construction, dataset-specific sliding-window strategies were applied:

**1)** For continuous datasets (e.g., EEG-MI-BCI (Cho et al., 2017), HGD (Schirrmeister et al., 2017), BCI-Comp-IV2a (Tangermann et al., 2012), BCI-Comp-IV2b (Tangermann et al., 2012), P300 (Won et al., 2022), SEED-V (Liu et al., 2022), and KUL Auditory (Bollens et al., 2023)), 6-s windows with 0.5-s hops were used.

**2)** For large MI datasets (Large-MI-Classic (Kaya et al., 2018), Large-MI-5F (Kaya et al., 2018), Online MI BCI (Stieger et al., 2021)), a 6-s window with a 2.5-s hop was applied to reduce redundancy.

**3)** For the SSVEP dataset (Liu et al., 2020), which consists only of 5-s stimulation epochs, 2-s windows with 0.125-s hops were extracted.

**4)** We also included EEG data recorded in our own lab to enrich electrode coverage. By combining heterogeneous datasets, the encoder was exposed to **142 unique EEG channels**, improving robustness for transfer to unseen datasets.

A validation split was retained for each dataset: for BCI-Comp-IV2a, the official test set was used, while for others, 20% of the data was held out. Overall, this yielded more than **8 million overlapping EEG segments**.

### E.5 MAE PRE-TRAINING METHODOLOGY

The pre-training procedure follows the original MAE framework. EEG data are first divided into spatial–temporal segments, which are linearly projected into embeddings with added spatial (SPE) and temporal positional encoding (TPE). For the spatial positional embeddings, a learned embedding per channel was used, similar to the learned positional embedding in (Gehring et al., 2017), while for the temporal positional embeddings, a sine-cosine positional embedding approach was used, as shown in Eq E.1. A fixed ratio of 75% of tokens is randomly masked, and the remaining tokens are passed through a ViT-based encoder. Mask tokens with their positional embeddings are then concatenated with the encoder output and processed by a lightweight decoder. The decoder reconstructs the original EEG, and the objective is the mean squared error (MSE) between reconstructed and original signals over the masked segments.

$$p_t^{(i)} = f(t)^{(i)} := \begin{cases} \sin(\omega_k \cdot t), & \text{if } i = 2k \\ \cos(\omega_k \cdot t), & \text{if } i = 2k+1 \end{cases} \quad , \text{where} \quad \omega_k = \frac{1}{10000^{\frac{2k}{d}}} \tag{E.1}$$

### E.6 MAE PRE-TRAINING SETTINGS

The model is initialized with `xavier_uniform` (Glorot & Bengio, 2010). Optimization follows AdamW (Loshchilov & Hutter, 2019) with a base learning rate of 3e-4, weight decay of 0.05, batch size of 256, cosine learning rate decay (Loshchilov & Hutter, 2017), and a 10-epoch warmup (Goyal et al., 2018). The learning rate scales linearly with batch size according to Eq. E.2.

$$lr = base\_lr \times \frac{batch\ size}{256} \tag{E.2}$$

### E.7 PRACTICAL CONSIDERATIONS FOR MAE PRE-TRAINING

We conducted pre-training on a high-performance computing (HPC) cluster equipped with NVIDIA A100 GPUs (80 GB). Before the production run, benchmark experiments were performed to evaluate scaling efficiency and estimate wall-time under different job sizes. Each benchmark ran for one epoch across the full dataset, and the average epoch wall-time was extrapolated to 400 epochs, consistent with the original MAE setting.

Figure E.1 summarizes these benchmarks. The blue line shows relative efficiency compared to a single-GPU baseline (green dashed line), while the red dashed line indicates estimated wall-time in days. Panels (a–d) correspond to different model–hardware configurations: (a) small model on 40 GB A100, (b) small model on 80 GB A100, (c) base model on 80 GB A100, and (d) large model on 80 GB A100. Efficiency was defined as:

$$\text{Efficiency} = \frac{B_{\text{baseline}} \cdot T_{\text{baseline}}}{B_{\text{current}} \cdot T_{\text{current}}}$$

where $B$ is the number of CPU cores and $T$ the wall-time.

We ultimately selected the $16\times$A100-80GB configuration, which offered the best trade-off between efficiency (Efficiency $> 50\%$) and throughput. The benchmarks highlight that larger GPU memory enables larger batch sizes, reducing per-epoch training time. However, they also illustrate the extreme resource demands of large-scale EEG pre-training: even with optimized scaling, our production run consumed **32,614 GPU hours**. This underscores both the cost of developing EEG foundation models and the importance of transparent reporting of their computational footprint.

### E.8 MAE PRE-TRAINING RESULTS

The pre-training learning curves for the small, base, and large models are shown in Figure E.2. As expected, the large model achieves the lowest reconstruction loss, followed by the base model, and the small model yields the highest loss. The small and base models exhibit smooth and stable convergence, whereas the large model shows some instability during the early epochs before eventually converging. Overall, all model variants successfully converge, which directly contrasts with

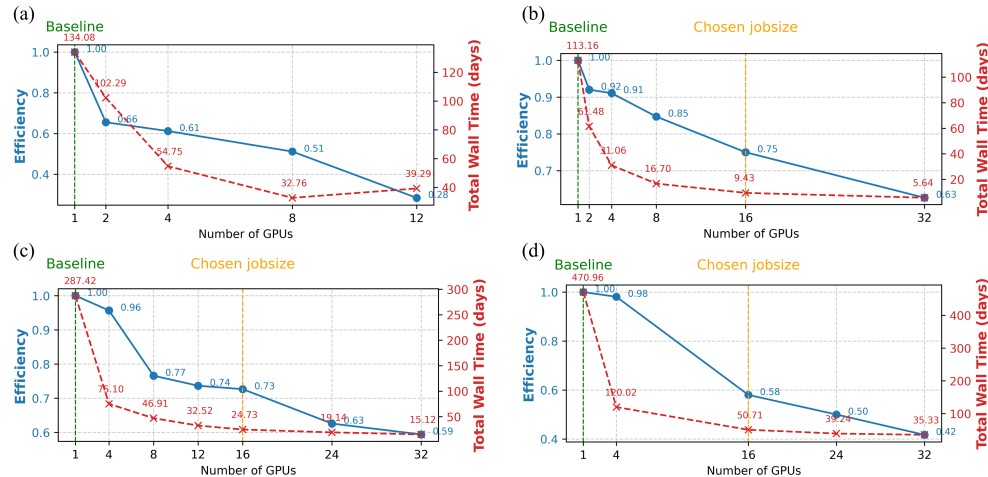

Figure E.1: GPU scaling benchmark for MAE pre-training. The blue line shows efficiency relative to the single-GPU baseline (green dashed line), while the red dashed line indicates estimated total wall-time in days for 400 epochs. Panels: (a) small model on A100-40GB, (b) small model on A100-80GB, (c) base model on A100-80GB, and (d) large model on A100-80GB. The chosen production configuration is indicated by the orange dashed line.

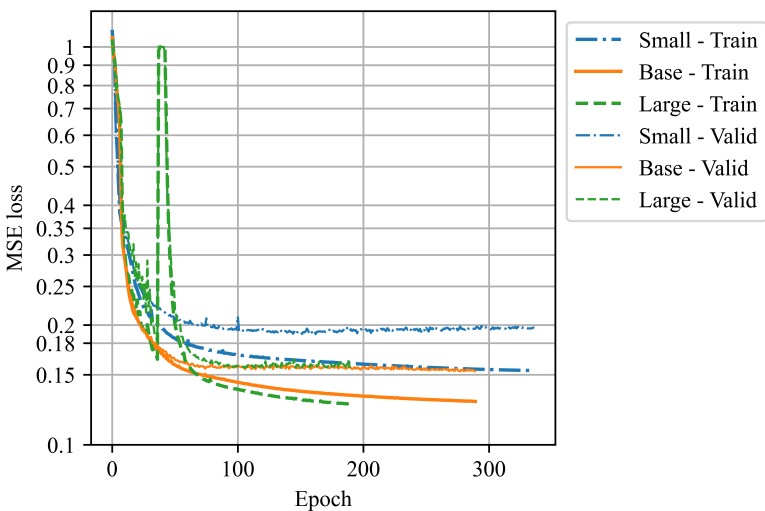

Figure E.2: Learning curves of the small, base, and large ST-EEG MAE models during the MAE pertaining phase.

the claim made in LaBraM (Jiang et al., 2024) that masked autoencoding is difficult to train on raw EEG signals.

Some examples of the reconstructed signals compared to the original ones are presented in figures E.3 and E.4. It is noteworthy that the model was able to effectively reconstruct the low-frequency trends, though it encountered difficulties in accurately reconstructing high-frequency spikes. This could be attributed to the lower signal-to-noise ratio (SNR) of high-frequency EEG components, making them more challenging to learn.

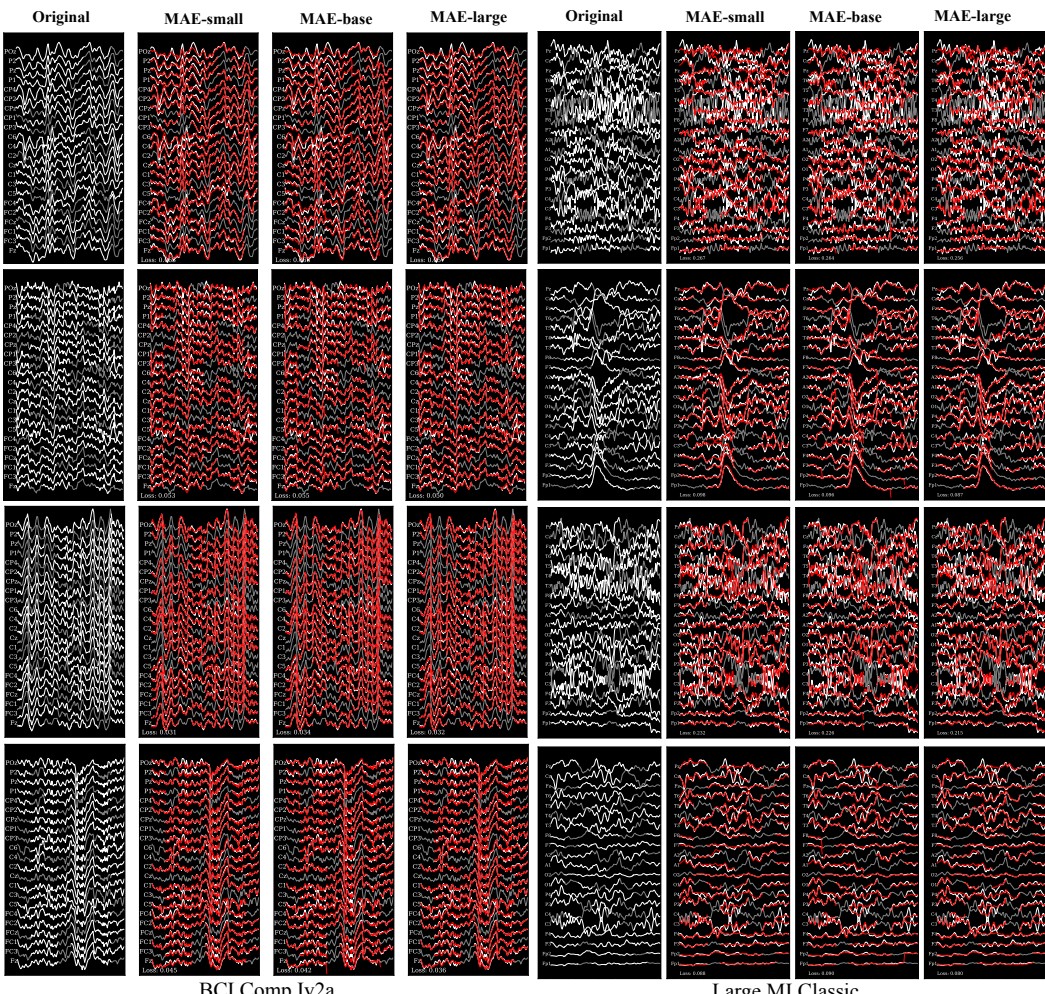

Figure E.3: Random samples from the BCI-IV-2A (the first 4 columns), Large-MI-Classic (the last 4 columns). For each example (4 rows), the following are shown, from left to right: the original signals with masked segments highlighted in white and unmasked segments in grey; the reconstructed signals in red produced by the MAE-Small model, overlaid on the original signals in white; the reconstructed signals in red produced by the MAE-Base model, overlaid on the original signals in white; and the reconstructed signals in red produced by the MAE-Large model, overlaid on the original signals in white. The corresponding mean squared error (MSE) loss is displayed at the bottom of each figure.

### E.9 ADDITIONAL MODEL VALIDATION ON PRE-TRAINING DATASETS

After MAE pre-training and before the large-scale benchmarks described in Appendix D, we conducted intermediate evaluations to assess the effectiveness of the pre-trained ST-EEGFormer. Population decoding was performed on four motor imagery/execution datasets (EEG-MI-BCI (Cho et al., 2017), HGD (Schirrmeister et al., 2017), Large-MI-Classic (Kaya et al., 2018), and Large-MI-5F (Kaya et al., 2018)), one P300 dataset (Won et al., 2022), ONE SSVEP dataset (Liu et al., 2020), and a single-channel seizure classification dataset (Andrzejak et al., 2002). The seizure dataset is not included in the pre-training corpora. ST-EEGFormer was compared against representative baselines, including EEGNet, EEG Conformer, BIOT, and LaBraM. Additionally, we implemented a simple linear model (Table E.2), consisting of a spatial filter, a feature extractor, and a fully connected layer without nonlinear activations. This model serves as a minimal yet informative baseline, with the extracted feature set summarized in Table E.3.

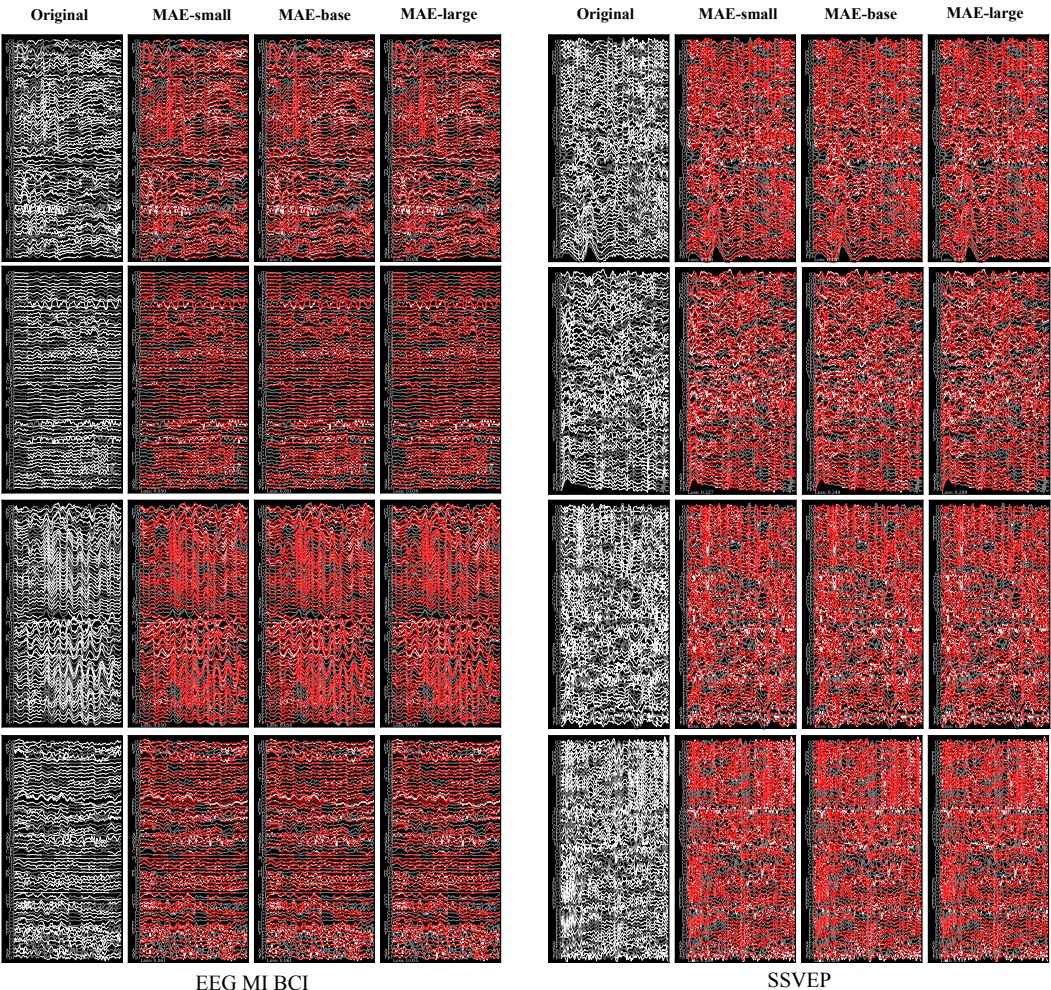

Figure E.4: Random samples from the EEG-MI-BCI (the first 4 columns), and SSVEP datasets (the last 4 columns). For each example (4 rows), the following are shown, from left to right: the original signals with masked segments highlighted in white and unmasked segments in grey; the reconstructed signals in red produced by the MAE-Small model, overlaid on the original signals in white; the reconstructed signals in red produced by the MAE-Base model, overlaid on the original signals in white; and the reconstructed signals in red produced by the MAE-Large model, overlaid on the original signals in white. The corresponding mean squared error (MSE) loss is displayed at the bottom of each figure.

### E.9.1 EXPERIMENT DETAILS

For all MI datasets, we employed a 5-fold cross-validation strategy, using the `StratifiedKFold` function from `sklearn.model_selection` to ensure class balance within each fold. This approach was applied individually to each recording. During the 5-fold cross-validation, 4 folds are used as the current training set, and the remaining set is the test set for this fold:

*1) Training and Validation Split*: For each fold, 20% of the training data was set aside as a validation set, used for model selection.

*2) Model Selection*: The model achieving the highest classification accuracy on this validation set was chosen as the best model during different training epochs.

*3) Testing*: The selected model was then evaluated on the test set of the current fold.

Table E.2: Linear model architecture. Input EEG data consist of $N_{ch}$ channels and $L$ time samples. The output corresponds to $N_{class}$, representing the number of different classes to classify.

| Layer | Name | Type | Layer specific settings | Output shape |
|---|---|---|---|---|
| 0 | Input | NA | NA | $(N_{ch} \times L)$ |
| 1 | Spatial filter | Conv1d | kernel size:$(N_{ch}, 1)$ number of kernels:8 | $(8 \times L)$ |
| 2 | Drop out | Dropout | $p=0.2$ | $(8 \times L)$ |
| 3 | Feature extractor | NA | see table E.3 | $(8 \times 12)$ |
| 4 | Flatten | NA | NA | $(1 \times 96)$ |
| 5 | Classification head | Linear | weights and bias shape: $(96, N_{class})$ | $(1, N_{class})$ |

Table E.3: Features calculated in the feature extractor layer.

| Feature | Definition | Remark |
|---|---|---|
| Mean | $\bar{x} = \frac{1}{n} \sum_{i=1}^{n} x_i$ | $n$: total number of samples, $x_i$: the $i$-th sample. |
| Variance | $\sigma^2 = \frac{1}{n} \sum_{i=1}^{n} (x_i - \bar{x})^2$ | $\bar{x}$: the mean value |
| Power | $P = \frac{1}{n} \sum_{i=1}^{n} x_i^2$ | NA |
| Skewness | $\tilde{\mu}_3 = \frac{\sum_{i=1}^{n}(x_i - \bar{x})^3}{(n-1)\cdot\sigma^3}$ | $\sigma$: the standard deviation |
| Kurtosis | $K = \frac{1}{n} \sum_{i=1}^{n} \left(\frac{x_i - \bar{x}}{\sigma}\right)^4$ | NA |
| Entropy | $H = -\sum_{i=1}^{n} p(x_i) \log\left(p(x_i) + \epsilon\right)$ | $n = 256$: the number of intensity bins $p(x_i)$: the probability of the $i$-th intensity bin $\epsilon = 10^{-8}$: a small constant for stability |
| Maximum | $\max(\mathbf{x}) = \max_{i=1}^{n} x_i$ | NA |
| Minimum | $\min(\mathbf{x}) = \min_{i=1}^{n} x_i$ | NA |
| The first quartile | $Q_1 = \text{Quantile}(\mathbf{x}, 0.25)$ | NA |
| The secoond quartile | $Q_2 = \text{Quantile}(\mathbf{x}, 0.50)$ | NA |
| The third quartile | $Q_3 = \text{Quantile}(\mathbf{x}, 0.75)$ | NA |
| Zero cross rate | $\text{ZCR} = \frac{1}{2n} \sum_{i=2}^{n} |\text{sgn}(x_i) - \text{sgn}(x_{i-1})|,$ $\text{sgn}(x) = \begin{cases} 1, & \text{if } x > 0 \\ 0, & \text{if } x = 0 \\ -1, & \text{if } x < 0 \end{cases}$ | NA |

Additionally, for the HGD (Schirrmeister et al., 2017) dataset, a separate hidden test set was available. This hidden test set was used as additional test set, and the models selected from the cross-validation step were further evaluated on this set to assess their performance comprehensively.

For the SSVEP (Liu et al., 2020) dataset, we followed the approach outlined in the SSVEP DNN paper (Guney et al., 2022), using a sliding window method to generate training samples of 1-second and 2-second lengths, with a hop size of 0.1 seconds. The test set also contains small segments of EEG data generated using the same sliding window on the hidden test trial data. We employed the same leave-one-session-out validation strategy for the experiment, as in (Guney et al., 2022), and the model selection process was consistent with that used in the MI experiments.

For the P300 (Won et al., 2022) dataset, we utilized the provided training and test sets. As in other P300 decoding experiments, we evaluated the model's performance under varying numbers of trial averaging. These trials were averaged based on the flashing of rows and columns during the experiment.

Remark that the training data in each fold were kept the same when training different models.

### E.9.2 MODEL IMPLEMENTATION AND TRAINING DETAILS

This intermediate benchmark was designed as a validation step; thus, only a subset of models was evaluated. The main objective was to examine the token fusion strategy in ST-EEGFormer. Following the ViT paradigm, we compared two variants: using the class token (Cls) or the average of all tokens (Avg) as input to the final classification layer. Training followed the general settings in Ap-

pendix F.1.2, except that we used a larger batch size (128) and a higher base learning rate ($3 \times 10^{-4}$) to accelerate convergence.

### E.9.3 INTERMEDIATE BENCHMARK RESULTS

**Movement-related datasets** The benchmark results on all movement-related datasets are summarized in Table E.4. Overall, the pre-trained ST-EEGFormer consistently achieved the highest classification accuracies across multiple BCI datasets. Among the model variants, the large version outperformed both the base and small models on most datasets. For all movement-related benchmarks, the fine-tuned base models yielded higher accuracies than their linearly probed counterparts, although the latter performed similarly to supervised linear models trained from scratch. Furthermore, the mean accuracies obtained with subject-specific linear models reported in Figure 3 of Gwon et al. (2023) (approximately 60%) were comparable to those achieved by our population-trained linear models. These findings highlight the effectiveness of our proposed approach: self-supervised pre-training on large-scale EEG recordings enables the foundation model to learn robust neural representations, yielding performance competitive with, or superior to, traditional linear classifiers.

**P300** The benchmark results on the P300 dataset are shown in Figure E.5. Model performance was evaluated under different numbers of row–column trial averages, a standard strategy in P300 decoding. Multiple runs were conducted with increasing repetitions, and the results are reported as performance curves. Training and test sets followed the official dataset split. Trial averaging was performed by aggregating EEG responses across repeated flashes of the same rows or columns, thereby enhancing the signal-to-noise ratio. As expected, performance consistently improved with the number of repetitions. Notably, BIOT underperformed even the simple linear baseline, while the best results were achieved by ST-EEGFormer-l, followed by EEGNet. EEG Conformer and LaBraM showed comparable performance, ranking below the top models.

**SSVEP** For the SSVEP dataset, we benchmarked asynchronous decoding performance using the small segmented windows generated by the sliding-window method. The results are shown in Table E.5. The best-performing model was ST-EEGFormer-l, followed by LaBraM. As expected, performance improved when longer window lengths were used, reflecting the benefit of increased temporal context. In contrast, the linear model performed poorly, achieving less than 10% accuracy in the 40-target classification task.

**Seizure classification** Additionally, we tested our approach on a single-channel seizure classification task using the famous Bonn dataset (Andrzejak et al., 2002). This dataset consists of human expert-selected single-channel EEG data from five healthy volunteers and five individuals with epilepsy. The data are divided into two classes for healthy volunteers, including scalp EEG segments recorded while the volunteers were relaxed and awake with eyes closed and open, respectively (Dataset A and B, referred to as "Eyes Closed" and "Eyes Open" in figure E.6 (b). Three classes of data are from epileptic patients, consisting of intracranial EEG (iEEG) segments recorded during pre-surgical evaluation. Specifically, one class contains interictal iEEG segments from the epileptogenic zone in the opposite hemisphere (dataset C, referred to as "NSeizure-Opposite" in figure E.6 (b), while another class includes interictal iEEG segments from the epileptogenic zone itself (dataset D, referred to as "NSeizure-Epileptogenic" in figure E.6 (b). The final class consists of iEEG segments recorded from the epileptogenic zone during seizure activity (dataset E, referred to as "Seizure" in figure E.6 (b). Each subset contains 100 single-channel EEG segments, each 23.6 seconds in duration (4096 samples). The data were sampled at 173.61 $Hz$, and any artifacts caused by muscle activity or eye movement were manually removed by the database owners after visual inspection.

The hypothesis is that if the model learns robust EEG representations from normal EEG-BCI recordings during the pre-training step, it should be able to classify abnormal EEG data as well. Therefore, in the first experiment, we varied the amount of learning examples from only 5% to 60% and compared the classification accuracies among different models. In this experiment, we tested the performance of 1) directly applying linear probing on the pre-trained model; 2) directly fine-tuning the pre-trained model; 3) further calibrating the model by performing the MAE task, followed by linear probing on the seizure dataset, and 4) further calibrating the model by performing the MAE task and then fine-tuning on the seizure dataset. This was done to determine which approach yields

Table E.4: Movement datasets benchmark results (reversed). "-cv" represents the average k-fold cross-validation accuracy, while "-test" represents the average accuracy on the hidden test set. The highest and second-highest accuracies are in bold, with the highest one marked in bold and surrounded by a box. For ST-EEGFormer, the default fine-tuning strategy is end-to-end fine-tuning with the average token, "lp" denotes a linear probed model, and "cls" refers to an end-to-end fine-tuned model using the class token.

| Model | EEG-MI-BCI-cv | HGD-cv | HGD-test | Large-MI-Classic-cv | Large-MI-5F-cv |
|---|---|---|---|---|---|
| Linear | $0.683 \pm 0.007$ | $0.631 \pm 0.017$ | $0.593 \pm 0.021$ | $0.442 \pm 0.009$ | $0.320 \pm 0.015$ |
| EEGNet | $0.781 \pm 0.011$ | $0.899 \pm 0.010$ | $0.859 \pm 0.003$ | $0.644 \pm 0.004$ | $0.479 \pm 0.006$ |
| Conformer | $0.821 \pm 0.012$ | $\mathbf{0.914 \pm 0.003}$ | $0.878 \pm 0.010$ | $0.722 \pm 0.004$ | $\mathbf{0.529 \pm 0.004}$ |
| BIOT | $0.718 \pm 0.020$ | $0.651 \pm 0.005$ | $0.612 \pm 0.015$ | $0.455 \pm 0.012$ | $0.287 \pm 0.008$ |
| LaBraM | $0.736 \pm 0.010$ | $0.892 \pm 0.007$ | $\mathbf{0.902 \pm 0.040}$ | $\mathbf{0.763 \pm 0.005}$ | $0.464 \pm 0.023$ |
| ST-EEGFormer-s | $0.905 \pm 0.020$ | $0.888 \pm 0.010$ | $0.858 \pm 0.011$ | $\mathbf{0.763 \pm 0.008}$ | $0.500 \pm 0.008$ |
| ST-EEGFormer-b | $\boxed{\mathbf{0.937 \pm 0.005}}$ | $0.874 \pm 0.011$ | $0.838 \pm 0.006$ | $0.754 \pm 0.006$ | $0.483 \pm 0.010$ |
| ST-EEGFormer-b-lp | $0.693 \pm 0.011$ | $0.630 \pm 0.014$ | $0.579 \pm 0.014$ | $0.439 \pm 0.004$ | $0.294 \pm 0.008$ |
| ST-EEGFormer-b-cls | $\mathbf{0.936 \pm 0.010}$ | $0.873 \pm 0.009$ | $0.817 \pm 0.007$ | $0.731 \pm 0.004$ | $0.462 \pm 0.003$ |
| ST-EEGFormer-l | $0.931 \pm 0.005$ | $\boxed{\mathbf{0.954 \pm 0.004}}$ | $\boxed{\mathbf{0.935 \pm 0.002}}$ | $\boxed{\mathbf{0.831 \pm 0.003}}$ | $\boxed{\mathbf{0.627 \pm 0.013}}$ |

the best performance. The results are presented in figure E.6 (a). The confusion matrix of the base model is shown in figure E.6 (b). Moreover, we also checked the effects of the mask ratio in the MAE pre-training step by varying the mask ratio and comparing the finetuned model and linear probing model performance under different mask ratios with only 5% training data. The results are presented in figure E.6 (c). Figure E.6 (a) demonstrates that all pre-trained ST-EEGFormer models outperformed both EEGNet and Conformer, particularly when training data were limited. Moreover, performance could be further improved by calibration, as the highest accuracy was obtained by the ST-EEGFormer base-cali model. In contrast to results from previous datasets, where linear-probed models significantly underperformed finetuned models, the linear-probed models in this study achieved satisfactory results, especially after calibration, surpassing other models. This success can be attributed not only to the robust EEG representations learned during the MAE pre-training stage that help classify abnormal EEG data but also to the relatively straightforward classification task, which exhibits distinguishable characteristics that are easily visually inspected, making linear probing more effective. These findings provide a solid foundation for the future application of ST-EEGFormer in seizure classification, as the model could potentially learn even better representations from large open public seizure datasets not included in this study.

**Summary**   The above benchmark experiments yield the following insights:
*1) Effectiveness of SSL pre-training:* Both the benchmarks on pre-training datasets and the calibration experiment on the seizure dataset demonstrate that self-supervised pre-training improves downstream task performance. This provides strong evidence for the utility of large EEG foundation models. However, the performance gap between calibrated and non-calibrated models suggests that certain useful representations are not fully captured during pre-training. This may be attributed to the limited availability of seizure-related data in pre-training or to representation shifts between the pre-training and downstream tasks.
*2) Weak linear probing performance:* Across all movement-related datasets, linear-probed ST-EEGFormer performed poorly, comparable to the simple linear baseline. This indicates that the representations learned during pre-training do not transfer effectively to downstream classification tasks, even when the same data were part of the pre-training corpus.
*3) Inferior class-token fusion:* In all experiments, using the class token for classification yielded worse results than averaging over all tokens. This suggests that the class token did not play a meaningful role during pre-training. Based on this finding, we adopt the average-token fusion strategy exclusively in all downstream benchmark experiments.

## F   MODEL IMPLEMENTATION DETAILS

In this section, we present all benchmarked models and implementation details used across the downstream tasks.

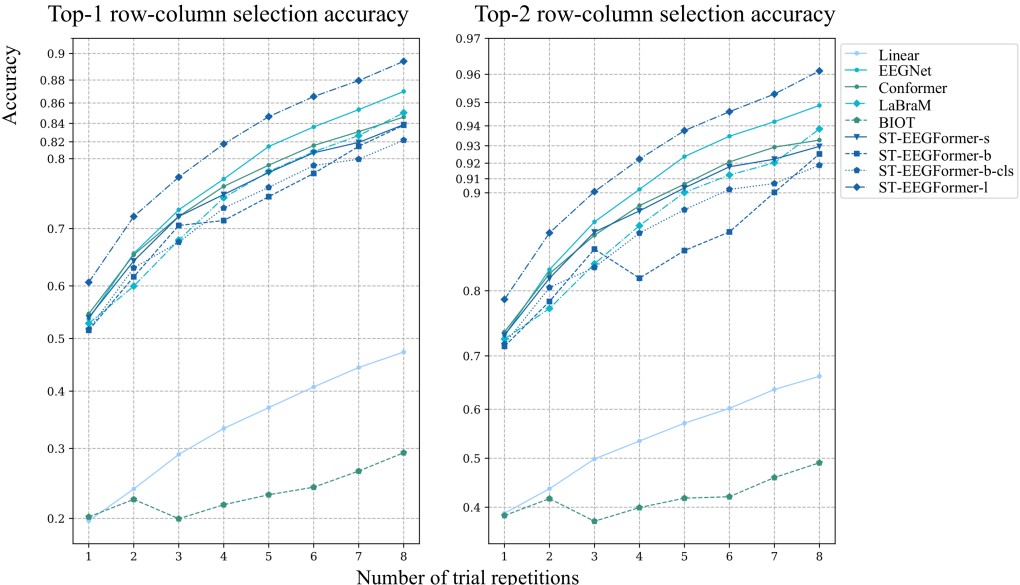

Figure E.5: **P300 benchmark results**. Row-column selection accuracy of the P300 BCI. The original interface consists of 6 rows and 6 columns. A prediction is made for the row in which the attended character is present after all rows have flashed for the specified number of repetition rounds and, similarly, for the columns. EEG data of the same row or column, but from different repetition rounds, are averaged to create an averaged epoch for classification. The chance level for selection accuracy is therefore 1/6 (16.7%).

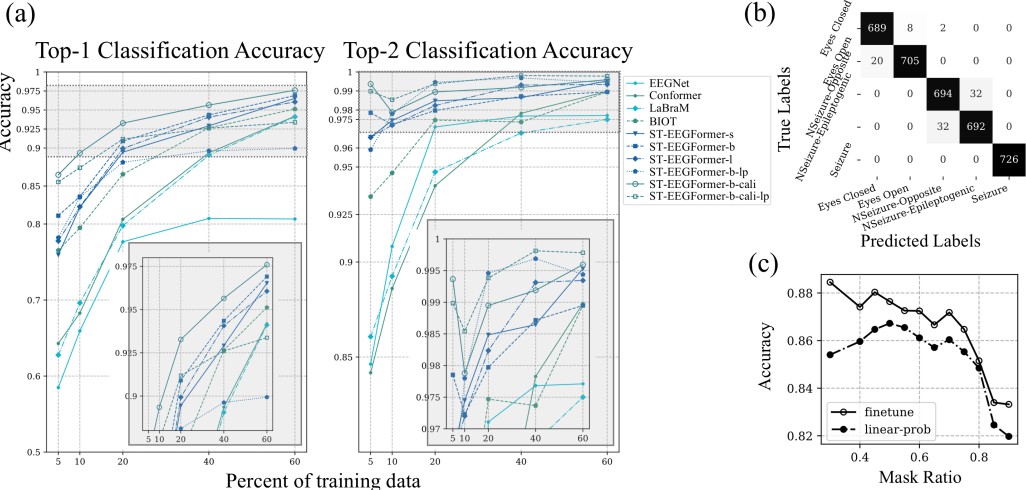

Figure E.6: (a) Top-1 and top-2 classification accuracy on the seizure dataset with varying training data sizes, comparing EEGNet, EEG Conformer, LaBraM, BIOT, fine-tuned ST-EEGFormer models (small, base, and large), and the linearly-probed base model (ST-EEGFormer-b-lp). Additionally, the base model was further calibrated on the seizure dataset by performing the MAE SSL task using a mask ratio of 0.75, then fine-tuned and linearly-probed, referred to as ST-EEGFormer-b-cali and ST-EEGFormer-b-cali-lp, respectively. (b) Confusion matrix of the ST-EEGFormer base model trained with 60% of the data. (c) Accuracy of fine-tuned and linearly-probed ST-EEGFormer base-cali models with varying mask ratios during the calibration stage. The chance level is 1/5 (20.0%).

Table E.5: SSVEP dataset benchmark results. The average accuracies from the leave-one-session-out experiment are reported. The highest and second-highest accuracies are in bold, with the highest one marked in bold and surrounded by a box. For ST-EEGFormer, the default fine-tuning strategy is end-to-end fine-tuning using the average token. Models denoted by "-cls" indicate end-to-end fine-tuned models utilizing the class token. The chance level is 1/40 (2.5%).

| Model | Window = 1s | | Window = 2s | |
|---|---|---|---|---|
| | Top1-Acc | Top2-Acc | Top1-Acc | Top2-Acc |
| Linear | 0.047 | 0.088 | 0.047 | 0.087 |
| EEGNet | 0.433 | 0.625 | 0.646 | 0.785 |
| Conformer | 0.328 | 0.517 | 0.419 | 0.618 |
| BIOT | 0.316 | 0.449 | 0.492 | 0.627 |
| LaBraM | **0.518** | **0.669** | **0.700** | **0.818** |
| SSVEP-DNN | 0.385 | 0.570 | 0.442 | 0.606 |
| ST-EEGFormer-s | 0.387 | 0.551 | 0.441 | 0.604 |
| ST-EEGFormer-b | 0.218 | 0.344 | 0.217 | 0.342 |
| ST-EEGFormer-l | **0.590** | **0.748** | **0.807** | **0.893** |
| ST-EEGFormer-b-cls | 0.251 | 0.385 | 0.267 | 0.404 |

## F.1 FOUNDATION MODELS

In this study, the following foundation models are benchmarked across all downstream tasks.

### F.1.1 MODEL INTRODUCTION

**BENDR** (BERT-inspired Neural Data Representations) (Kostas et al., 2021) is one of the earliest Transformer-based foundation models for EEG signals. Introduced by Kostas et al. (2021), BENDR combines a convolutional encoder with a Transformer decoder and adapts the self-supervised training strategy from wav2vec 2.0 (Baevski et al., 2020) to multi-channel EEG data. In pre-training, contiguous spans of the input EEG are masked and the model learns to reconstruct their latent representations using a contrastive objective. This approach enables BENDR to learn general-purpose EEG features from large unlabeled corpora, which can then be fine-tuned on specific tasks. In our implementation, we leverage the official BENDR code and pre-trained weights provided by its authors as a baseline foundation model (`https://github.com/SPOClab-ca/BENDR`).

**BIOT** (Biosignal Transformer) (Yang et al., 2023) is a Transformer-based encoder designed for cross-dataset biosignal learning, demonstrated on EEG data. The BIOT architecture tokenizes each EEG channel into fixed-length segments (local signal "patches") and then concatenates these segments from all channels into a long "sentence" representation. Channel-specific embeddings and relative positional encodings are added to each token to preserve spatial and temporal context, allowing BIOT to handle mismatched electrode montages, variable sequence lengths, and even missing channels across different datasets. Pre-trained on multiple EEG datasets in the wild, BIOT has shown superior performance over task-specific models by learning from diverse data sources. The BIOT model used in this study is the version pre-trained on all six EEG datasets, obtained from the official repository (`https://github.com/ycq091044/BIOT`).

**LaBraM** (Large Brain Model) (Jiang et al., 2024) is a large-scale EEG foundation model that aims to learn generic representations from tremendous amounts of EEG data. To enable cross-dataset learning, LaBraM segments raw EEG signals into channel-wise patches and employs a vector-quantized autoencoder to convert each patch into a discrete neural code (acting as a "token"). A Transformer model is then pre-trained to predict masked patch codes from their surrounding context, similar in spirit to masked language modeling in NLP. This two-step approach (neural tokenizer + masked code prediction) allows the model to capture rich semantic information from the EEG. The published LaBraM models were pre-trained on approximately 2,500 hours of EEG recordings drawn from about 20 different datasets encompassing various BCI tasks, achieving state-of-the-art results on diverse downstream evaluations. In our study, we utilize the base version of LaBraM ("labram-base" checkpoint) released by the authors (`https://github.com/935963004/LaBraM/tree/main`).

**EEGPT**    (Wang et al., 2024) is a recently proposed Transformer-based foundation model for EEG that strives to produce universal and reliable EEG feature representations. The model introduces a dual masked self-supervised learning strategy: it performs simultaneous masking in both the spatial (electrode) dimension and the temporal dimension, and learns to predict the high-level representations of these masked portions. By focusing the learning objective on higher-level latent representations (with higher signal-to-noise ratio) rather than raw signal reconstruction, EEGPT's pre-training task emphasizes more robust and salient EEG features. Additionally, the EEGPT architecture uses a hierarchical design that processes spatial correlations and temporal dynamics in separate stages, which improves training efficiency and adaptability to various EEG paradigms. The model is pretrained on a large-scale compilation of EEG data from multiple tasks, and it achieves top-tier performance on a range of downstream benchmarks (often evaluated via linear probing on the learned features). We integrate EEGPT into our pipeline using the official implementation and pre-trained weights provided by its authors (`https://github.com/BINE022/EEGPT`).

**CBraMod**    (Criss-Cross Brain Model) (Wang et al., 2025) is an EEG foundation model that employs a specialized criss-cross Transformer architecture to capture EEG's distinct spatial and temporal dependencies. In contrast to standard Transformers that entangle spatial and temporal attention, CBraMod's design uses two parallel self-attention streams: one operates across the channel dimension to model spatial relationships between electrodes, while the other operates along the time dimension to model temporal dynamics. This separated attention mechanism addresses the heterogeneity of EEG signals and allows the model to learn rich spatiotemporal features. CBraMod is pre-trained on a large EEG corpus using a patch-based masked reconstruction objective, where patches of the input are masked and the model learns to reconstruct them, akin to a masked autoencoder for EEG. Furthermore, it introduces an asymmetric conditional positional encoding scheme to effectively handle varying EEG channel layouts and session formats. Thanks to these innovations, CBraMod has achieved state-of-the-art performance across a broad range of BCI tasks (evaluated on up to 10 different EEG datasets), demonstrating excellent generalizability. In our experiments, we employ the official CBraMod code and pre-trained model checkpoint made available by the authors (`https://github.com/wjq-learning/CBraMod`).

**ST-EEGFormer**    Three different ST-EEGFormers (small, base, and large models) were pre-trained using the method described in Appendix E. Details about the benchmarked model can be found in table E.1.

### F.1.2    MODEL TRAINING STRATEGIES

We evaluate two training strategies for foundation models: linear probing and fine-tuning. In the linear probing setting, the pre-trained backbone is kept frozen, and only a classification head is trained on the downstream task, thereby directly assessing the representational quality of the pretrained embeddings. In the fine-tuning setting, all model parameters—including the backbone—are updated jointly with the classification head, allowing the model to adapt its internal representations to the specific downstream dataset.

As detailed in Table F.1, the classification head architecture varies considerably across foundation models, which may strongly influence downstream performance. Two main sources of variation are: (i) token fusion strategy and (ii) classification head design. For token fusion, some models (e.g., BENDR, EEGPT, CBraMod) flatten all tokens without compression, while others compute the average token embedding. For classification heads, while LaBraM and our ST-EEGFormer adopt the simplest ViT-style design (average pooling followed by a linear layer), BIOT and CBraMod introduce non-linear activations (ELU), and EEGPT and CBraMod employ multiple stacked layers. Notably, CBraMod combines a flattening strategy with a multi-layer head, resulting in an exceptionally large head (22.44M parameters)—even exceeding the size of its backbone (4.88M).

These architectural inconsistencies across models have not been systematically compared in prior work, and have often been ignored, yet they likely introduce significant performance differences in downstream tasks. Motivated by this observation, our proposed ST-EEGFormer adopts a consistent and minimal design: averaging token embeddings followed by a single linear layer, thereby eliminating potential confounds from large and heterogeneous classification heads.

Table F.1: Implementation details of EEG foundation models. The classification head parameters are computed for a 62-channel, 3-second input with a 4-class output.

| Model | Sampling rate (Hz) | Data Normalization | Classification Head | Head Params |
|---|---|---|---|---|
| BENDR | 256 | [-1, 1] standardization | Flatten-Linear | 0.016388M |
| BIOT | 200 | 95-percentile standardization | Avg-ELU-Linear | 0.001028M |
| LaBraM | 200 | rescale to $0.1mV$ | Avg-Linear | 0.000804M |
| EEGPT | 256 | rescale to $1mV$ | Flatten-Linear-Flatten-Linear | 0.033556M |
| CBraMod | 200 | rescale to $0.1mV$ | Flatten-Linear-ELU-Linear-ELU-Linear | 22.441604M |
| ST-EEGFormer-s | 128 | z-standardization | Avg-Linear | 0.002052M |
| ST-EEGFormer-b | 128 | z-standardization | Avg-Linear | 0.003076M |
| ST-EEGFormer-l | 128 | z-standardization | Avg-Linear | 0.004100M |

Table F.2: ST-EEGFormer training configurations under two strategies.

(a) Fine-tuning

| CONFIG | VALUE |
|---|---|
| Optimizer | AdamW |
| Base learning rate | 5e-4 |
| Weight decay | 0.05 |
| Optimizer momentum | $\beta_1, \beta_2 = 0.9, 0.999$ |
| Layer-wise LR decay (Bao et al., 2022; Clark et al., 2020) | 0.75 |
| Batch size | 64 |
| LR schedule | Cosine decay |
| Warmup epochs | 10 |
| Training epochs | 100 |
| Label smoothing (Szegedy et al., 2015) | 0.1 |
| Drop path (Huang et al., 2016) | 0.1 |

(b) Linear probing

| CONFIG | VALUE |
|---|---|
| Optimizer | AdamW |
| Base learning rate | 0.005 |
| Weight decay | 0.05 |
| Optimizer momentum | $\beta_1, \beta_2 = 0.9, 0.999$ |
| Batch size | 64 |
| LR schedule | Cosine decay |
| Warmup epochs | 10 |
| Training epochs | 100 |
| Label smoothing | 0.1 |

During the training of foundation models, we adopt several optimization strategies summarized below. For ST-EEGFormer, both fine-tuning and linear probing follow the standard Vision Transformer (ViT) practice, as shown in Table F.2. For LaBraM, fine-tuning additionally incorporates layer-wise learning rate decay (LRD) and skips weight decay on specific parameters, as detailed in Table F.3. For all other foundation models, a default fine-tuning and linear-probing strategy is used, summarized in Table F.5. Finally, for LOO fine-tuning experiments, we adopt a lighter configuration with reduced learning rate, smaller batch size, and fewer epochs, as shown in Table F.4.

## F.2 CLASSIC NN MODELS

In this study, the following classic neural network EEG decoders are benchmarked across all downstream tasks.

### F.2.1 MODEL INTRODUCTION

**DeepConvNet** (Schirrmeister et al., 2017) is a deep convolutional neural network architecture developed for EEG signal decoding. It consists of a series of convolutional layers (for temporal feature extraction and spatial filtering), each typically followed by a nonlinear activation and pooling, which progressively transform raw multi-channel EEG data into higher-level representations. By leveraging a deeper hierarchy of conv-pooling blocks, DeepConvNet can automatically learn complex discriminative patterns from the data without any handcrafting features. It has become a standard baseline in brain–computer interface research, demonstrating that sufficiently deep CNNs can achieve strong performance on tasks like motor imagery classification and EEG-based pathology detection. The corresponding model architecture can be found in Table F.6.

Table F.3: LaBraM fine-tuning configuration.

| CONFIG | VALUE |
| --- | --- |
| Optimizer | AdamW |
| Base learning rate | 5e-4 |
| Weight decay | 0.05 |
| Optimizer momentum | $\beta_1, \beta_2 = 0.9, 0.999$ |
| Layer-wise LR decay | 0.75 |
| Batch size | 64 |
| LR schedule | Cosine decay |
| Warmup epochs | 10 |
| Training epochs | 100 |
| Label smoothing | 0.1 |

Table F.4: LOO fine-tuning configuration for foundation models.

| CONFIG | VALUE |
| --- | --- |
| Optimizer | AdamW |
| Base learning rate | 5e-5 |
| Weight decay | 0.01 |
| Optimizer momentum | $\beta_1, \beta_2 = 0.9, 0.999$ |
| Batch size | 32 |
| Training epochs | 50 |
| Warmup epochs | 5 |
| Label smoothing | 0.1 |

Table F.5: Default training configurations under two strategies.

(a) Fine-tuning

| CONFIG | VALUE |
| --- | --- |
| Optimizer | AdamW |
| Base learning rate | 5e-4 |
| Weight decay | 0.05 |
| Optimizer momentum | $\beta_1, \beta_2 = 0.9, 0.999$ |
| Batch size | 64 |
| LR schedule | Cosine decay |
| Warmup epochs | 10 |
| Training epochs | 100 |
| Label smoothing | 0.1 |

(b) Linear probing

| CONFIG | VALUE |
| --- | --- |
| Optimizer | AdamW |
| Base learning rate | 0.005 |
| Weight decay | 0.05 |
| Optimizer momentum | $\beta_1, \beta_2 = 0.9, 0.999$ |
| Batch size | 64 |
| LR schedule | Cosine decay |
| Warmup epochs | 10 |
| Training epochs | 100 |
| Label smoothing | 0.1 |

**EEGNet** (Lawhern et al., 2018) is a compact CNN architecture specifically tailored for EEG-based brain–computer interfaces. It employs depthwise separable convolutions to efficiently extract features, essentially splitting the filtering operation into temporal convolution (to capture frequency-specific patterns in each channel) and spatial convolution (to learn relationships across channels). This lightweight design drastically reduces the number of trainable parameters while still capturing key temporal-spectral characteristics of EEG signals. In practice, EEGNet has proven effective across many EEG decoding tasks and is widely used as a benchmark model, valued for its balance of simplicity, efficiency, and strong classification performance. In this study, the architecture of EEGNet follows the original implementation as shown in Table F.7

**EEG Conformer** (Song et al., 2023) is a hybrid convolutional–Transformer network designed to capture both local features and long-range dependencies in EEG data. Its architecture integrates an initial convolutional module that learns low-level temporal patterns and spatial features from the input signals, followed by a self-attention based Transformer module that models global temporal correlations. By uniting CNN and Transformer components in this way, the EEG Conformer can leverage the strengths of both: identifying fine-grained short-term EEG patterns as well as broader context across time. This approach has achieved state-of-the-art results on various EEG classification benchmarks, establishing the EEG Conformer as a leading example of modern EEG decoding architectures. The corresponding model architecture can be found in Table F.8.

**CTNet** (Convolutional Transformer Network) (Zhao et al., 2024) is another hybrid model combining convolutional feature extraction with Transformer-based attention, introduced for high-performance EEG signal classification (with a particular focus on motor imagery decoding). In this architecture, a front-end convolutional module—inspired by earlier EEG-specific networks like EEGNet—first extracts temporally filtered and spatially filtered features from the raw EEG, producing a condensed feature sequence. That sequence is then passed into a Transformer encoder module, which uses self-attention to capture global temporal dependencies and refine the representation before final classification. By integrating CNN-driven local pattern learning with global sequence

Table F.6: DeepConvNet architecture. Input EEG data consist of $N_{ch}$ channels and $L$ time samples (with sampling rate $f_s$, so $L = f_s \times$ trial duration). The network comprises four convolutional-pooling blocks with increasing filters (25, 50, 100, 200). A dropout layer ($p = 0.5$) follows each pooling.

| Layer | Type | Input shape | Output shape | Kernels | Kernel size | Stride | Padding |
|---|---|---|---|---|---|---|---|
| 0 | Input | $(N_{ch} \times L)$ | $(N_{ch} \times L)$ | NA | NA | NA | NA |
| 1 | Conv2d (temporal) | $(N_{ch} \times L)$ | $(25 \times N_{ch} \times L)$ | 25 | $(1, 5)$ | $(1, 1)$ | same |
| 2 | Conv2d (spatial) | $(25 \times N_{ch} \times L)$ | $(25 \times 1 \times L)$ | 25 | $(N_{ch}, 1)$ | $(1, 1)$ | $(0,0)$ |
| 3 | BatchNorm2d | $(25 \times 1 \times L)$ | $(25 \times 1 \times L)$ | NA | NA | NA | NA |
| 4 | ELU | $(25 \times 1 \times L)$ | $(25 \times 1 \times L)$ | NA | NA | NA | NA |
| 5 | MaxPool2d | $(25 \times 1 \times L)$ | $(25 \times 1 \times L/3)$ | NA | $(1, 3)$ | $(1, 3)$ | $(0,0)$ |
| 6 | Dropout | $(25 \times 1 \times L/3)$ | $(25 \times 1 \times L/3)$ | NA | NA | NA | NA |
| 7 | Conv2d | $(25 \times 1 \times L/3)$ | $(50 \times 1 \times L/3)$ | 50 | $(1, 5)$ | $(1, 1)$ | same |
| 8 | BatchNorm2d | $(50 \times 1 \times L/3)$ | $(50 \times 1 \times L/3)$ | NA | NA | NA | NA |
| 9 | ELU | $(50 \times 1 \times L/3)$ | $(50 \times 1 \times L/3)$ | NA | NA | NA | NA |
| 10 | MaxPool2d | $(50 \times 1 \times L/3)$ | $(50 \times 1 \times L/9)$ | NA | $(1, 3)$ | $(1, 3)$ | $(0,0)$ |
| 11 | Dropout | $(50 \times 1 \times L/9)$ | $(50 \times 1 \times L/9)$ | NA | NA | NA | NA |
| 12 | Conv2d | $(50 \times 1 \times L/9)$ | $(100 \times 1 \times L/9)$ | 100 | $(1, 5)$ | $(1, 1)$ | same |
| 13 | BatchNorm2d | $(100 \times 1 \times L/9)$ | $(100 \times 1 \times L/9)$ | NA | NA | NA | NA |
| 14 | ELU | $(100 \times 1 \times L/9)$ | $(100 \times 1 \times L/9)$ | NA | NA | NA | NA |
| 15 | MaxPool2d | $(100 \times 1 \times L/9)$ | $(100 \times 1 \times L/27)$ | NA | $(1, 3)$ | $(1, 3)$ | $(0,0)$ |
| 16 | Dropout | $(100 \times 1 \times L/27)$ | $(100 \times 1 \times L/27)$ | NA | NA | NA | NA |
| 17 | Conv2d | $(100 \times 1 \times L/27)$ | $(200 \times 1 \times L/27)$ | 200 | $(1, 5)$ | $(1, 1)$ | same |
| 18 | BatchNorm2d | $(200 \times 1 \times L/27)$ | $(200 \times 1 \times L/27)$ | NA | NA | NA | NA |
| 19 | ELU | $(200 \times 1 \times L/27)$ | $(200 \times 1 \times L/27)$ | NA | NA | NA | NA |
| 20 | MaxPool2d | $(200 \times 1 \times L/27)$ | $(200 \times 1 \times L/81)$ | NA | $(1, 3)$ | $(1, 3)$ | $(0,0)$ |
| 21 | Dropout | $(200 \times 1 \times L/81)$ | $(200 \times 1 \times L/81)$ | NA | NA | NA | NA |
| 22 | Linear (Softmax) | $(200 * L/81)$ | $(N_{class})$ | NA | NA | NA | NA |

Table F.7: EEGNet architecture. Input EEG data consist of $N_{ch}$ channels and $L$ time samples. The output corresponds to $N_{class}$, representing the number of different classes to classify. The dropout ratio is set to 0.40.

| Layer | Type | Input shape | Output shape | Kernels | Kernel size | Stride | Padding |
|---|---|---|---|---|---|---|---|
| 0 | Input | $(N_{ch} \times L)$ | $(N_{ch} \times L)$ | NA | NA | NA | NA |
| 1 | Conv2d | $(N_{ch} \times L)$ | $(8 \times N_{ch} \times L)$ | 8 | $(1, fs/2)$ | $(1, 1)$ | same |
| 2 | BatchNorm2d | $(8 \times N_{ch} \times L)$ | $(8 \times N_{ch} \times L)$ | NA | NA | NA | NA |
| 3 | Depthwise Conv2d | $(8 \times N_{ch} \times L)$ | $(32 \times 1 \times L)$ | 32 | $(N_{ch}, 1)$ | $(1, 1)$ | $(0, 0)$ |
| 4 | BatchNorm2d | $(32 \times 1 \times L)$ | $(32 \times 1 \times L)$ | NA | NA | NA | NA |
| 5 | ELU | $(32 \times 1 \times L)$ | $(32 \times 1 \times L)$ | NA | NA | NA | NA |
| 6 | AvgPool2d | $(32 \times 1 \times L)$ | $(32 \times 1 \times L/(fs/32))$ | NA | $(1, fs/32)$ | $(1, fs/32)$ | $(0, 0)$ |
| 7 | Dropout | $(32 \times 1 \times L/(fs/32))$ | $(32 \times 1 \times L/(fs/32))$ | NA | NA | NA | NA |
| 8 | Seperable Conv2d | $(32 \times 1 \times L/(fs/32))$ | $(32 \times 1 \times L/(fs/32))$ | 32 | $(1, 16)$ | $(1, 1)$ | same |
| 9 | BatchNorm2d | $(32 \times 1 \times L/(fs/32))$ | $(32 \times 1 \times L/(fs/32))$ | NA | NA | NA | NA |
| 10 | ELU | $(32 \times 1 \times L/(fs/32))$ | $(32 \times 1 \times L/(fs/32))$ | NA | NA | NA | NA |
| 11 | AvgPool2d | $(32 \times 1 \times L/(fs/32))$ | $(32 \times 1 \times L/(fs/8))$ | NA | $(1, 4)$ | $(1, 4)$ | $(0, 0)$ |
| 12 | Dropout | $(32 \times 1 \times L/(fs/8))$ | $(32 \times 1 \times L/(fs/8))$ | NA | NA | NA | NA |
| 13 | Linear | $(1 \times (256L/fs))$ | $(1 \times N_{class})$ | NA | NA | NA | NA |

modeling, CTNet effectively leverages both fine-scale EEG features and long-range context, leading to improved accuracy in EEG decoding and exemplifying the advance of CNN–Transformer hybrids in brain signal analysis. The corresponding model architecture can be found in Table F.9.

### F.2.2 MODEL TRAINING STRATEGIES

For all classic NN models (DeepConvNet, EEGNet, EEG Conformer, and CTNet), we adopt a unified and straightforward training strategy, as summarized in Table F.10. The table reports the settings for two scenarios side by side: the left panel corresponds to population training (full training from scratch), while the right panel corresponds to fine-tuning (adaptation on held-out subjects). To ensure fair comparison in the LOO performance-drop setting, the fine-tuning protocol for these models is aligned with that of the foundation models. This design choice avoids confounding factors such as differences in learning rate or training epochs, ensuring that performance differences can be attributed to the models themselves rather than to training hyperparameters.

Table F.8: EEG Conformer architecture. Input EEG data consist of $N_{ch}$ channels and $L$ time samples. The output corresponds to $N_{class}$, representing the number of different classes to classify.

| Layer | Name | Type | Layer specific settings | Output shape |
|---|---|---|---|---|
| 0 | Input | NA | NA | $(N_{ch} \times \mathrm{L})$ |
| 1 | CNN-module | Conv2d | kernel size:$(1, 25)$
number of kernels:40 | $(40 \times N_{ch} \times L)$ |
| 2 | CNN-module | Conv2d | kernel size:$(N_{ch}, 1)$
number of kernels:40 | $(40 \times 1 \times L)$ |
| 3 | CNN-module | BatchNorm2d | NA | $(40 \times 1 \times L)$ |
| 4 | CNN-module | ELU | NA | $(40 \times 1 \times L)$ |
| 5 | CNN-module | AvgPool2d | kernel size:$(1, 37)$
stride:$(1, 7)$ | $(40, \lfloor \frac{L-37}{7} \rfloor + 1)$ |
| 6 | CNN-module | Dropout | dropout_p=0.5 | $(40, \lfloor \frac{L-37}{7} \rfloor + 1)$ |
| 7 | CNN-module | Conv2d | kernel size:$(1, 1)$
number of kernels:40 | $(40, \lfloor \frac{L-37}{7} \rfloor + 1)$ |
| 8 | Transformer-module | Transformer encoder layers | embed size:40
number of heads:10
drop_p:0.5
forward_expansion:4
forward_drop_p:0.5
depth:6 | $(40, \lfloor \frac{L-37}{7} \rfloor + 1)$ |
| 9 | Classification head | Linear | weights and bias shape:
$(40 \times \lfloor \frac{L-37}{7} \rfloor + 1, 256)$ | $(1, 256)$ |
| 10 | Classification head | ELU | NA | $(1, 256)$ |
| 11 | Classification head | Dropout | dropout_p=0.5 | $(1, 256)$ |
| 12 | Classification head | Linear | weights and bias shape:
$(256, 32)$ | $(1, 32)$ |
| 13 | Classification head | Elu | NA | $(1, 32)$ |
| 14 | Classification head | Dropout | dropout_p=0.3 | $(1, 32)$ |
| 15 | Classification head | Linear | weights and bias shape:
$(32, N_{class})$ | $(1, N_{class})$ |

Table F.9: CTNet architecture. Input EEG data consist of $N_{ch}$ channels and $L$ time samples (with sampling rate $f_s$). The convolutional front-end applies temporal, spatial, and separable convolutions to extract local features, followed by a Transformer encoder for global dependencies, and a fully connected classifier.

| Layer | Name | Type | Layer specific settings | Output shape |
|---|---|---|---|---|
| 0 | Input | NA | NA | $(N_{ch} \times L)$ |
| 1 | CNN-module | Conv2d (temporal) | kernel size: $(1, 64)$, kernels: 20 | $(20 \times N_{ch} \times L)$ |
| 2 | CNN-module | Conv2d (spatial) | kernel size: $(N_{ch}, 1)$, kernels: 40 | $(40 \times 1 \times L)$ |
| 3 | CNN-module | BatchNorm2d | NA | $(40 \times 1 \times L)$ |
| 4 | CNN-module | ELU | NA | $(40 \times 1 \times L)$ |
| 5 | CNN-module | AvgPool2d | kernel size: $(1, 8)$, stride: $(1, 8)$ | $(40 \times 1 \times L/8)$ |
| 6 | CNN-module | Dropout | $p = 0.5$ | $(40 \times 1 \times L/8)$ |
| 7 | CNN-module | Separable Conv2d | kernel size: $(1, 16)$, kernels: 40 | $(40 \times 1 \times L/8)$ |
| 8 | CNN-module | BatchNorm2d | NA | $(40 \times 1 \times L/8)$ |
| 9 | CNN-module | ELU | NA | $(40 \times 1 \times L/8)$ |
| 10 | CNN-module | AvgPool2d | kernel size: $(1, 8)$, stride: $(1, 8)$ | $(40 \times 1 \times L/64)$ |
| 11 | CNN-module | Dropout | $p = 0.5$ | $(40 \times 1 \times L/64)$ |
| 12 | Transformer-module | Transformer encoder | embed=40, heads=4, depth=6, drop=0.5 | $(40 \times L/64)$ |
| 13 | Classification head | Linear | $(40 * L/64, 256)$ | $(1, 256)$ |
| 14 | Classification head | ELU | NA | $(1, 256)$ |
| 15 | Classification head | Dropout | $p = 0.5$ | $(1, 256)$ |
| 16 | Classification head | Linear (Softmax) | $(256, N_{class})$ | $(1, N_{class})$ |

## F.3 CLASSIC NON-NN MODELS

We included different classic non-neural network models for each type of downstream tasks.

Table F.10: Training settings for classic NN models under different strategies.

(a) Population training

| Config | Value |
|---|---|
| Optimizer | AdamW |
| Base learning rate | $3 \times 10^{-3}$ |
| Weight decay | 0.05 |
| Optimizer momentum | $\beta_1, \beta_2 = 0.9, 0.999$ |
| Batch size | 64 |
| Epochs | 100 |
| Learning rate schedule | cosine decay |
| Warmup epochs | 10 |
| Label smoothing | 0.1 |

(b) Fine-tuning

| Config | Value |
|---|---|
| Optimizer | AdamW |
| Base learning rate | $5 \times 10^{-5}$ |
| Weight decay | 0.05 |
| Optimizer momentum | $\beta_1, \beta_2 = 0.9, 0.999$ |
| Batch size | 32 |
| Epochs | 50 |
| Label smoothing | 0.1 |

### F.3.1 MOVEMENT AND SPEECH CLASSIFICATION TASKS

For the BCI-IV-2A, Upper Limb Motor Execution, Upper Limb Motor Imagination, and Inner Speech datasets, which involve movement or speech tasks, we implemented two types of decoding pipelines. The first category was CSP-based methods, including CSP-LDA, CSP-SVM (Ramoser et al., 2000), FBCSP-LDA, and FBCSP-SVM (Ang et al., 2008). The second category was Riemannian geometry-based classifiers, including Minimum Distance to Mean (MDM) (Barachant et al., 2012a), Fisher Geodesic MDM (FgMDM) (Barachant et al., 2012a), and tangent space mapping with ElasticNet (TS-ElasticNet) (Corsi et al., 2022). The input EEG signals were first band-pass filtered (3rd-order Butterworth filter) into the 4-40 Hz frequency band. Alternatively, when using filter banks, we band-passed the EEG signals into 9 consecutive frequency bands according to the formula: $4k$ - $4(k+1)$ Hz, $k = 1, 2, 3, \ldots, 9$. After preprocessing, the filtered signals were subjected to feature extraction. CSP learns spatial filters by minimizing the variance of power features within each class while maximizing the variance between classes in a supervised manner. In this study, we used four spatial filters, resulting in a four-dimensional feature vector for each EEG epoch. FBCSP extends this approach by applying CSP to multiple frequency bands, thereby producing nine times more features per epoch. The extracted features were used as inputs to LDA and SVM with a radial basis function kernel. In addition, the filtered EEG signals were transformed into covariance matrices, which are Symmetric Positive Definite (SPD) and reside in the Riemannian space. The MDM classifier computes the class centers in the Riemannian space and assigns unseen samples based on the geodesic distance between their covariance matrices and the class centers. FgMDM incorporates Fisher LDA into MDM, thereby enhancing robustness against noise. Furthermore, samples on the Riemannian manifold can be projected onto the tangent space, yielding vectorized feature representations. TS-ElasticNet then applies the ElasticNet model to these projected features. For all covariance matrix computations, the Oracle Approximating Shrinkage (OAS) estimator was employed to ensure robust estimation.

Signal band-pass filtering and CSP feature extraction were implemented using the MNE toolbox (MNE v1.9.0: `https://mne.tools/stable/generated/mne.filter.filter_data.html`). LDA, SVM, and ElasticNet were based on scikit-learn (Scikit-learn v1.4.2: `https://scikit-learn.org/1.4/modules/classes.html`), while MDM, FgMDM, and covariance estimation were implemented using pyRiemann (pyRiemann v0.6: `https://pyriemann.readthedocs.io/en/latest/api.html`).

### F.3.2 ERN DETECTION TASK

For the ERN detection task on the ERN dataset, we implemented five baseline models: xDAWN-LDA (Rivet et al., 2009), xDAWNCov-MDM (Barachant, 2014), xDAWNCov-TS-SVM (Chevallier et al., 2018), ERPCov-MDM (Barachant & Congedo, 2014), and DCPM (Xiao et al., 2020). All models were trained on EEG signals that were band-pass filtered between 1–20 Hz using a 3rd-order Butterworth filter. xDAWN is a widely used spatial filtering technique that improves the signal-to-noise ratio of evoked potentials (Rivet et al., 2009). For xDAWN-LDA, we applied xDAWN with two spatial filters to enhance signal quality, followed by downsampling to 32 Hz. Temporal features from all channels were then concatenated into vector representations and used to train an LDA clas-

sifier. For xDAWNCov-MDM, the band-pass filtered EEG signals were augmented with prototype matrices (trial-averaged template) spatially filtered by xDAWN (four spatial filters). The augmented signals were subsequently transformed into covariance matrices, which were classified using MDM. For xDAWNCov-TS-SVM, the same data augmentation procedure was applied, after which the covariance matrices were projected onto the tangent space to obtain vectorized representations. These features were then used to train an SVM classifier with a radial basis function kernel. ERPCov-MDM is a simplified version of xDAWNCov-MDM that omits the spatial filtering step. Finally, DCPM is an ensemble method that integrates variations of LDA and canonical correlation analysis (CCA) for ERP classification (Xiao et al., 2020).

The xDAWN algorithm, covariance estimation, and MDM classifier were implemented using pyRiemann , while LDA, SVM, and CCA were based on scikit-learn.

### F.3.3 ALZHEIMER'S DIAGNOSIS TASK

For Alzheimer's diagnosis using the Alzheimer's dataset, we implemented four decoding pipelines described in the dataset paper (Miltiadous et al., 2023). Each pipeline extracts Relative Band Power (RBP) features and applies them to different classifiers: Random Forest (RBP-RF), SVM (RBP-SVM), k-Nearest Neighbors (RBP-kNN), and LightGBM (RBP-LightGBM). For each trial, EEG signals from all channels were decomposed using the Welch method to estimate the power spectral density (PSD). RBP features were computed by integrating the PSD within standard frequency bands (Delta: 0.5–4 Hz, Theta: 4–8 Hz, Alpha: 8–13 Hz, Beta: 13–25 Hz, Gamma: 25–45 Hz) and normalizing by the total power across 0.5–45 Hz. The resulting relative powers from all channels and frequency bands were concatenated to form a feature vector for each trial. The extracted RBP features were used to train RF (100 trees), SVM (Polynomial kernel), kNN, and LightGBM (100 boosted trees with a learning rate of 0.05).

The implementations of RF, SVM, and kNN were based on scikit-learn , while LightGBM was implemented using the official LightGBM package (LightGBM v4.6.0: `https://lightgbm.readthedocs.io/en/v4.6.0/Python-API.html`).

### F.3.4 SSVEP TARGET RECOGNITION TASK

For SSVEP target recognition on the Binocular SSVEP dataset, we implemented two decoding models—Filter Bank Canonical Correlation Analysis (FBCCA) and Task-Related Component Analysis (TRCA)—as described in the dataset paper (Yike et al., 2024). Both models relied on filter banks for EEG preprocessing, with five frequency bands defined as [5, 95] Hz, [12, 95] Hz, [19, 95] Hz, [27, 95] Hz, and [35, 95] Hz. For FBCCA, sinusoidal templates with five harmonics were constructed for each target according to its stimulation frequencies. It should be noted that we used the binocular swap dataset for evaluation, which is not ideal for FBCCA since half of the targets share identical stimulation frequencies. Nevertheless, FBCCA was included to examine the potential of a training-free model. For TRCA, we adopted the ensemble version, where class-specific filters were combined to form a universal spatial filter. The filter bank weights followed the rule $k^{-1.25} + 0.25$, $k = 1, 2, 3, \ldots, 5$.

The implementations followed the provided code demo (`https://gigadb.org/dataset/102557`) and relied on the MEEGkit (MEEGkit v0.1.9:`https://github.com/nbara/python-meegkit`)).

## G BENCHMARK RESULTS

### G.1 BENCHMARK DATASET RESULTS

In this section, we show the results for each downstream task.

### G.1.1 ERN

Figure G.1 presents the benchmark results on the ERN dataset, reporting balanced accuracy across six evaluation protocols, along with aggregated model rankings and statistical significance analysis.

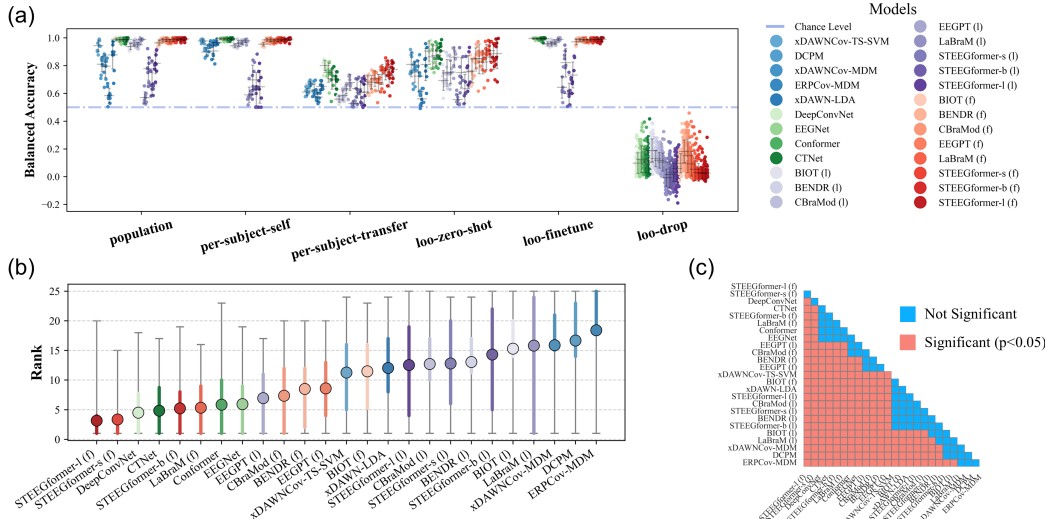

Figure G.1: Benchmark results on the ERN dataset. (a) Balanced accuracy of each model across six evaluation protocols: Population, Per-Subject (Self), Per-Subject (Transfer), LOO Zero-Shot, LOO Fine-Tune, and LOO Drop. Each dot represents one subject, with box plots summarizing distributions. Colors indicate model groups: blue—classic non-neural network decoders; green—classic neural networks; purple—foundation models (linear probing); red—foundation models (fine-tuning). (b) Aggregated average rank per model, ordered from best (left) to worst (right). Circles show mean rank, colored bars indicate ± standard deviation, and grey whiskers represent the minimum and maximum range. (c) Pairwise statistical significance matrix in the same model order as (b), computed via permutation tests ($n_{\text{resamples}} = 50,000$) with Bonferroni correction; red—significant ($p < 0.05$), blue—non-significant.

The task appears relatively simple, with many models achieving near-perfect accuracy. Nevertheless, several classic non-NN models and most linear-probed foundation models perform noticeably worse than the top performers. All models show reduced performance when transferred to unseen subjects, with a clear drop under the Per-Subject-Transfer and LOO Zero-Shot protocols. In the overall ranking, both the small and large variants of our proposed ST-EEGFormer achieve the highest scores, followed by the classic NN models DeepConvNet and CTNet. In general, almost all linear-probed foundation models—except EEGPT—rank among the lowest-performing methods.

### G.1.2 BCI-IV-2A

Figure G.2 summarizes the benchmark results on the BCI-IV-2A dataset. The results show that certain classic non-NN models (e.g., TS-ElasticNet) achieve top performance in the Per-Subject-Self and Per-Subject-Transfer protocols, but perform worse in the Population and LOO Zero-Shot settings. Classic NN models remain highly competitive across all protocols. Ranking and statistical analyses indicate that the best NN model, CTNet, is not statistically different from the overall top-ranked model, fine-tuned ST-EEGFormer-l, with both significantly outperforming all other models. All linear-probed foundation models perform the worst, ranking significantly below even the classic non-NN methods, except EEGPT.

### G.1.3 INNER SPEECH

Figure G.3 presents the benchmark results on the Inner Speech dataset. This task appears highly challenging, with most models achieving accuracies close to random chance. The best results are predominantly achieved by classic NN models, which occupy three of the top four positions, with fine-tuned CBraMod ranking second. However, overall performance differences across models are small, suggesting that the inner speech decoding task remains intrinsically difficult.

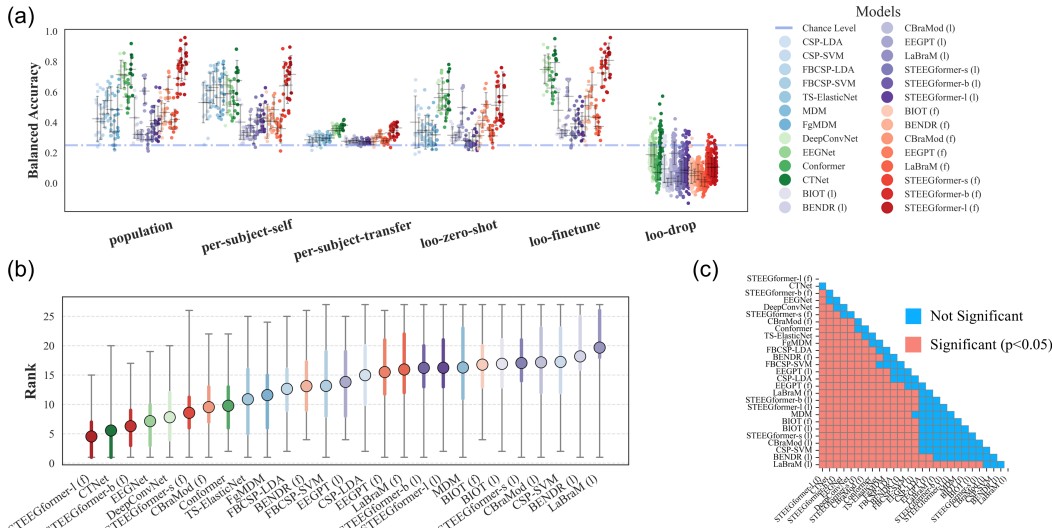

Figure G.2: Benchmark results on the BCI-IV-2A dataset, using the same notation and panel layout as in Figure G.1. Results are reported across six evaluation protocols, with colors denoting model groups and statistical significance assessed via permutation testing with Bonferroni correction.

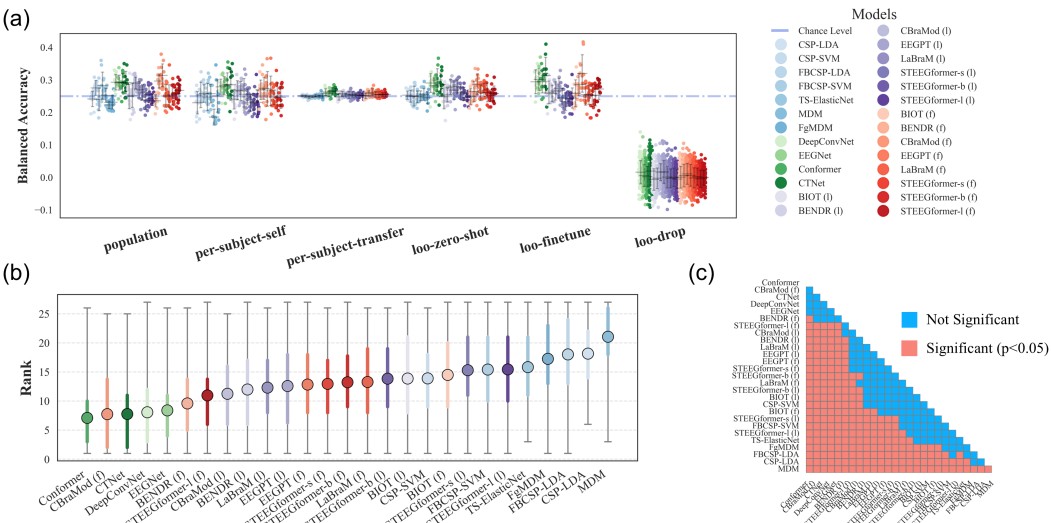

Figure G.3: Benchmark results on the Inner Speech dataset, using the same notation and panel layout as in Figure G.1. Results are reported across six evaluation protocols, with colors denoting model groups and statistical significance assessed via permutation testing with Bonferroni correction.

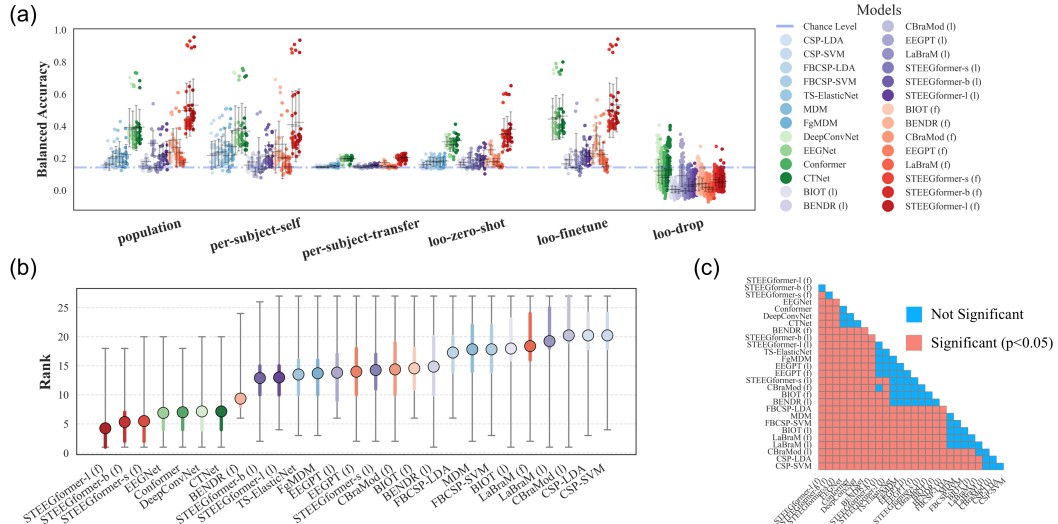

Figure G.4: Benchmark results on the Upper Limp Motor Execution dataset, using the same notation and panel layout as in Figure G.1. Results are reported across six evaluation protocols, with colors denoting model groups and statistical significance assessed via permutation testing with Bonferroni correction.

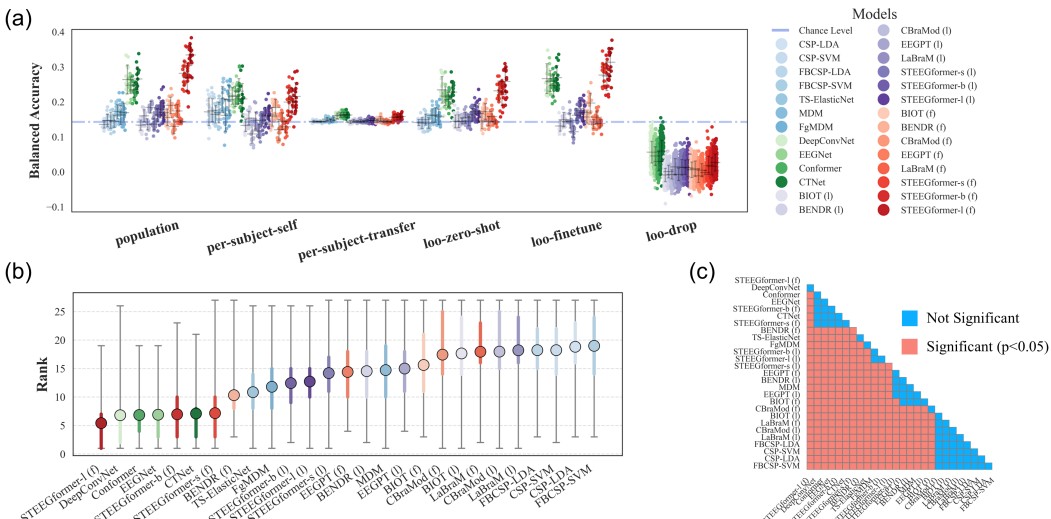

Figure G.5: Benchmark results on the Upper Limp Motor Imagination dataset, using the same notation and panel layout as in Figure G.1. Results are reported across six evaluation protocols, with colors denoting model groups and statistical significance assessed via permutation testing with Bonferroni correction.

### G.1.4 UPPER LIMB MOTOR EXECUTION

Figure G.4 presents the benchmark results on the Upper Limb Motor Execution dataset. Overall, classic non-NN methods perform worse than classic NN models, while our proposed ST-EEGFormer achieves the highest performance. A large degree of subject variability is evident, with accuracies ranging from above 90% to near chance level. Most foundation models—both fine-tuned and linear-probed—are statistically equivalent to each other and generally underperform compared to classic NN models.

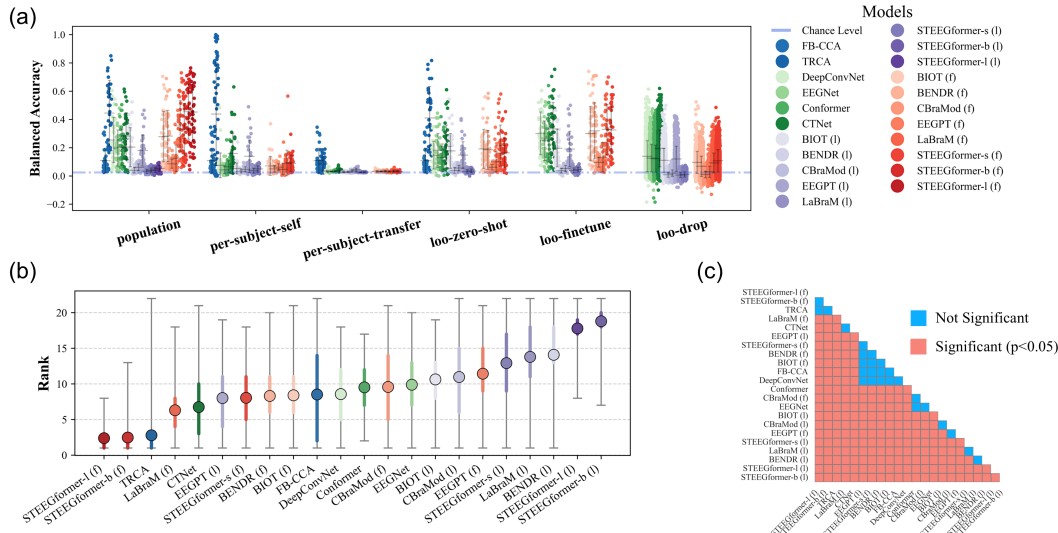

Figure G.6: Benchmark results on the Binocular SSVEP dataset, using only the synchronous trials (the first 1-s trial post-onset) using the same notation and panel layout as in Figure G.1. Results are reported across six evaluation protocols, with colors denoting model groups and statistical significance assessed via permutation testing with Bonferroni correction.

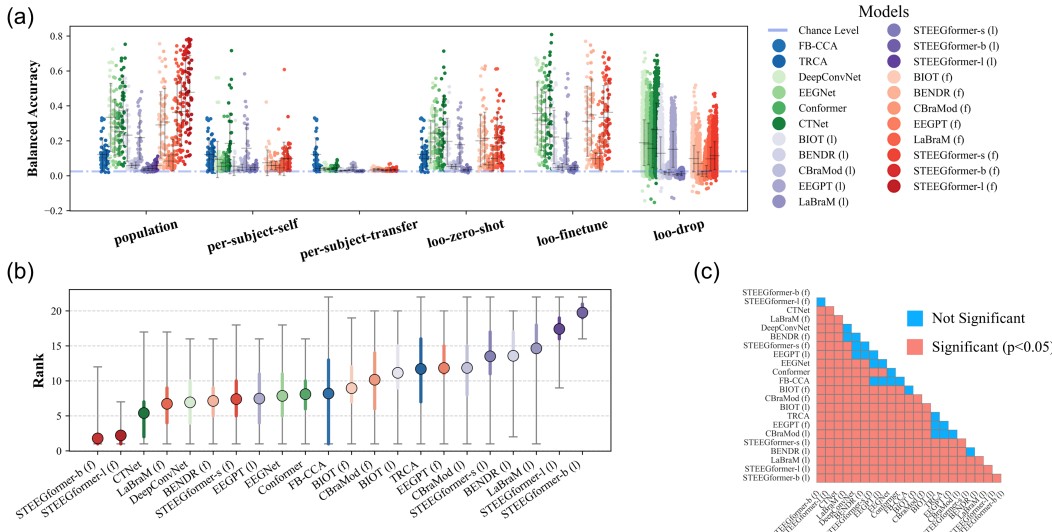

Figure G.7: Benchmark results on the Binocular SSVEP dataset, using the asynchronous trials (1-s trials cut by a sliding window) using the same notation and panel layout as in Figure G.1. Results are reported across six evaluation protocols, with colors denoting model groups and statistical significance assessed via permutation testing with Bonferroni correction.

### G.1.5 UPPER LIMB MOTOR IMAGINATION

Figure G.5 presents the benchmark results on the Upper Limb Motor Imagination dataset. Overall, a similar trend to that in Section G.1.4 is observed: classic non-NN methods perform worse than classic NN models, while our proposed ST-EEGFormer achieves the highest performance. However, overall accuracy drops substantially compared to the execution task, with the best subjects achieving below 40% accuracy, indicating that motor imagery is more variable and challenging. In this difficult setting, there is no statistically significant difference among the top-performing models, except for the fine-tuned large variant of ST-EEGFormer, which outperforms the rest.

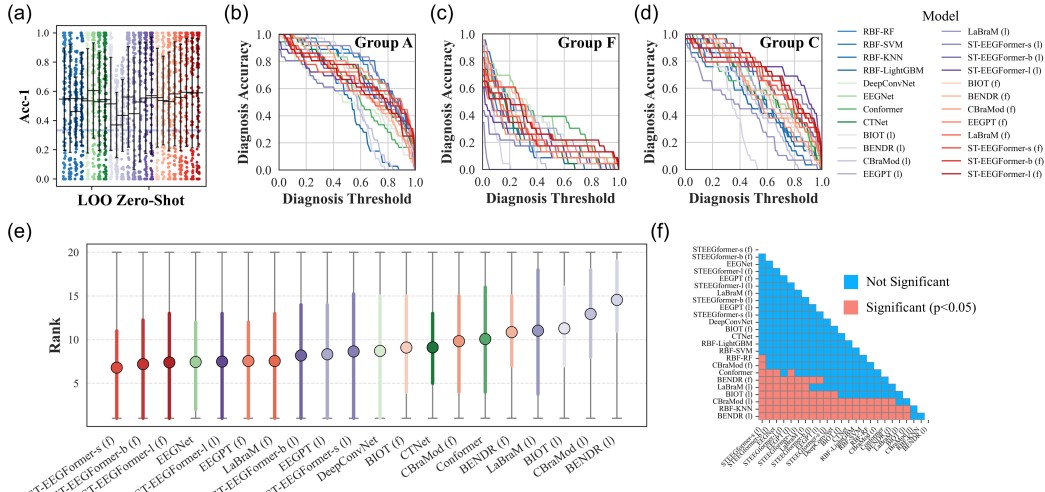

Figure G.8: Benchmark results on the Alzheimer's dataset. This figure summarizes subject-level diagnostic performance of different models under multiple evaluation views. (a) Model performance under the LOO Zero-Shot evaluation protocol. (b–d) Threshold–Accuracy curves for diagnosing Alzheimer's disease (A), frontal dementia (F), and healthy controls (C). Diagnosis is determined when the percentage of trials from a subject classified as the target class exceeds the diagnosis threshold. (e) Aggregated ranking of all models. (f) Pairwise statistical significance matrix, following the same format as Figure G.1. Model groups are color-coded, and statistical significance is assessed via permutation testing with Bonferroni correction.

### G.1.6 BINOCULAR SSVEP

For the Binocular SSVEP dataset, we evaluated two settings: synchronous trial classification (using the first 1-second trial post-onset) and asynchronous trial classification (using 1-second windows extracted via a sliding window). This design allows a fair comparison with classic non-NN methods, which are typically evaluated in synchronous settings. Results are shown in Figure G.6 and Figure G.7, respectively. We observe that TRCA performs exceptionally well in synchronous classification but degrades substantially in the asynchronous setting. In contrast, NN-based decoders, including foundation models, show stronger performance in the more challenging asynchronous condition, indicating better generalization to difficult tasks. Notably, in both settings, the training-free FB-CCA method outperforms many foundation models. These results illustrate that well-developed classic methods can achieve comparable—or even superior—performance on specific downstream BCI tasks with limited data, whereas NN-based approaches tend to generalize better when task difficulty increases, suggesting potential gains with larger downstream datasets.

### G.1.7 ALZHEIMER'S

Figure G.8 summarizes the benchmark results on the Alzheimer's dataset. Since this task is designed to mimic a clinical diagnostic application, we evaluated models only under the LOO Zero-Shot protocol. As shown in Figure G.8a, all models achieve similar performance, with some linear-probed foundation models showing slightly lower average accuracy. A closer look at the diagnosis accuracy for each group in Figures G.8b–d reveals that all models classify the Alzheimer's group (A) and the healthy control group (C) more easily than the frontal dementia group (F). Overall, most models do not differ significantly in performance on this dataset, with the weakest results observed for several linear-probed foundation models and the EEG Conformer.

### G.1.8 TUEV

Figure G.9 presents the benchmark results on TUEV. Fine-tuned foundation models perform significantly better than classic neural decoders; notably, the top seven models are all fine-tuned foundation models, with no significant differences among them. By contrast, linear-probed foundation models remain comparatively weak.

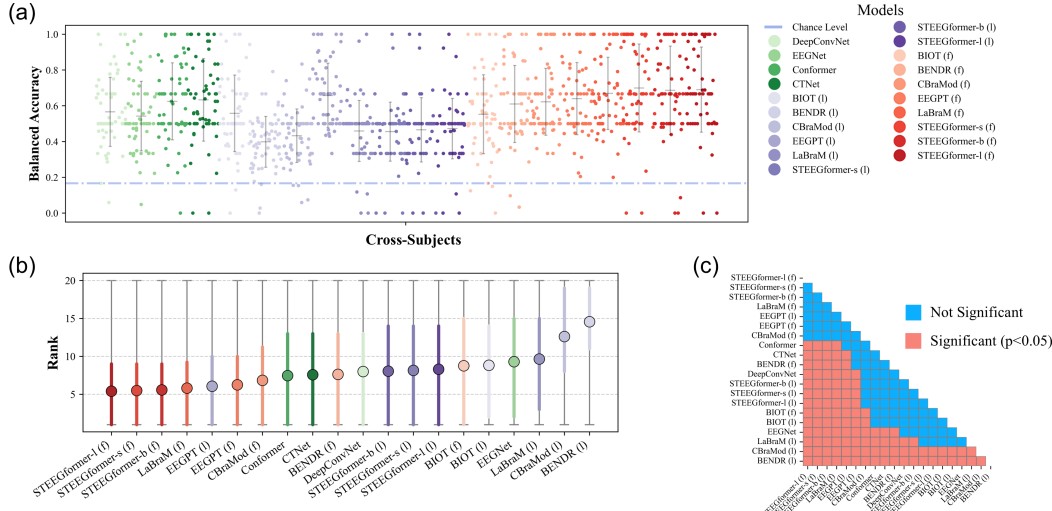

Figure G.9: Benchmark results on the TUEV dataset, using the same notation and panel layout as in Figure G.1. Results are reported using the conventional cross-subject protocol (LOO Zero-Shot), with colors denoting model groups and statistical significance assessed via permutation testing with Bonferroni correction.

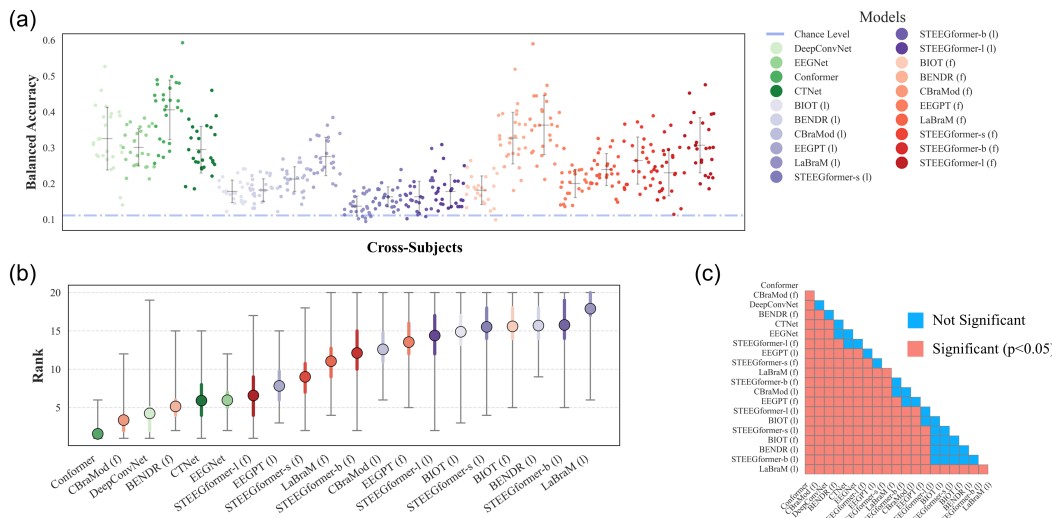

Figure G.10: Benchmark results on the FACED dataset, using the same notation and panel layout as in Figure G.1. Results are reported using the conventional cross-subject protocol (LOO Zero-Shot), with colors denoting model groups and statistical significance assessed via permutation testing with Bonferroni correction.

### G.1.9 FACED

Figure G.10 reports the benchmark results on FACED. The EEG Conformer is the top-performing model and significantly outperforms all others, followed by a fine-tuned CBraMod. Classic neural decoders remain highly competitive on this dataset, whereas linear-probed foundation models lag behind.

### G.1.10 DTU

Figure G.11 presents the benchmark results on the auditory regression DTU dataset. In this relatively simple regression task—predicting a single value at a time—classic NN models outperform all foundation models, with the notable observation that linear-probed foundation models surpass

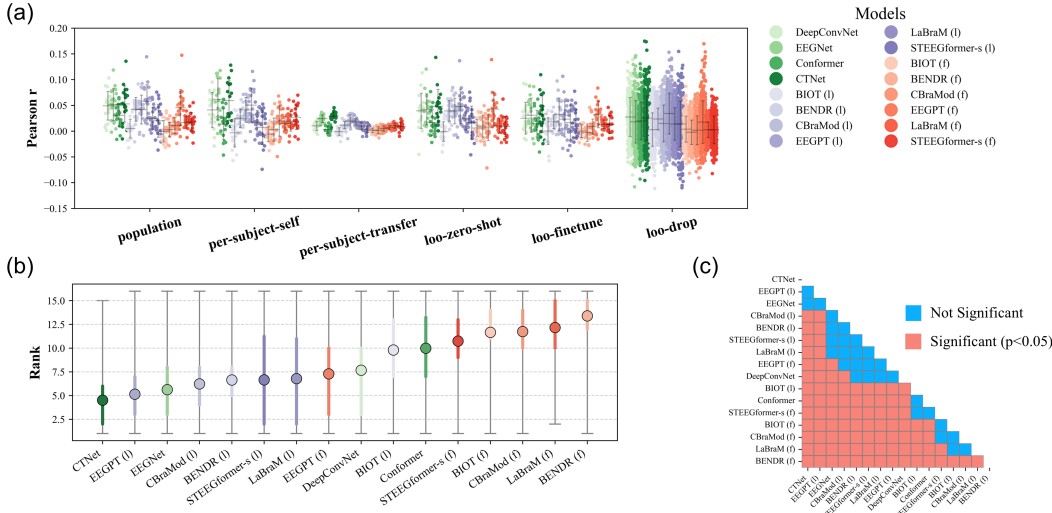

Figure G.11: Benchmark results on the DTU dataset, following the same notation and panel layout as in Figure G.1. In panel (a), scatter plots display the Pearson correlation coefficient ($R$). Panel (b) shows aggregated rankings computed from both MSE and Pearson $R$. Results are reported across six evaluation protocols, with colors indicating model groups, and statistical significance assessed via permutation testing with Bonferroni correction.

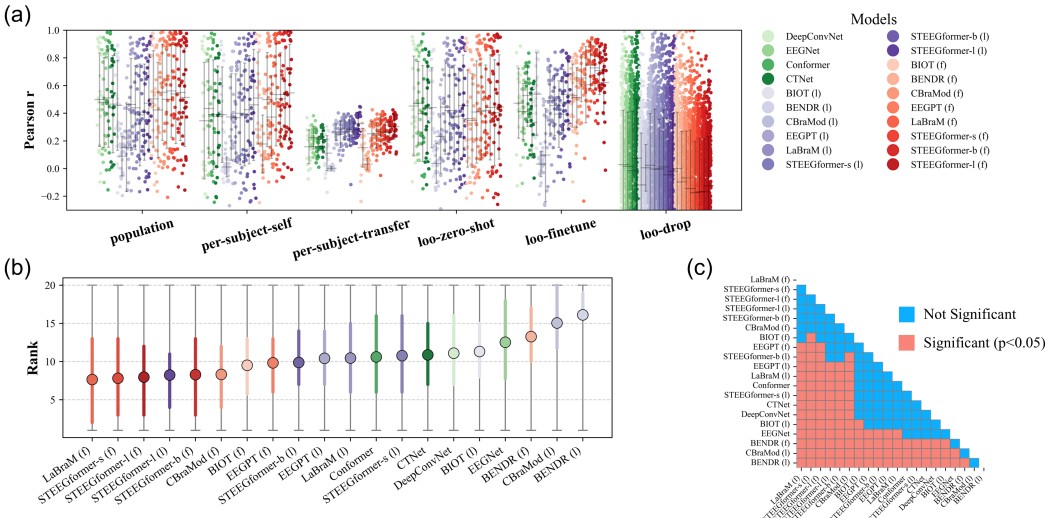

Figure G.12: Benchmark results on the SEED-VIG dataset, following the same notation and panel layout as in Figure G.1. In panel (a), scatter plots display the Pearson correlation coefficient ($R$). Panel (b) shows aggregated rankings computed from both MSE and Pearson $R$. Results are reported across six evaluation protocols, with colors indicating model groups, and statistical significance assessed via permutation testing with Bonferroni correction.

their fine-tuned counterparts. This trend may be due to the limited subject-specific data available for fine-tuning. However, even the best-performing NN model (CTNet) substantially underperforms compared to state-of-the-art regression models such as Sea-Wave (Yang et al., 2024), which achieve Pearson $R$ values around 0.2, whereas all benchmarked models remain below 0.1. These findings suggest that representations learned by EEG classification-oriented models transfer poorly to regression tasks. Future progress may require foundation models with dedicated encoder–decoder architectures explicitly designed for regression.

### G.1.11 SEED-VIG

Figure G.12 reports the benchmark results on the SEED-VIG vigilance regression task. Overall, fine-tuned foundation models outperform classic neural decoders, and there is no statistically significant difference among the seven best-performing models. On average, methods achieve Pearson correlations above 0.4 on this dataset, suggesting it is comparatively less challenging than the auditory regression task.

### G.1.12 SUMMARY OF ALL BENCHMARK RESULTS

Overall, the benchmark results vary considerably across datasets. Based on statistically evaluated performance rankings, the models can generally be grouped into three tiers. In most cases, the lowest tier consists of linear-probed foundation models. The top tier often includes the large fine-tuned ST-EEGFormer together with a few classic NN models such as CTNet, while certain classic non-neural methods achieve strong results but more often fall into the middle tier alongside other foundation models.

These findings underscore a critical point: performance gaps between models are highly task-dependent, and reporting a single accuracy value—as is common in many foundation model papers—can be misleading. Without statistical analysis, such reporting may obscure the fact that observed differences are not significant. For example, in the Alzheimer's dataset results (Figure G.2), one might conclude from accuracy alone that ST-EEGFormer-s (f) is the best model; however, it is statistically indistinguishable from 11 other models.

### G.2 LINEAR PROBING VS. FINE-TUNING

We compare the average performance of foundation models under linear probing and fine-tuning strategies. Table G.1 reports results aggregated across all six evaluation schemes for each model, while Table G.2 presents results for each evaluation scheme aggregated across models.

Table G.1: Accuracy (Mean $\pm$ Std) and Paired Wilcoxon Test p-values for Linear Probe and Fine-Tuning Across Foundation Models

| Model | Linear Probe | Fine-Tuning | P-Value |
|---|---|---|---|
| BIOT | $0.256 \pm 0.266$ | $0.301 \pm 0.271$ | $p = 6.84 \times 10^{-28}$ |
| BENDR | $0.177 \pm 0.267$ | $0.310 \pm 0.275$ | $p = 1.15 \times 10^{-43}$ |
| CBraMod | $0.186 \pm 0.261$ | $0.233 \pm 0.284$ | $p = 1.95 \times 10^{-27}$ |
| EEGPT | $0.295 \pm 0.265$ | $0.225 \pm 0.267$ | $p = 5.75 \times 10^{-37}$ |
| LaBraM | $0.153 \pm 0.230$ | $0.320 \pm 0.272$ | $p = 2.57 \times 10^{-43}$ |
| ST-EEGFormer-s | $0.179 \pm 0.252$ | $0.373 \pm 0.284$ | $p = 6.65 \times 10^{-46}$ |
| ST-EEGFormer-b | $0.425 \pm 0.261$ | $0.608 \pm 0.290$ | $p = 1.22 \times 10^{-17}$ |
| ST-EEGFormer-l | $0.428 \pm 0.283$ | $0.637 \pm 0.280$ | $p = 1.22 \times 10^{-17}$ |

Table G.2: Accuracy (Mean $\pm$ Std) and Paired Wilcoxon Test p-values for Linear Probe and Fine-Tuning Across All Datasets and Evaluation Protocols

| Evaluation Protocol | Linear Probe | Fine-Tuning | P-Value |
|---|---|---|---|
| Population | $0.228 \pm 0.238$ | $0.349 \pm 0.263$ | $p = 2.62 \times 10^{-79}$ |
| Per-Subject (Self) | $0.194 \pm 0.234$ | $0.234 \pm 0.267$ | $p = 8.25 \times 10^{-52}$ |
| LOO Fine-Tune | $0.219 \pm 0.235$ | $0.347 \pm 0.259$ | $p = 4.48 \times 10^{-89}$ |
| Per-Subject (Zero-Shot) | $0.159 \pm 0.200$ | $0.171 \pm 0.216$ | $p = 8.48 \times 10^{-42}$ |
| LOO Zero-Shot | $0.333 \pm 0.318$ | $0.400 \pm 0.326$ | $p = 3.05 \times 10^{-48}$ |
| LOO Drop | $0.037 \pm 0.064$ | $0.047 \pm 0.052$ | $p = 6.86 \times 10^{-10}$ |

## G.3 NN MODELS COMPARISON

Per-task and per-evaluation scheme results are shown in Figure G.13. The results highlight that classic NN decoders remain highly competitive, particularly when compared to linear-probed foundation models. Foundation model performance exhibits substantial variability across both downstream tasks and evaluation protocols. While fine-tuned foundation models often achieve top performance in population and LOO Fine-Tune settings, they tend to underperform compared to classic NNs in Per-Subject (self) and Per-Subject (transfer) evaluations.

To enable a comprehensive comparison of model performance across datasets, evaluation protocols, and subjects, we computed model ranks separately for each metric of interest. For each combination of dataset, evaluation protocol, and subject, models completing the same experiment were ranked according to their metric value. Ranking was performed independently for each metric. In cases of ties, we applied a competition ranking scheme ("1, 1, 3, . . . "), where models with identical scores received the same (lowest) rank, and the next model was ranked as if the previous positions were occupied. This approach accommodates ties and missing values, enabling robust aggregation and statistical comparison across heterogeneous experimental settings.

We report results for all six evaluation protocols in Figure G.14 (within-subject: models have access to the test subject during training) and Figure G.15 (cross-subject: models do not see the test subject during training, plus LOO Drop). Aggregated results for within-subject and cross-subject evaluations are presented in Figure G.16.

Based on the results in Figure G.14, Figure G.15, and Figure G.16, several important patterns emerge. While the fine-tuned large ST-EEGFormer achieves the highest overall performance, its advantage over the second-best model, CTNet, is often not statistically significant—particularly in Per-Subject (self) and LOO Fine-Tune protocols. In fact, foundation models of comparable size (under 30M parameters) generally do not outperform compact classic NN decoders such as CTNet (a few million parameters) or even EEGNet (a few thousand parameters).

The clearest advantage for foundation models emerges in the LOO Drop protocol, suggesting a stronger ability to retain and leverage previously seen examples. Our results also indicate that foundation model performance may be biased toward the characteristics of their pre-training datasets. Many existing EEG foundation models are pre-trained on large-scale clinical EEG datasets (Obeid & Picone, 2016), potentially favoring tasks such as abnormal EEG detection while providing limited benefits for more diverse BCI paradigms. In contrast, our proposed ST-EEGFormer—pre-trained exclusively on BCI datasets—achieves higher average performance across tasks. Nevertheless, its small and base variants still underperform compared to simple classic NN models in several settings, underscoring the need for further investigation into model architecture, pre-training data diversity, and task alignment in EEG foundation model development.

Finally, our statistical analysis underscores an important point: conclusions based solely on mean accuracy, as is common in prior work, can be misleading. Rigorous statistical testing often reveals that differences between top models are not significant, challenging the narrative of superiority. Together, these findings expose current limitations in EEG foundation models and emphasize the necessity of fair, statistically-grounded benchmarking to drive meaningful progress.

## G.4 BEST MODEL COMPARISON

For each downstream task, we compare the accuracy distributions across subjects for the best-performing model in each of the four decoder groups: classic non-NN, classic NN, linear-probed foundation models, and fine-tuned foundation models. The results are shown in Figures G.17 to G.24

In summary, the statistical test results reveal that, for many downstream tasks, differences in mean performance between the top models from different EEG decoder types do not translate into statistically significant differences. Substantial variability is observed across datasets, underscoring that model performance is heavily task-dependent. Nevertheless, certain trends emerge: classic non-NN models consistently underperform compared to NN-based approaches in population and transfer settings, yet remain competitive in per-subject (self) evaluations.

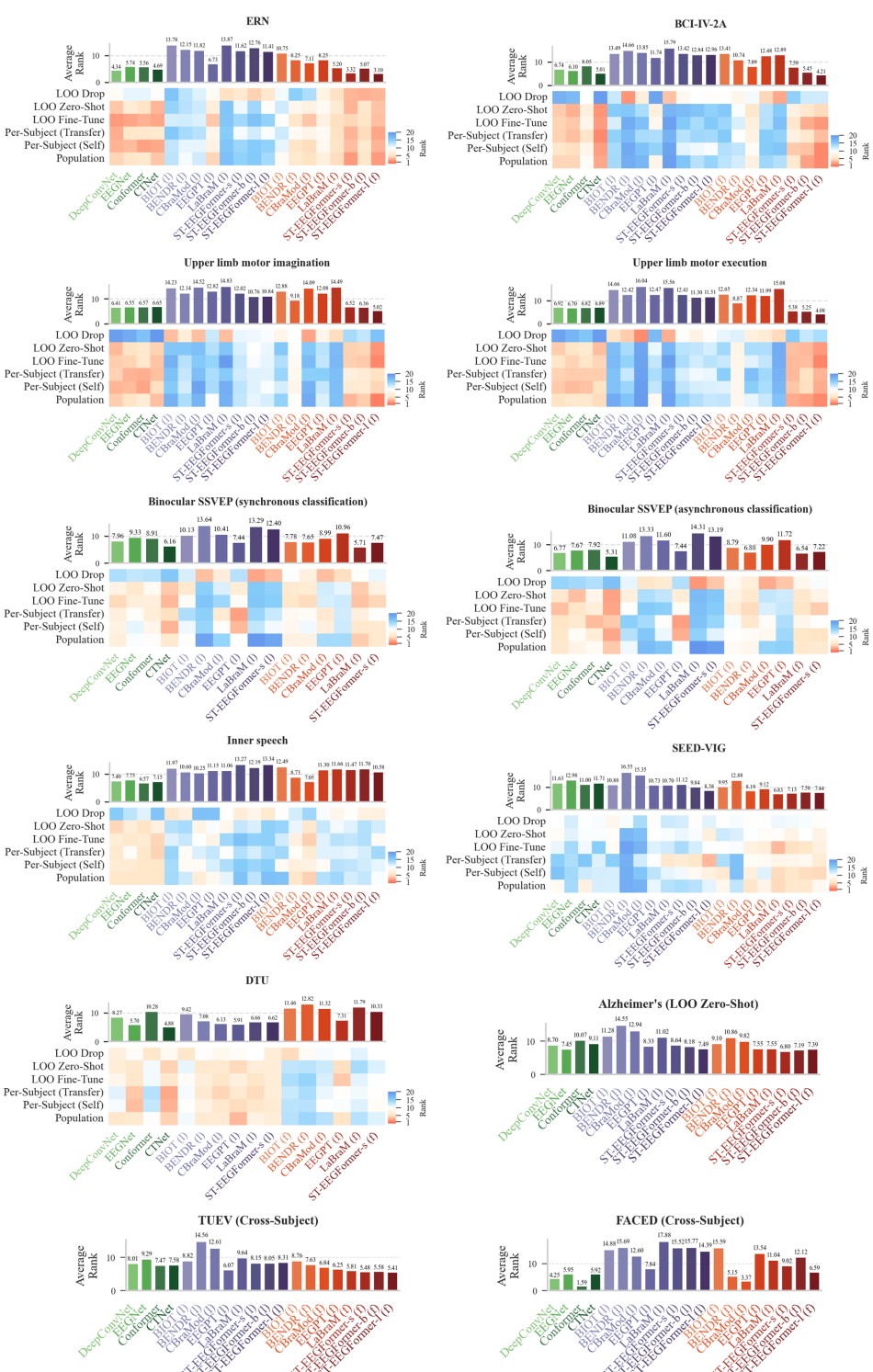

Figure G.13: Comparison of classic NN models (green) with linear-probed foundation models (purple) and fine-tuned foundation models (red) across six evaluation protocols on all benchmarked downstream tasks. The bar plot depicts the average aggregated rank for each model (lower rank indicates better performance). The heatmap shows the average aggregated rank per evaluation protocol.

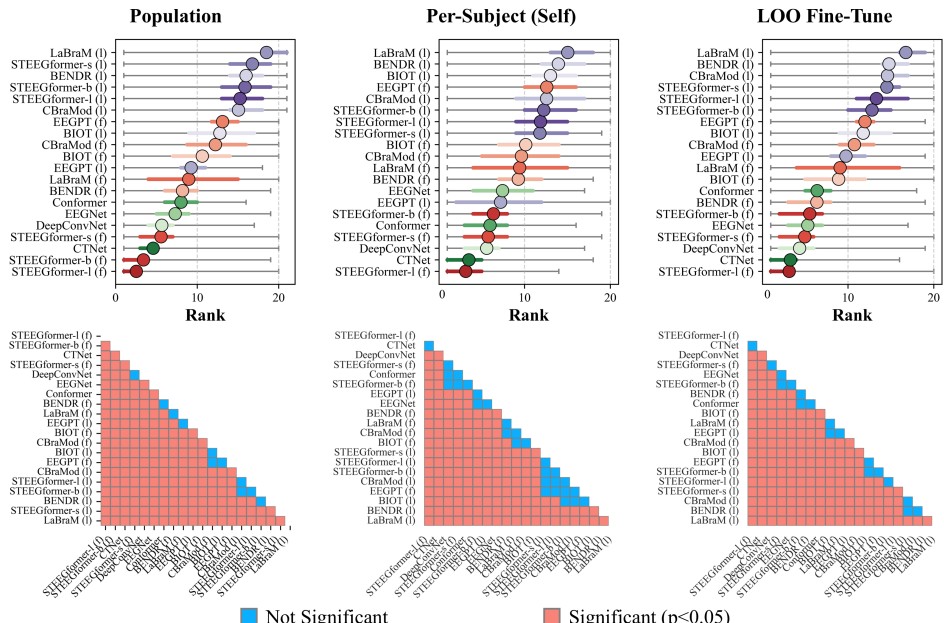

Figure G.14: Comparison of classic NN models (green) with linear-probed foundation models (purple) and fine-tuned foundation models (red) under three within-subject evaluation protocols: Population, Per-Subject (self), and LOO Fine-Tune. Top row: Model rank distributions, ordered from best (bottom) to worst (top). Circles indicate mean rank, horizontal bars represent the interquartile range (25th–75th percentiles), and whiskers denote the minimum and maximum ranks. Bottom row: Corresponding pairwise statistical significance matrices from 50,000-run permutation tests with Bonferroni correction. Red cells indicate significant differences ($p < 0.05$), and blue cells indicate no significant difference. Model order (top–down, left–right) matches the ranking plots above.

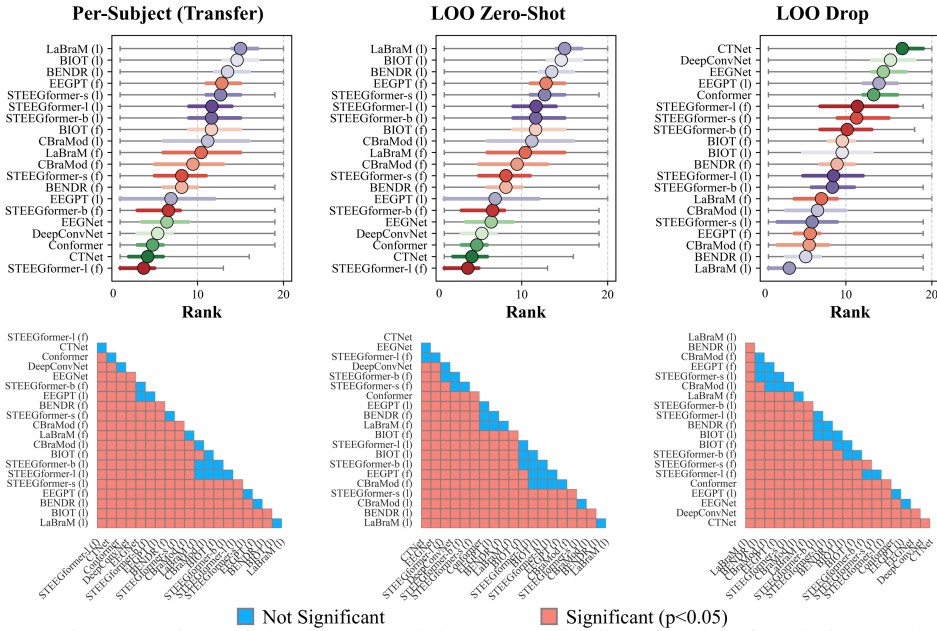

Figure G.15: Comparison of classic NN models (green) with linear-probed foundation models (purple) and fine-tuned foundation models (red) under two cross-subject evaluation protocols—Per-Subject (transfer) and LOO Zero-Shot—as well as the LOO Drop protocol. The notation and panel layout follow those in Figure G.14.

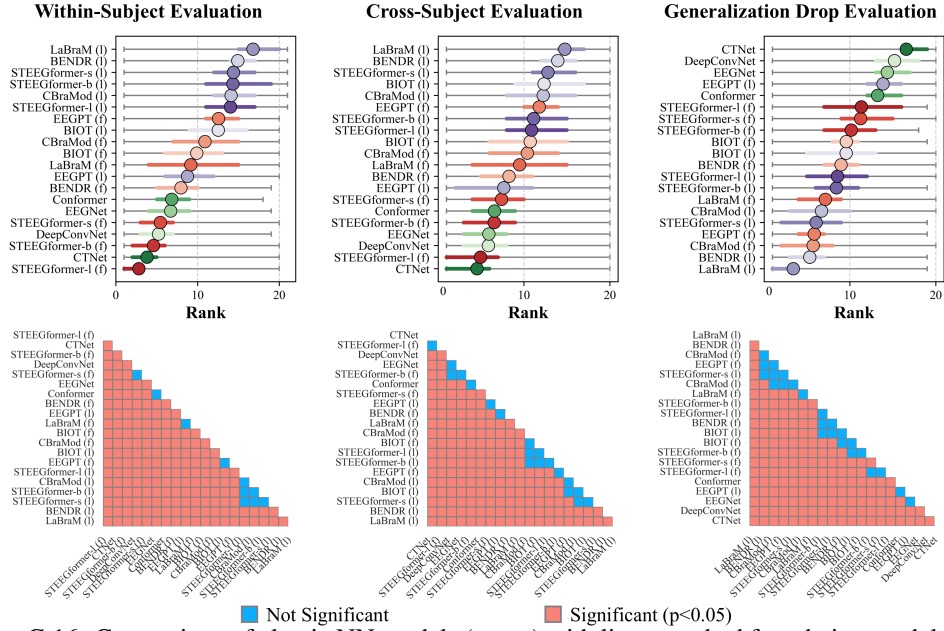

Figure G.16: Comparison of classic NN models (green) with linear-probed foundation models (purple) and fine-tuned foundation models (red) under two aggregated evaluation settings plus the LOO Drop protocol: (i) within-subject, combining Population, Per-Subject (self), and LOO Fine-Tune protocols; and (ii) cross-subject, combining Per-Subject (transfer), LOO Zero-Shot protocols. The notation and panel layout follow those in Figure G.14.

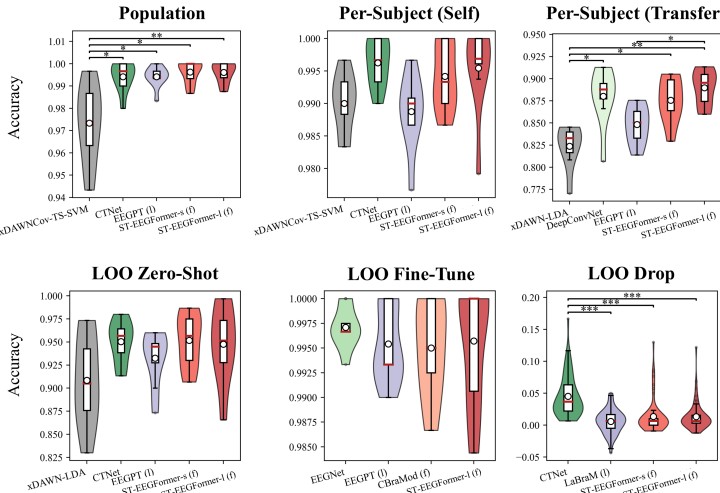

Figure G.17: Subject-wise accuracy distributions on the ERN dataset for the best-performing model from each decoder group: classic non-NN models (blue), classic NN models (green), linear-probed foundation models (purple), and fine-tuned foundation models (red). The notation and panel layout follow those in Figure 4.

## G.5 ATTENTION MAPS COMPARISON

We plot the attention maps on the BCIC-IV-2A dataset of BIOT, LabraM and EEGPT models. Each map is categorized based on the adaptation method (linear probing or fine-tuning) and the input data (left-hand MI or right-hand MI). Therefore, each model has a total of four attention maps. For visualization, we first average the attention scores across all heads and temporal positions. Following the attention rollout approach described below Abnar & Zuidema (2020), we then aggregate the attention scores across all layers and project them onto the topographic head map for visualization. The results are shown in Figures G.25 to G.36.

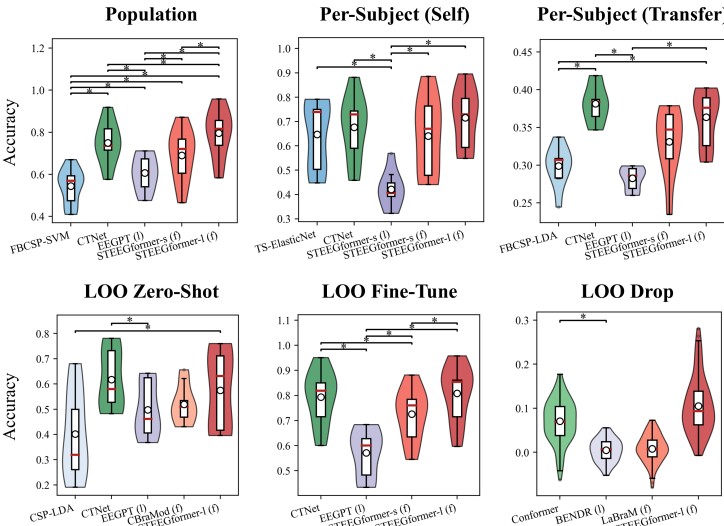

Figure G.18: Subject-wise accuracy distributions on the BCI-IV-2A dataset for the best-performing model from each decoder group: classic non-NN models (blue), classic NN models (green), linear-probed foundation models (purple), and fine-tuned foundation models (red). The notation and panel layout follow those in Figure 4.

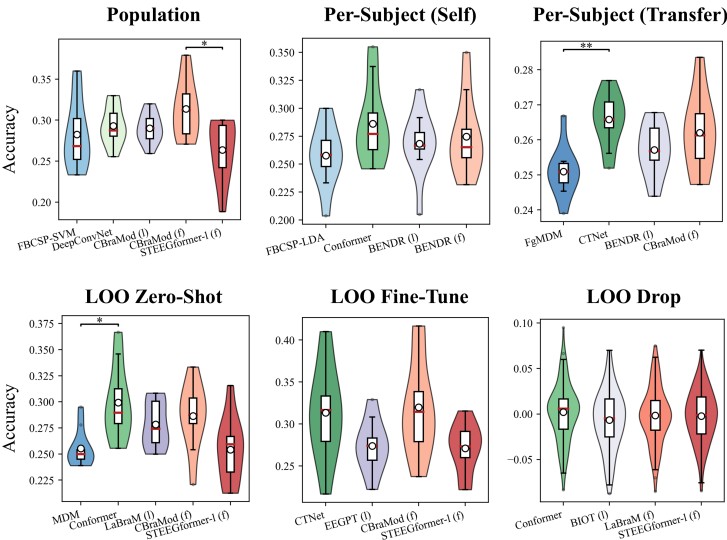

Figure G.19: Subject-wise accuracy distributions on the Inner speech dataset for the best-performing model from each decoder group: classic non-NN models (blue), classic NN models (green), linear-probed foundation models (purple), and fine-tuned foundation models (red). The notation and panel layout follow those in Figure 4.

### G.5.1 ATTENTION ROLLOUT

Attention rollout tracks the information flow from the input layer to the final layer in a transformer model through Eq G.1 (Abnar & Zuidema, 2020), where, $\tilde{A}$ is the attention rollout, and $A(l_i)$ the raw attention matrix in layer $i$. In order to focus on the most important tokens while ignoring less relevant ones, we apply a discard ratio that retains only the largest rollout weights at each layer. For instance, a discard ratio of 0.9 will keep only the top 10% of the largest weights, setting the remaining weights to zero. After calculating the rollout, each head produces a weight matrix. The final weights are obtained by fusing the weights across different heads, using one of the following methods: mean fusion, where the final weight is the average of all head weights; max fusion, where the final weight is the maximum value across all heads; and min fusion, where the final weight is the minimum value across all heads.

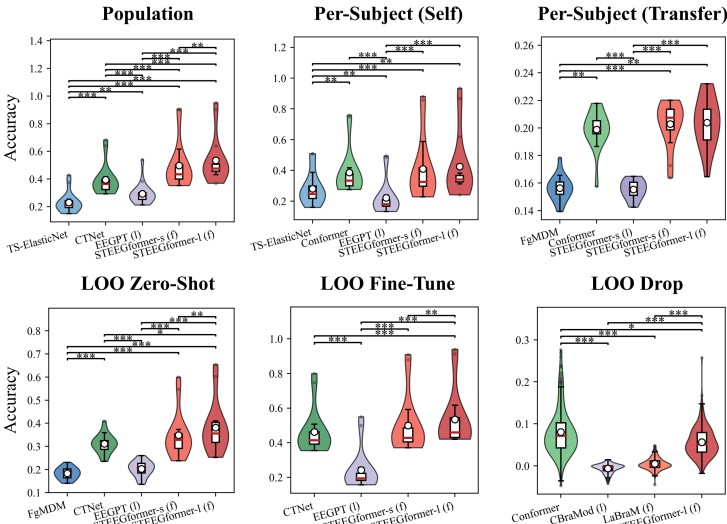

Figure G.20: Subject-wise accuracy distributions on the Upper limb motor execution dataset for the best-performing model from each decoder group: classic non-NN models (blue), classic NN models (green), linear-probed foundation models (purple), and fine-tuned foundation models (red). The notation and panel layout follow those in Figure 4.

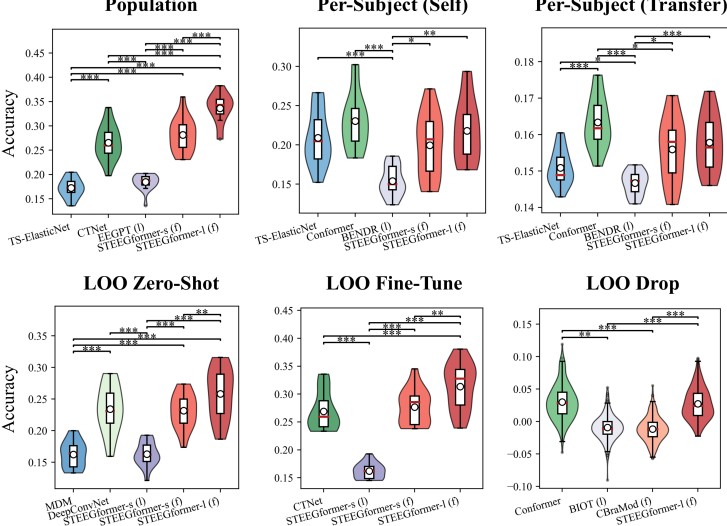

Figure G.21: Subject-wise accuracy distributions on the Upper limb motor imagination dataset for the best-performing model from each decoder group: classic non-NN models (blue), classic NN models (green), linear-probed foundation models (purple), and fine-tuned foundation models (red). The notation and panel layout follow those in Figure 4.

$$\tilde{A}(l_i) = \begin{cases} (A(l_i) + I)\tilde{A}(l_{i-1}) & \text{if } i > 1 \\ A(l_i) + I & \text{if } i = 1 \end{cases} \tag{G.1}$$

### G.5.2 DISCUSSION

In summary, our analysis yields the following observations:

1. **Shifted attention after fine-tuning.** With the exception of EEGPT, most foundation models attend to different regions after fine-tuning on downstream classification tasks. This suggests that representations learned during pre-training are not fully aligned with task-specific ones.

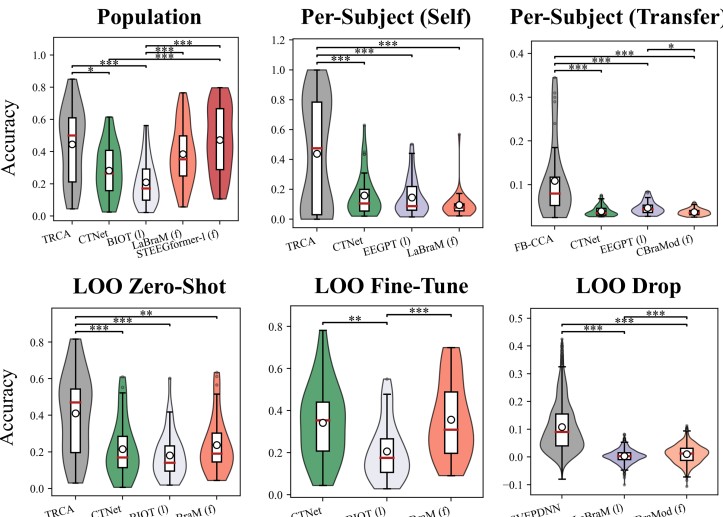

Figure G.22: Subject-wise accuracy distributions on the Binocular SSVEP dataset (synchronous classification) for the best-performing model from each decoder group: classic non-NN models (blue), classic NN models (green), linear-probed foundation models (purple), and fine-tuned foundation models (red). The notation and panel layout follow those in Figure 4.

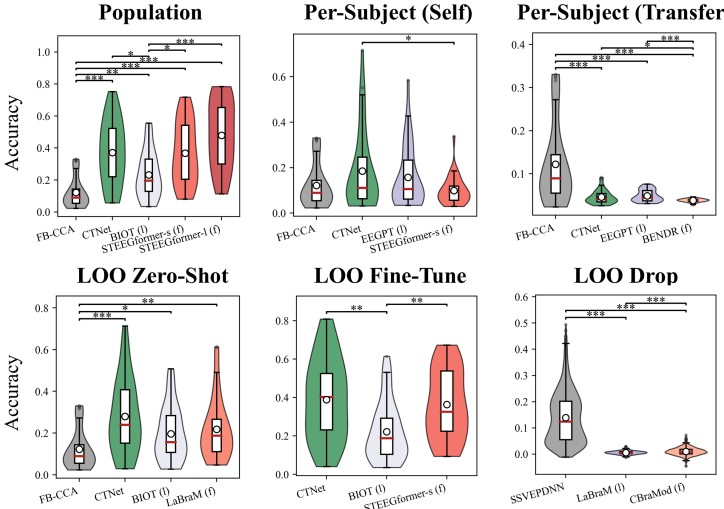

Figure G.23: Subject-wise accuracy distributions on the Binocular SSVEP dataset (asynchronous classification) for the best-performing model from each decoder group: classic non-NN models (blue), classic NN models (green), linear-probed foundation models (purple), and fine-tuned foundation models (red). The notation and panel layout follow those in Figure 4.

2. **Consistent spatial patterns across models.** Despite architectural differences, all models exhibit symmetric attention patterns, with increased focus on regions near the motor cortex after fine-tuning.

3. **Limited distinction between left- and right-hand MI.** Attention maps for left- and right-hand motor imagery are highly similar across models, indicating that key discriminative features may lie beyond the coarse spatial patterns captured by attention rollout.

**LOO Zero-Shot**

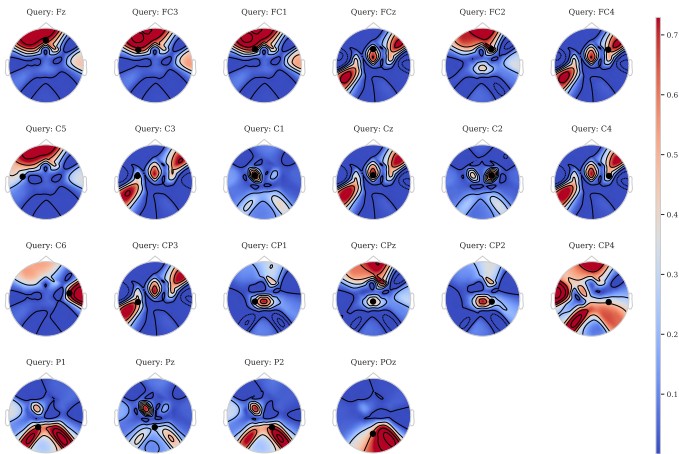

Figure G.24: Subject-wise accuracy distributions on the Alzheimer's dataset for the best-performing model from each decoder group: classic non-NN models (blue), classic NN models (green), linear-probed foundation models (purple), and fine-tuned foundation models (red). The notation and panel layout follow those in Figure 4.

Figure G.25: Attention topographical visualization for the LabraM model using linear probing on left-hand motor imagery data from the BCI-IV-2A dataset. The topographic map shows attention scores averaged across all heads and temporal positions, with all layers aggregated.

## H  ADDITIONAL EXPERIMENTS

### H.1  TOKEN FUSION AND CLASSIFICATION HEAD STRATEGIES

We performed an ablation on BCI-IV-2A and ERN datasets using three foundation models—CBraMod, EEGPT, and our ST-EEGFormer-small (abbrev. ST-EEGFormer-s)—across five evaluation protocols, including LOO Zero-Shot, LOO Fine-Tune, LOO Drop, Per-Subject (Self), and Per-Subject (Transfer). For each model, we compared two variants that differ in (i) how token features are fused and (ii) the final classification head (Figure H.1 and Figure H.2):

- **Simple** Average token pooling followed by a single linear classifier.

- **Complex** For CBraMod and EEGPT, use the model's default classification head; For ST-EEGFormer-s, use a full-token head with two linear layers.

Across both datasets and all five protocols, the *Complex* design yields better performance in most settings—often with statistical significance—especially under linear probing. These findings suggest that classification head capacity and token fusion choices are important to downstream performance. In the case of linear-probed foundation models, relying solely on average token fusion may discard valuable spatiotemporal information that a more expressive classification head can leverage. We recommend future work to systematically explore head architectures for EEG foundation models to unlock further gains.

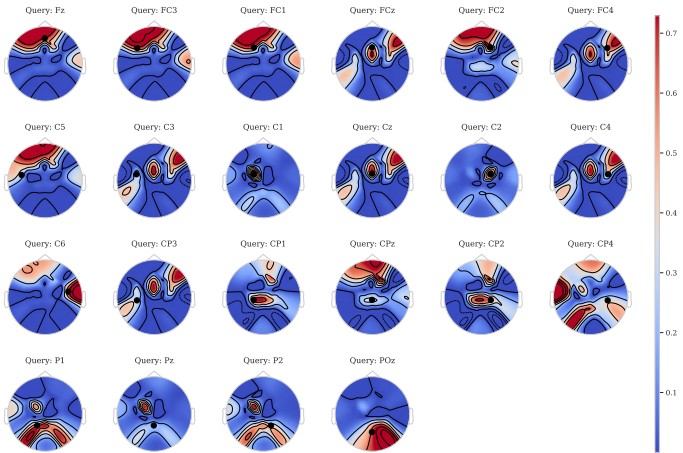

Figure G.26: Attention topographical visualization for the LabraM model using linear probing on right-hand motor imagery data from the BCI-IV-2A dataset. The topographic map shows attention scores averaged across all heads and temporal positions, with all layers aggregated.

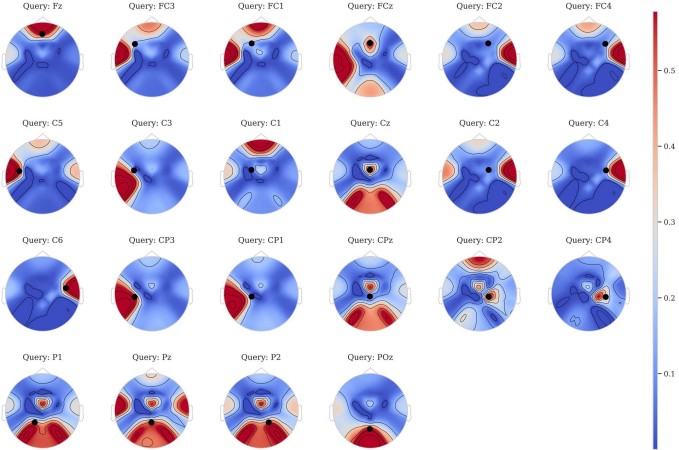

Figure G.27: Attention topographical visualization for the LabraM model using fine-tuning on left-hand motor imagery data from the BCI-IV-2A dataset. The topographic map shows attention scores averaged across all heads and temporal positions, with all layers aggregated.

## H.2 EFFECT OF TRAINING-SET SIZE

We varied the proportion of labeled training data from 20% to 80% on BCI-IV-2A and ERN, benchmarking all neural decoders under the five evaluation protocols. Results are shown in Figure H.3 and Figure H.4.

**Key observations**

1. **Monotonic gains with more data.** Across models and protocols, performance generally increases as the training fraction grows.

2. **Linear probing is consistently weakest.** Linear-probed foundation models are the worst performers across training ratios and protocols.

3. **Strong competitiveness of classic NNs.** Classic neural decoders remain highly competitive—often statistically better than foundation-model variants when labeled data are scarce. Aside from the LOO Drop protocol, classic decoders also tend to exhibit larger performance drops after fine-tuning.

4. **Best model comparison.** When comparing the top model from each decoder group (panel b), classic decoders frequently have a higher mean, but differences from fine-tuned foundation mod-

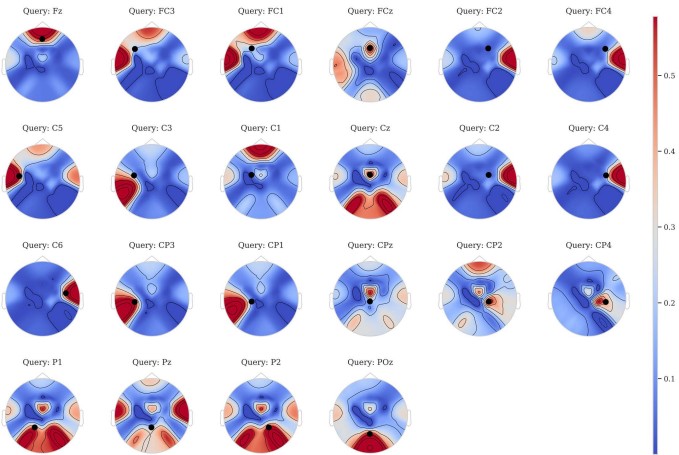

Figure G.28: Attention topographical visualization for the LabraM model using fine-tuning on right-hand motor imagery data from the BCI-IV-2A dataset. The topographic map shows attention scores averaged across all heads and temporal positions, with all layers aggregated.

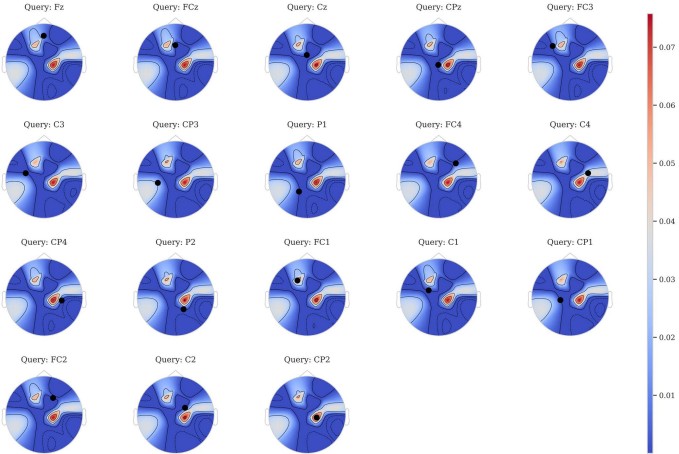

Figure G.29: Attention topographical visualization for the BIOT model using linear probing on left-hand motor imagery data from the BCI-IV-2A dataset. The topographic map shows attention scores averaged across all heads and temporal positions, with all layers aggregated.

els are typically not statistically significant. In low-data regimes, the best fine-tuned foundation models sometimes attain a higher mean, again without a significant difference relative to classic decoders.

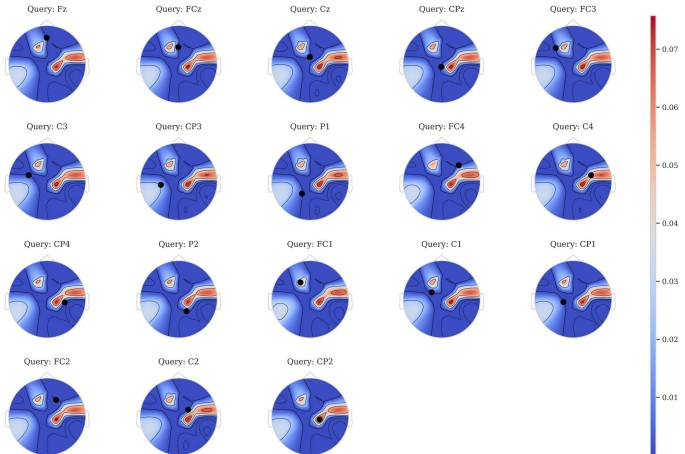

Figure G.30: Attention topographical visualization for the BIOT model using linear probing on right-hand motor imagery data from the BCI-IV-2A dataset. The topographic map shows attention scores averaged across all heads and temporal positions, with all layers aggregated.

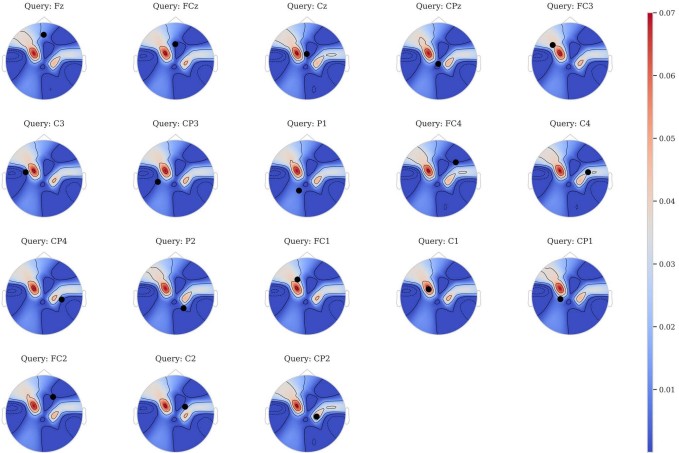

Figure G.31: Attention topographical visualization for the BIOT model using fine-tuning on left-hand motor imagery data from the BCI-IV-2A dataset. The topographic map shows attention scores averaged across all heads and temporal positions, with all layers aggregated.

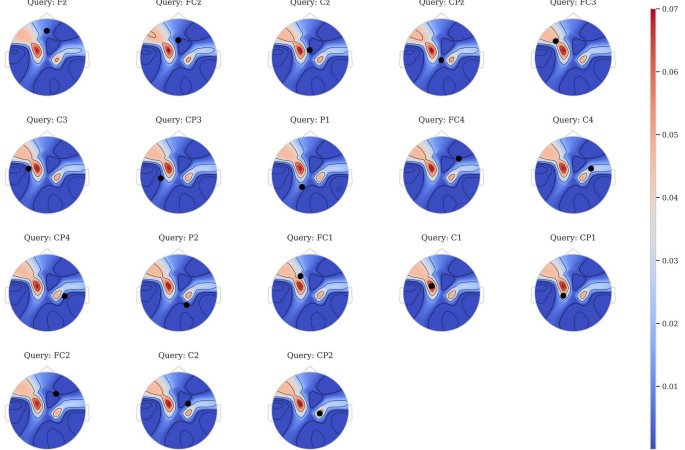

Figure G.32: Attention topographical visualization for the BIOT model using fine-tuning on right-hand motor imagery data from the BCI-IV-2A dataset. The topographic map shows attention scores averaged across all heads and temporal positions, with all layers aggregated.

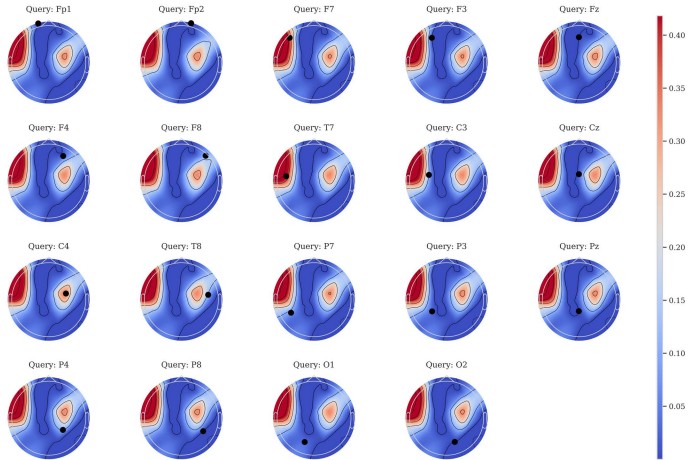

Figure G.33: Attention topographical visualization for the EEGPT model using linear probing on left-hand motor imagery data from the BCI-IV-2A dataset. The topographic map shows attention scores averaged across all heads and temporal positions, with all layers aggregated.

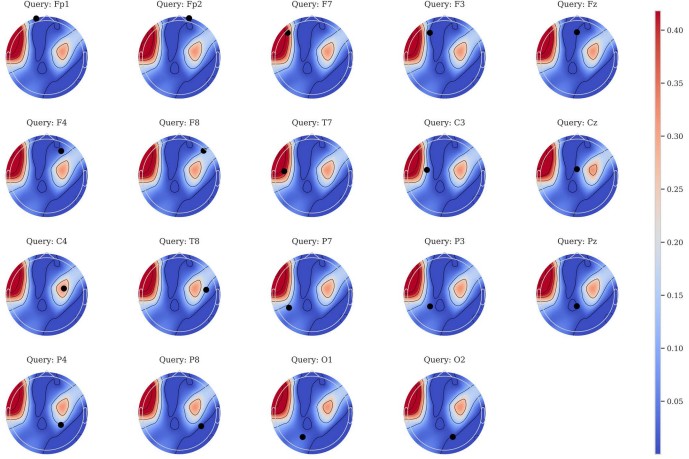

Figure G.34: Attention topographical visualization for the EEGPT model using linear probing on right-hand motor imagery data from the BCI-IV-2A dataset. The topographic map shows attention scores averaged across all heads and temporal positions, with all layers aggregated.

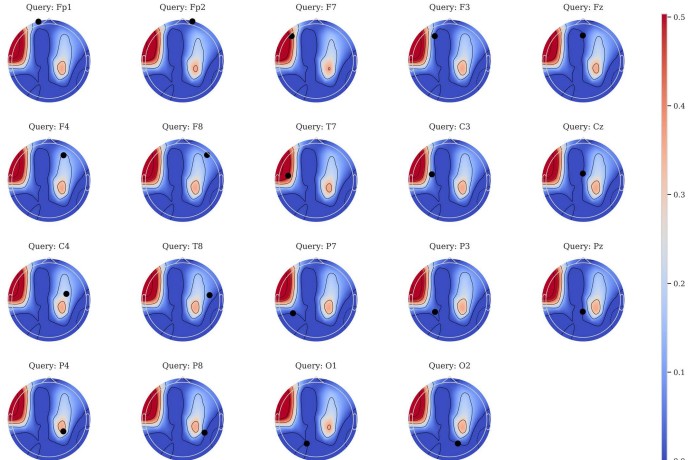

Figure G.35: Attention topographical visualization for the EEGPT model using fine-tuning on left-hand motor imagery data from the BCI-IV-2A dataset. The topographic map shows attention scores averaged across all heads and temporal positions, with all layers aggregated.

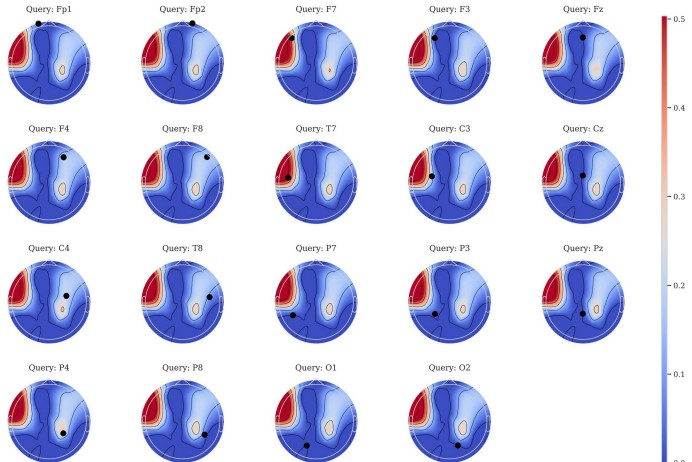

Figure G.36: Attention topographical visualization for the EEGPT model using fine-tuning on right-hand motor imagery data from the BCI-IV-2A dataset. The topographic map shows attention scores averaged across all heads and temporal positions, with all layers aggregated.

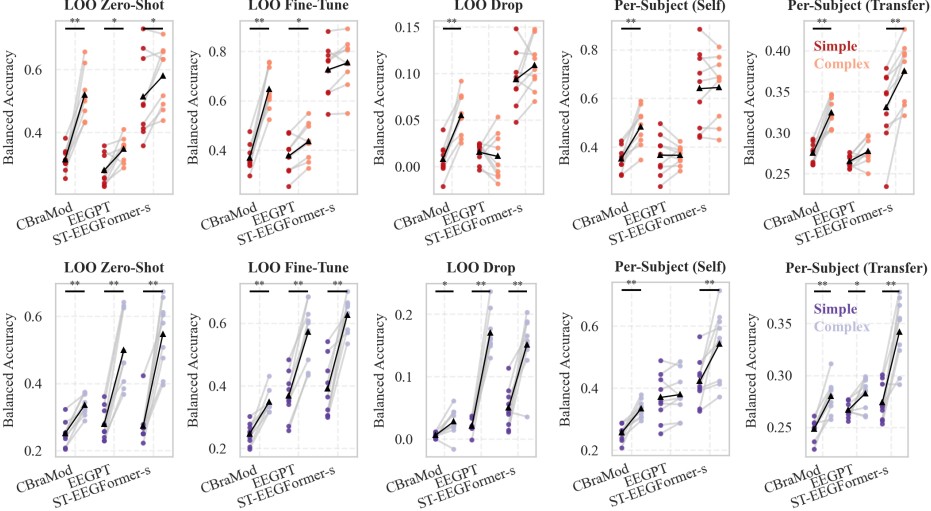

Figure H.1: BCI-IV-2A ablation of token fusion and classification head strategies for CBraMod, EEGPT, and ST-EEGFormer-s. *Simple* (red and purple): average-token pooling + linear head. *Complex* (light red and purple): default head for EEGPT and CBraMod; for ST-EEGFormer-s, a full-token two-layer head. Top row: fine-tuning; bottom row: linear probing. Wilcoxon signed-rank test: ***: $p < 0.001$, **: $p < 0.01$, *: $p < 0.05$.

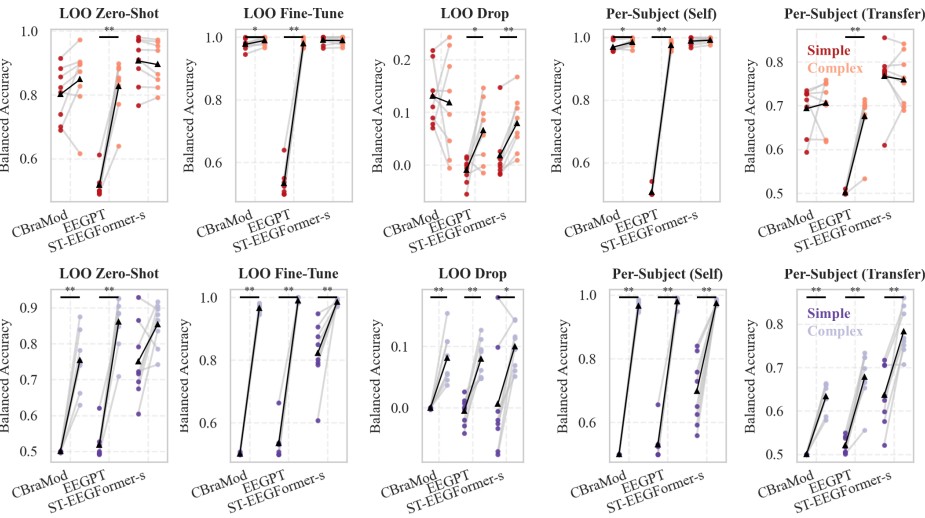

Figure H.2: ERN ablation of token fusion and classification head strategies for CBraMod, EEGPT, and ST-EEGFormer-s. *Simple* (red and purple): average-token pooling + linear head. *Complex* (light red and purple): default head for EEGPT and CBraMod; for ST-EEGFormer-s, a full-token two-layer head. Top row: fine-tuning; bottom row: linear probing. Wilcoxon signed-rank test: ***: $p < 0.001$, **: $p < 0.01$, *: $p < 0.05$.

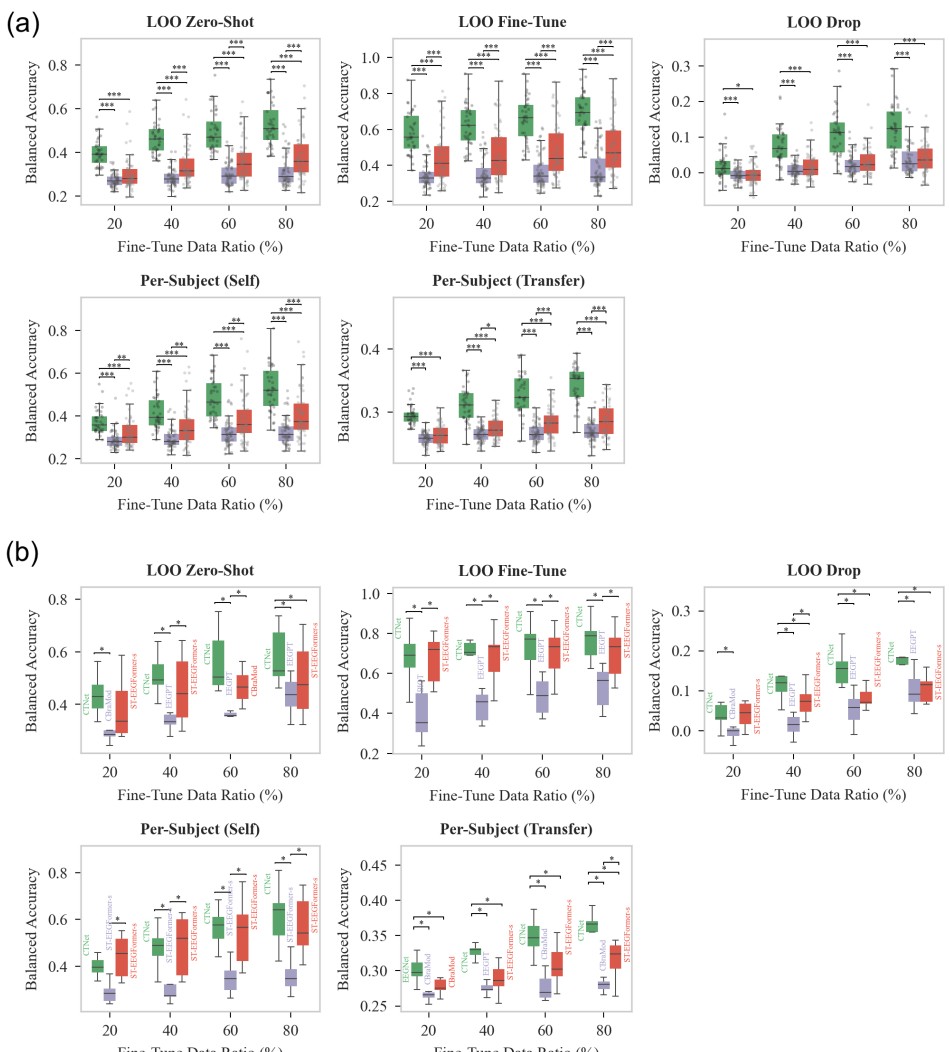

Figure H.3: BCI-IV-2A results when varying the training fraction from 20% to 80% under five evaluation protocols. **(a)** Group-level comparison: classic neural decoders (green), linear-probed foundation models (purple), and fine-tuned foundation models (red). Boxplots reflect balanced accuracy over all test subjects. **(b)** Best-performing model from each group (model names annotated). Wilcoxon signed-rank test with Bonferroni correction: ***: $p < 0.001$, **: $p < 0.01$, *: $p < 0.05$.

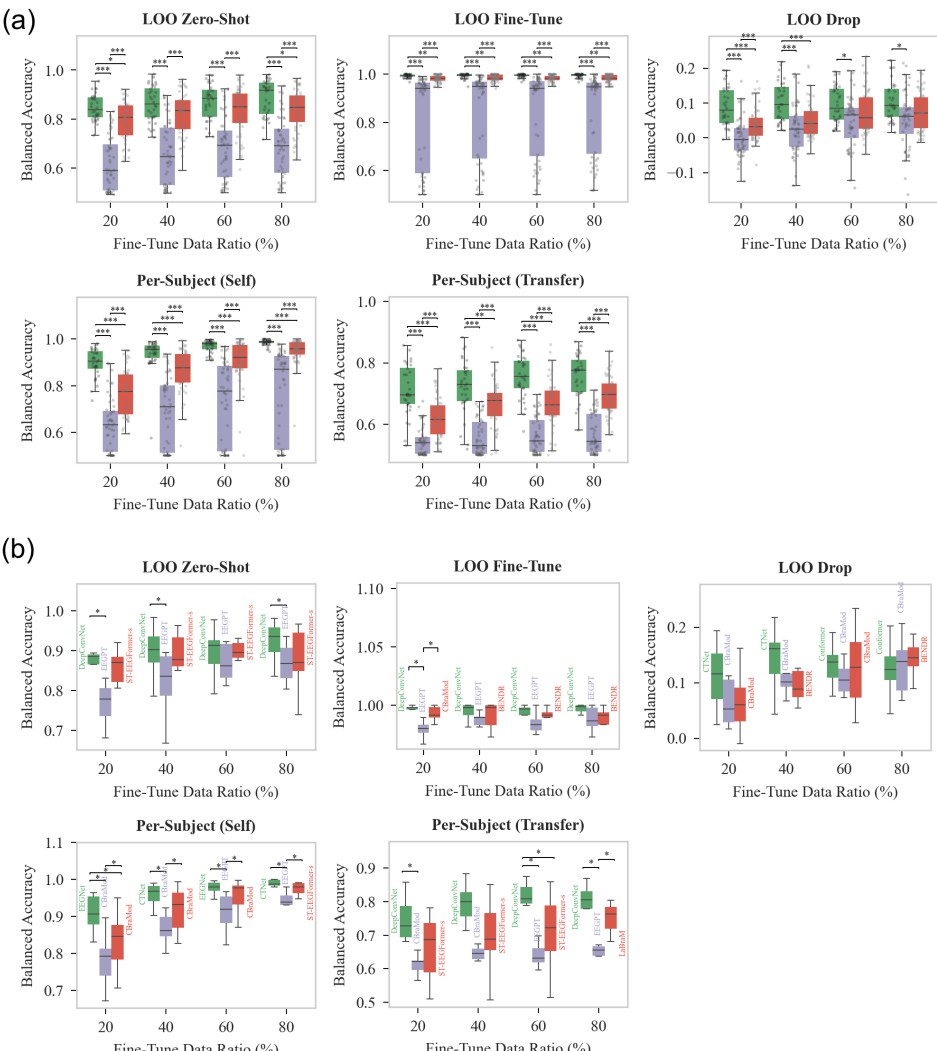

Figure H.4: ERN results when varying the training fraction from 20% to 80% under five evaluation protocols. (a) Group-level comparison: classic neural decoders (green), linear-probed foundation models (purple), and fine-tuned foundation models (red). Boxplots reflect balanced accuracy over all test subjects. (b) Best-performing model from each group (model names annotated). Wilcoxon signed-rank test with Bonferroni correction: ***: $p < 0.001$, **: $p < 0.01$, *: $p < 0.05$.

