# OpenReview forum: "Are EEG Foundation Models Worth It? Comparative Evaluation with Traditional Decoders in Diverse BCI Tasks"
_ICLR.cc/2026/Conference — ICLR 2026 Poster_

### Official Review · Reviewer_gwxQ · 2025-10-28

**Soundness:** 3
**Presentation:** 4
**Contribution:** 3
**Rating:** 4
**Confidence:** 5

**Summary:**

The paper introduces a benchmark to compare EEG foundation models and baseline deep neural networks. It also introduces a ViT based model that is used to compare the other foundation models with.

**Strengths:**

The paper is very well organised. The authors provide a plethora of experiments, comparisons as well as ablation studies. These experiments range from various models, experimental setups (e.g. how the various ways one can treat training subjects) and classification / regression tasks. All of these make the analysis meticulously thorough.

**Weaknesses:**

1. Although very thorough, I fail to see the novelty behind the paper. For example, a recent work [1], although it covers less in experimental results, it comes across the exact same observations / conclusions. One could argue that this newest work extends [1] but its insights do not seem novel.
2. The main message of the paper is also complicated. Is it a benchmarking paper or a new model (in this case ST-EEGFormer) introduction ?
3. The authors challenge LaBraM’s claim that direct MAE in EEG signals is ineffective but they fail to provide any evidence against it. For example, a comparison between reconstruction capabilities of LaBraM’s tokenizer and ST-EEGFormer would be useful here.

Writing:
The paper is well-written.

Overall:
The paper is clear and with insightful observations / results but the main message or its novelty need to be clearly decided.

[1]: Na Lee, Konstantinos Barmpas, Yannis Panagakis, Dimitrios Adamos, Nikolaos Laskaris, & Stefanos Zafeiriou (2025). Are Large Brainwave Foundation Models Capable Yet ? Insights from Fine-Tuning. In Forty-second International Conference on Machine Learning.

**Questions:**

1. What is the main purpose of the paper and if it’s just benchmarking what’s the novelty compared to [1] ?
2. How about the reconstruction ? Any examples here.
3. An interesting addition to distinguish yourself from [1] would be an analysis of the number of training samples during deep baselines training and foundation model fine tuning. For instance, if currently 100 samples (all) are used for training of both models and the performance is similar how about if you use 10-20-40-60-80 random samples (over multiple sampling) for deep baselines training and foundation model fine tuning. Do the foundation models have any advantage there ?

[1]: Na Lee, Konstantinos Barmpas, Yannis Panagakis, Dimitrios Adamos, Nikolaos Laskaris, & Stefanos Zafeiriou (2025). Are Large Brainwave Foundation Models Capable Yet ? Insights from Fine-Tuning. In Forty-second International Conference on Machine Learning.

---

> ### Author Response · Authors · 2025-11-24
> **Response to Reviewer gwxQ-part1**
>
> # Response to Reviewer gwxQ
>
> We sincerely thank Reviewer gwxQ for the thoughtful feedback. We are grateful for your recognition of the thoroughness, organization, and clarity of our study, and we appreciate your constructive suggestions. Below, we provide detailed responses and summarize the substantial updates made to the manuscript that directly address your concerns.
>
> ---
>
> ### 1. **Clarifying the Main Purpose of the Paper (Weakness 1, 2 & Question 1)**
> Our work is intentionally both a comprehensive benchmarking study and a proposal of a transparent, reproducible foundation model baseline. We clarify our positioning along three aspects:
>
> **1) A dense, systematic benchmarking study**
>    As detailed in our overall response, the paper is built around an extensive benchmarking effort, covering:
>    - Multiple evaluation protocols (including population, per-subject, and leave-one-out schemes).
>    - A broad spectrum of model families (foundation models, compact neural models, and classical non-neural decoders).
>    - A diverse set of downstream datasets and tasks (9 classification and 2 regression tasks).
>
>
> **2) Why we propose a new foundation model (ST-EEGFormer)**
>    During this work, we observed that many recent EEG foundation models adopt some variant of MAE-based ViT architectures, but there is a lack of a simple, fully open, ViT-style MAE baseline that the community can reliably build on and compare with. As summarized in our overall response, several prior works claim to release large variants, but many of these models or checkpoints are still **not publicly available**, limiting reproducibility and fair comparison.
>
> We thus introduce ST-EEGFormer specifically to fill this gap: a plain, scalable, MAE-pretrained ViT-style model whose code and weights are designed to be fully open-sourced after review, serving as a strong but transparent reference point.
>
> **3) Beyond benchmarking: addressing EEG research field concerns**
>    Our objective is not only to report benchmark numbers, but also to answer:
>    > Are EEG foundation models 'worth it' relative to classic and compact models when evaluated fairly across tasks and protocols?
>
>    This framing explicitly distinguishes our study from, for example, Lee et al., 2025, which focuses mainly on fine-tuning insights from two limited foundation models across five tasks and a single evaluation protocol. In contrast, our work spans **more models**, covers **more tasks and modalities** , and uses **multiple evaluation protocols with statistical testing** to provide a broader and more principled view of when (or on which cases) EEG foundation models actually provide added value.
>
> In summary, the paper should be read as a **benchmark-driven investigation** of EEG foundation models, supported by a **simple, open, and scalable baseline (ST-EEGFormer)** that is introduced to address current gaps in transparency and reproducibility.
>
>
>
> ---
>
> (to be continued...)

---

> > ### Author Response · Authors · 2025-11-24
> > **Response to Reviewer gwxQ-part2**
> >
> > ---
> >
> > ### 2. **Comparison with Prior Work (Lee et al., 2025) (Weakness 1 & Question 1)**
> > Thank you for bringing the work by Lee et al. to our attention; we agree it is a valuable study in BCI field. We first would like to highlight that our study and the work by Lee et al. should be considered **concurrent publications**, as their manuscript appeared on arXiv within three months of our initial ICLR submission.
> >
> > Furthermore, our work provides a distinct contribution compared to their study in the following critical aspects:
> >
> > - **Scope and Diversity** Our study incorporates a significantly broader and more up-to-date scope of models, including a larger selection of recent foundation models, classic neural networks and machine learning models. We also evaluate performance across more diverse evaluation protocols and both classification and regression downstream tasks, leading to more robust and generalizable results.
> >
> > - **Statistical Rigor** Crucially, our findings are supported by rigorous statistical analysis performed on a comprehensive set of decoders. This level of statistical validation provides greater reliability and validation for our reported performance differences, a key element often missing in related benchmarking efforts.
> >
> > - **Unique Observations** In our study, the large fine-tuned ST-EEGFormer achieves significantly superior performance on certain tasks, such as ERN, where it outperforms both other foundation models and classic neural networks (with statistical significance). We also identify specific regimes where foundation models clearly hold an advantage over other decoders—for example, under the LOO Drop protocol and in data-rich population decoding (Fig. 4a), where fine-tuned foundation models achieve statistically better performance than classic neural baselines. Moreover, in the revised manuscript, additional ablation experiments on token fusion strategies and classification head design (Appendix Figures H.1 and H.2) shed light on previously overlooked design choices and point to promising directions for future model improvements.
> >
> > | Study             | Foundation Models | Classic Models | Evaluation Protocols | Downstream Tasks | Statistical Analysis |
> > |------------------|---------------------|----------------|----------------------|------------------|----------------------|
> > | Lee et al. (2025)| 2                   | 2              | 1                    | 5 (classification only) | ❌ |
> > | **This Study**   | 6 (including ours)  | 22             | 6                    | 9 (classification) + 2 (regression) | ✅ |
> >
> >
> > ---
> >
> > ### 3. **On LaBraM’s Claim About Direct MAE on Raw EEG (Weakness 3 & Question 2)**
> > In LaBraM, the authors state (p.5): *“In our previous experiments, the loss fails to converge while directly reconstructing raw EEG signals.”* On the contrary, in this work we have shown that **performing direct MAE on raw EEG signals is feasible**.
> >
> > Concretely, we provide two forms of evidence in the revised manuscript:
> >
> > **1) Training convergence**
> >    In Appendix Figure E.2, we show that when pretraining ST-EEGFormer with a plain MAE objective directly on raw EEG, the **reconstruction loss converges stably** over training. After 100 epochs, all models training-validation loss converged to a stable value below 0.2. This directly contradicts the notion that such a setup cannot converge.
> >
> > **2) Reconstruction quality**
> >    In Appendix Figures E.3 and E.4, we present **qualitative reconstruction examples** (original vs. reconstructed waveforms). These visualizations demonstrate that the model can recover meaningful temporal structure and amplitude patterns in the raw signals using this simple MAE formulation.
> >
> > ---
> > (to be continued...)

---

> > > ### Author Response · Authors · 2025-11-24
> > > **Response to Reviewer gwxQ-part3**
> > >
> > > ---
> > >
> > > ### 4. **Analysis of Training Sample Size (Few-shot) (Question 3)**
> > > We fully agree with the importance of this question. In the revised manuscript, we conducted a new experiment (Figures H.3–H.4 in Appendix) that examines how model performance changes as the available training data is varied. Specifically, we trained both foundation models and baseline neural networks using **20%, 40%, 60%, and 80%** of the training data under five evaluation protocols: Per-Subject (Self), Per-Subject (Transfer), LOO Zero-Shot, LOO Fine-Tune, and LOO Drop.
> > >
> > > **Key Findings:**
> > >
> > > - **Performance generally improves with more labeled data**  Across models and protocols, accuracy exhibits a mostly monotonic increase as training data grows.
> > >
> > > - **Linear probing remains the weakest setting**  Linear-probed foundation models consistently perform worst across all data fractions and protocols, reinforcing that frozen representations are insufficient for EEG decoding.
> > >
> > > - **Classic neural decoders are highly competitive, especially in low-data regimes**  When labeled samples are scarce, compact baseline models frequently match or exceed foundation models. Additionally, for the LOO Drop protocol, classic decoders tend to exhibit larger performance drops after fine-tuning, suggesting stronger subject-dependence.
> > >
> > > - **Comparison of best-in-class models**  When comparing the best model from each family (panel b), classic decoders often show a slightly higher mean performance, though differences relative to fine-tuned foundation models are typically not statistically significant. In low-data regimes, certain fine-tuned foundation models exhibit a slightly higher mean—but again without significant separation relative to classic NN baselines.
> > >
> > > Overall, this experiment shows that while foundation models benefit from larger training sets, **their advantage is not universal**, and compact neural models remain highly effective and sometimes preferable in practical scenarios where labeled EEG data are limited.
> > >
> > >
> > >
> > > ---
> > >
> > >
> > > We deeply appreciate your thoughtful feedback, which helped us improve the clarity, scope, and rigor of our paper. We hope the added analyses and clarifications address your concerns and demonstrate the value of our contribution as a benchmark and reference point for the EEG foundation model community.

---

> ### Comment · Reviewer_gwxQ · 2025-11-24
> **Response to the reviewers**
>
> I would like to thank the reviewers for their analysis and responses. Unfortunately, although a very detailed analysis, (as highlighted by another reviewer) I don't think the work is novel. Two other works [1] & [2] are mentioned that cover the exact same topics. Although this one has more tasks and models the conclusions align closely with the other two works - which makes it not novel for the research community.
>
>
> [1] Wu, Jiamin, et al. "Adabrain-bench: Benchmarking brain foundation models for brain-computer interface applications." arXiv preprint arXiv:2507.09882 (2025).
> [2] Lee, Na, et al. "Are Large Brainwave Foundation Models Capable Yet? Insights from Fine-tuning.", ICML 2025

---

> > ### Author Response · Authors · 2025-11-24
> > **Response to Reviewer gwxQ**
> >
> > We thank the reviewer for acknowledging the thoroughness of our benchmark. However, we respectfully disagree with the claim that our work lacks novelty and does not contribute to the field, and would like to clarify several points:
> >
> > ### 1. Clarification on Concurrent Work Policy
> >
> > Based on ICLR’s official policy (https://iclr.cc/Conferences/2025/FAQ), papers are considered **concurrent** if they were published at a peer-reviewed venue within four months of the submission deadline. Our paper was submitted in **September 2025**, meaning any work published after **May 2025** is classified as contemporaneous.
> >
> > - **Lee et al. (2025)** was submitted to arXiv in **July 2025**. Even though it may have undergone peer review earlier (in May), it should still be considered concurrent with our submission.
> > - **Wu et al. (2025)** is also an arXiv preprint submitted in **July 2025**, and, to the best of our knowledge, remains **unpublished (not peer-reviewed)** at the time of this response.
> >
> > Therefore, under ICLR’s guidelines, we find it inappropriate to judge our work's novelty based on papers that are either **not peer-reviewed** or **concurrent at submission time**.
> >
> > ### 2. Our Unique Contributions and Insights
> >
> > We believe we do offer novel and useful contributions:
> >
> > - **Inclusion of classical non-neural decoders.**
> >   To our knowledge, we are the first to systematically evaluate whether traditional non-neural models (often overlooked in FM literature) remain competitive. While their average performance is lower, our results show that several well-designed, task-specific methods are still strong contenders in certain regimes. Our benchmark thus provides a practical reference for researchers to identify which classical models remain useful for their own applications.
> >
> > - **Identification of token fusion effects.**
> >   We show that token fusion strategies (e.g., [CLS], averaging, flattening) significantly affect downstream performance in foundation models. This implementation detail is rarely discussed in prior work, yet we demonstrate its substantial impact, especially under linear probing.
> >
> > - **Statistical rigor and protocol diversity.**
> >   Unlike many existing studies that report only mean scores, we apply formal statistical testing across all experiments and use multiple evaluation protocols. This enables us to show that EEG foundation models significantly outperform baselines in certain regimes (e.g., **population-level decoding**, **LOO-Drop**), a conclusion that prior benchmarks did not establish under rigorous hypothesis testing.
> >
> > - **Fully open-source large-scale models.**
> >   While several EEG foundation model papers claim to be open source, in practice most do not release complete weights and training code, especially for their largest variants. In contrast, **ST-EEGFormer** is designed to be **fully open-source** across **small, base, and large** variants, directly addressing a major **reproducibility gap** in the community.
> >
> > We hope this clarification helps contextualize the **novelty and value** of our work.

---

### Official Review · Reviewer_ABR4 · 2025-10-28

**Soundness:** 2
**Presentation:** 3
**Contribution:** 2
**Rating:** 2
**Confidence:** 5

**Summary:**

This paper offers a systematic, large-scale comparison of EEG foundation models against classic neural and non-neural decoders across multiple datasets, tasks, and six evaluation protocols. This paper also introduces a baseline, ST-EEGFormer (a ViT trained with MAE on ~8M raw EEG segments). This work systematically summarizes how different types of models perform under various evaluation protocols and draws a series of performance-related conclusions.

**Strengths:**

The paper delivers a systematic, comprehensive analysis spanning EEG foundation models, neural network–based approaches, and classical baselines. The empirical workload is substantial and reflects significant effort. It also provides a clear synthesis with thorough descriptions of existing models, and the manuscript is well written and highly readable.Its calls to the community are measured and well-judged.

**Weaknesses:**

Despite the substantial effort and extensive scope, the work lacks originality. As a benchmarking study, it largely aggregates and uniformly evaluates existing open source methods rather than introducing genuinely new ideas. Although the paper proposes ST-EEGFormer, the architecture amounts to a straightforward Transformer combined with MAE-style pretraining—a paradigm already widely used in EEG self-supervised learning (e.g., EEGPT [1], CBraMod [2], EEG2Rep [3]). Similarly, the spatio-temporal Transformer concept has ample precedent (e.g., CBraMod [2] , Brant-2 [4]). In addition, the manuscript contains several inappropriate or insufficiently rigorous elements (see Questions for details). Taken together, I do not consider this submission a good fit for ICLR; it reads more like an engineering report than a research contribution. Moreover, comparable benchmarking studies already exist such as Adabrain-bench [5], which is open source , so this is not the first of its kind.

[1] Wang, Guangyu, et al. "Eegpt: Pretrained transformer for universal and reliable representation of eeg signals." Advances in Neural Information Processing Systems 37 (2024): 39249-39280.

[2] Wang, Jiquan, et al. "Cbramod: A criss-cross brain foundation model for eeg decoding." arXiv preprint arXiv:2412.07236 (2024).

[3] Mohammadi Foumani, Navid, et al. "Eeg2rep: enhancing self-supervised eeg representation through informative masked inputs." Proceedings of the 30th ACM SIGKDD Conference on Knowledge Discovery and Data Mining. 2024.

[4] Yuan, Zhizhang, et al. "Brant-2: Foundation model for brain signals." CoRR (2024).

[5] Wu, Jiamin, et al. "Adabrain-bench: Benchmarking brain foundation models for brain-computer interface applications." arXiv preprint arXiv:2507.09882 (2025).

**Questions:**

1. The six “evaluation protocols” described in this work do not always align with commonly used paradigms or terminology in EEG research—for example, the notions of “population” and the specific zero-shot configurations. To my knowledge, widely adopted paradigms include subject-dependent, subject-independent, cross-subject, cross-session, and cross-dataset. Approximately, your (2) within-subject seems to correspond to subject-dependent; your (4) leave-one-subject-out zero-shot to subject-independent; and your (3) per-subject zero-shot to cross-subject. However, other mainstream settings I mentioned appear to be missing, and some of your configurations—such as (5) and (6)—do not have clear counterparts in established experimental practice. Moreover, the terminology for all protocols is presented without supporting citations. I recommend clarifying the provenance and rationale of each protocol, aligning your naming with standard conventions (or explicitly motivating any deviations), and adding appropriate references.

2. The claim that this work is the “first comprehensive benchmark” does not seem justified; comparable efforts already exist—for example, Adabrain-bench [5].

3. The pretraining paradigm and architecture of ST-EEGFormer are not novel, as closely related designs already exist, thus this model lacks originality.

4. Moreover, while ST-EEGFormer-s outperforms works like LaBraM and BENDR overall—and ST-EEGFormer-b and -l achieve further gains—I suspect these improvements appear largely attributable to substantially larger model sizes. For example, ST-EEGFormer-s has 32.7M parameters compared with only 5.8M for LaBraM-base, and ST-EEGFormer-b and -l scale up to 110.9M and 328.4M parameters, respectively. This also raises my next question.

5. In Section 4.5 the authors write: “although there is a slight upward trend in normalized accuracy with increasing model size, the poor logarithmic fit suggests that a clear scaling law does not exist for downstream EEG classification tasks.” I believe this analysis is problematic. The comparison mixes models of different sizes and different architectures rather than holding the architecture fixed, yet architectural choices are known to have a large impact on EEG performance; prior classic neural-network baselines for EEG have clearly demonstrated the sensitivity to model design. As a result, the paper’s conclusion about the absence of a scaling law is insufficiently rigorous. Moreover, within the authors’ own ST-EEGFormer family, a clearer scaling trend appears: moving from the small model (32.7M) to the base (110.9M) and then to the large (328.4M) yields consistent performance gains. This, if anything, suggests the presence of a scaling law and contradicts the stated conclusion.

6. The figures are difficult to read—especially the radar charts, which include too many elements to distinguish clearly. While I appreciate the substantial scope of the work, the visual presentation should be improved for clarity.

7. If positioned as a benchmarking suite, the current coverage of models and datasets is relatively limited and leaves ample room for improvement. Mainstream EEG tasks—such as seizure detection, emotion recognition, and mental workload detection—should be included, and additional foundation models (e.g., EEG2Rep [3]) as well as classic deep-learning baselines (e.g., LGGNet [6]) should be incorporated.

In sum, I agree the authors’ call to build a more cohesive and standardized EEG community; however, this advocacy cannot substitute for substantive novelty and technical contributions in the present work.

[6] Ding, Yi, et al. "LGGNet: Learning from local-global-graph representations for brain–computer interface." IEEE Transactions on Neural Networks and Learning Systems 35.7 (2023): 9773-9786.

---

> ### Author Response · Authors · 2025-11-24
> **Response to Reviewer ABR4-part1**
>
> # Response to Reviewer ABR4
>
> We sincerely thank the reviewer for the time and thoughtful comments. We respectfully address each concern below and provide clarifications for several misunderstandings.
>
> ---
>
> ### 1. On Novelty and Research Contribution (Weakness + Question 2, 3)
>
> Our work has two main contributions:
> (1) A comprehensive, statistically grounded benchmarking study
> (2) A transparent, reproducible foundation model baseline for the community
>
> As outlined in our overall response (*1. Motivation*), we identified a key open question in the current literature:
>
> > Do EEG foundation models truly and consistently outperform classic neural and non-neural models with statistical significance in 2025?
>
> Our study is explicitly designed around this question. Through dense experiments and rigorous cross-task comparisons, we distinguish our work from prior efforts, including the concurrent **Adabrain-bench** (Wu et al., 2025) mentioned by the reviewer. We also note that Adabrain-bench was released less than two months before our submission and has not yet been peer-reviewed. More importantly, our benchmark offers:
>
> - broader coverage of model families (including classic non-neural decoders),
> - more evaluation protocols, and
> - a wider range of downstream tasks.
>
> A comparative summary is provided in the first table of our overall response.
>
> Regarding model architecture and training methods, we emphasize that our aim is not to introduce a radically new foundation-model design, but rather to provide a simple, scalable, and fully reproducible baseline:
>
> - Many recent EEG foundation models use MAE-based ViT variants, yet there is **no widely adopted, fully open ViT+MAE baseline** that the community can reliably use and compare against.
> - As summarized in the second table of our overall response, several prior works **claim** to release large variants, but many models remain **unavailable** (e.g., **EEG2Rep**, as recommended by the reviewer).
> - Similarly, for **Brant-2**, the public GitHub repository (https://github.com/yzz673/Brant-2) is currently empty (no code or weights), which limits its utility for reproducible comparison.
>
> **ST-EEGFormer** is introduced specifically to fill this reproducibility gap: it is a **plain ViT with MAE pretraining**, designed to be **fully open-sourced** after review and to serve as a **strong yet transparent reference**. Combined with our broad benchmark (including non-neural network baselines), this positions our work not just as a new novel foundation model paper, but as a **field-level examination** of when EEG foundation models are actually *worth it* relative to smaller, more traditional decoders.
>
>
> ---

---

> > ### Author Response · Authors · 2025-11-24
> > **Response to Reviewer ABR4-part2**
> >
> > ---
> >
> > ### 2. On Evaluation Protocols and Terminology (Question 1)
> > We acknowledge that adopting community-standard terminology enhances readability and prevents potential confusion. However, we contend that the widely used terms, such as subject-dependent and subject-independent, are too general to accurately and practically differentiate between BCI application scenarios and training methodologies.
> >
> > **Limitations of Traditional Terms**
> >
> > According to the definition of cross-subject transfer learninig in BCI field (Wu et al., 2020), the source domain and the target domain contains different composition of subjects. The term of subject-independent model means when a model is trained with subjects' data from source domain and apply to the target subject. However, it fails to explicitly differentiate the case when a model is trained with both source subject's data and target subject's data between the case when a model is exclusively trained with source subjects' data.
> >
> > **Our Proposed Naming System**
> >
> > To achieve greater precision that reflects practical BCI development, our naming system is grounded in the principle of model training and application methodology:
> > | Training Data Source             | Our Terminology | Description | Link to Traditional Use |
> > |------------------|---------------------|----------------|----------------------|
> > | Group of subjects' data| Population model                   | Trained on pooled data from multiple subjects.              | Closely relates to subject-independent model                   |
> > | Only target subject's data   | Per-subject model  | Trained exclusively on one target subject's data.             | Closely relates to subject-dependent model                   |
> >
> > Based on this principle, we can precisely define the application scenarios:
> > - **Population** (protocol 1): The model's training process comprises data from all subjects.
> >
> > - **LOO Zero-Shot** (protocol 4): The model is trained only on source subjects' data and then applied directly to the target subject (without target data in training). This corresponds to a zero-shot generalization scenario.
> >
> > - **LOO Fine-Tune** (protocol 5): A highly practical BCI scenario where the model is pre-trained on source subjects and then fine-tuned with a small amount of the target subject's data for better personalization and generalization.
> >
> > - **LOO Drop** (protocol 6): Derived from LOO Fine-Tune, this configuration tests the model's capability when the personalized model is applied to other subjects, measuring how much its subject-specific tuning degrades performance.
> >
> > We have provided a detailed explanation of these naming conventions and their linkage to the standard subject-dependent and subject-specific terms in Section 3.1 of the manuscript to guide readers.
> >
> > * Wu, Dongrui, Yifan Xu, and Bao-Liang Lu. "Transfer learning for EEG-based brain–computer interfaces: A review of progress made since 2016." IEEE Transactions on Cognitive and Developmental Systems 14.1 (2020): 4-19.
> >
> > ---
> >
> > (to be continued...)

---

> ### Author Response · Authors · 2025-11-24
> **Response to Reviewer ABR4-part3**
>
> ---
>
> ### 3. On Scaling Law Analysis (Question 4 & 5)
>
> We thank the reviewer for this thoughtful critique and agree that our original wording in Section 4.5 could be interpreted as overstating the “absence” of a scaling law. We clarify and refine our position as follows:
>
>  **Scaling within the same model architecture**
>   We acknowledge that within our own ST-EEGFormer family, increasing model size from small (32.7M) → base (110.9M) → large (328.4M) often yields improved downstream performance, which is compatible with a scaling trend. Our intention was not to deny this pattern, but to highlight that even the largest, best-performing ST-EEGFormer variant does not uniformly dominate smaller neural decoders across all tasks and protocols.
>
>  **Lack of open sourced large foundation models**
>   As mentioned in our earlier response, we did not intentionally use small size models, however, other foundation models were either not open sourced (like EEG2Remp, and Brant-2) or only partially open sourced (like LabraM, and EEGPT). In practice, the community currently has access to only a limited set of large, fully reproducible models. We propose ST-EEGFormer to help fill this gap with a **simple, transparent, ViT-MAE–style baseline**.
>
>  **Our primary question: scale vs. task dependence**
>   Prior foundation model work has already shown that more pretraining data (or tokens) or a larger model size within the same architecture tends to improve performance (e.g., CBraMod and LaBraM). Instead of re-demonstrating these known scaling curves, our paper addresses a different question:
>   > Do large EEG foundation models actually outperform smaller neural and non-neural decoders across diverse tasks and evaluation protocols?
>
>   Our results show that **they do not universally dominate**—for example, on tasks such as inner speech and DTU auditory decoding, large foundation models do not show statistically significant advantages over compact baselines. This underscores that EEG decoding performance is strongly task dependent, and cannot be understood purely through model scale.
>
> ---
>
> ### 4. On Visual Clarity (Question 6)
>
> In the revised version:
> - We replaced the previous radar plot in Figure 3 with a more straightforward heatmap + bar plot combination.
> - The bar plot now summarizes the average rank each model achieves across all protocols.
> - The heatmap provides the corresponding per-protocol rank details in a more readable format.
> - For per-dataset performance, we refer readers to Appendix Figure G.13, which reports results at the dataset level.
>
> We believe these changes make the comparative performance of different models easier to interpret at a glance, while still providing detailed information in the appendix.
>
> ---
>
> ### 5. On Benchmark Scope (Question 7)
>
> We made substantial updates to broaden the benchmark in response to this concern:
>
> - **Added 3 new datasets** to increase task and domain coverage:
>   - **SEED-VIG** – vigilance prediction (**regression**)
>   - **FACED** – **emotion recognition**
>   - **TUEV** – **clinical event classification**
>
> Regarding **benchmarked models**, as summarized in the first table of our overall response, our study already benchmark one of the largest models to date, spanning non-neural decoders, classic neural networks and EEG foundation models.
>
> We acknowledge that task-specialized architectures—for example, LGGNet for emotion or fatigue decoding—may achieve even higher performance on their specific datasets. However, incorporating every task-specific SOTA model would not alter our central conclusion:
>
> > Compact, task-agnostic neural networks remain highly competitive with EEG foundation models across diverse decoding tasks.
>
> Our focus is intentionally on task-agnostic models applicable across many datasets and evaluation protocols, which is essential for assessing whether EEG foundation models are truly “worth it” as broadly applicable solutions. In our latest manuscript, we have introduced those task-specific NN models, like LGGNet,  in our related work section in Appendix C.2.
>
>
> ---
>
>
> We respectfully disagree that the work “reads more like a report.” It addresses clear, novel research questions with methodological rigor, extensive statistical evaluation, and actionable insights that challenge prevailing assumptions in the EEG foundation model literature.
>
> While we understand the reviewer’s high standards for technical novelty, we emphasize that **reproducibility, clarity, breadth of evaluation, and statistical rigor are equally vital scientific contributions**, especially in a fast-moving and sometimes fragmented field.
>
> We hope this detailed clarification helps contextualize the contribution of our work for the AC and other reviewers.

---

### Official Review · Reviewer_Jg9y · 2025-10-30

**Soundness:** 3
**Presentation:** 4
**Contribution:** 4
**Rating:** 8
**Confidence:** 5

**Summary:**

This paper presents a comprehensive benchmark of EEG foundation models against traditional neural and non-neural decoders across diverse BCI tasks and evaluation protocols. The authors introduce ST-EEGFormer, a Vision Transformer-based model pre-trained with masked autoencoding on over 8 million EEG segments. The study finds that while fine-tuned foundation models perform well in data-rich, population-level settings, they often do not significantly outperform simpler neural networks or classical methods in data-scarce or subject-specific scenarios. Linear probing consistently underperforms, and no clear scaling law is observed. The work emphasizes the need for large-scale datasets and rigorous, reproducible benchmarking.

**Strengths:**

*   **Comprehensive and Rigorous Benchmarking:** The paper provides an exceptionally thorough evaluation, comparing a wide range of EEG foundation models against both classic neural networks and non-neural baselines across six diverse tasks and six distinct evaluation protocols, supported by extensive statistical testing.
*   **Introduction of a Strong, Transparent Baseline:** The proposed ST-EEGFormer model serves as a valuable and reproducible baseline, effectively demonstrating that a simple MAE pre-training strategy on raw EEG can achieve competitive performance, challenging assumptions from prior work.
*   **Critical and Actionable Findings:** The study delivers nuanced, evidence-based conclusions that question the universal superiority of foundation models, highlighting their limitations in data-scarce settings and the weak performance of linear probing, which provides crucial guidance for future research directions in the field.

**Weaknesses:**

**Limited Scope of Downstream Task Evaluation**: The selection of seven downstream datasets, while diverse, may not be fully representative of the broad spectrum of EEG applications. The benchmark omits several common and challenging tasks such as emotion recognition (e.g., FACED), sleep staging, and seizure or depression detection, which are frequently used to evaluate foundation models. Furthermore, the chosen tasks like ERP and SSVEP often have simple, stereotypical patterns that can be decoded effectively by classical non-neural methods, potentially skewing the conclusion that foundation models offer limited advantages. The low overlap with the evaluation benchmarks from existing foundation model papers also raises concerns about the generalizability of the findings across the field's common evaluation practices.

**Concerns over Data Transparency for Community Reference**: The heavy reliance on distribution plots (e.g., violin plots, box plots) to present aggregated results, while useful for visualizing statistical distributions, limits the transparency of the raw results. This makes it difficult for the community to directly reference specific performance numbers (e.g., exact accuracy on a specific dataset) or to perform alternative meta-analyses. Providing detailed numerical results in supplementary tables for each model-dataset-protocol combination would enhance reproducibility and utility for future comparative studies.

**Questions:**

1.  **Data Transparency:** The extensive use of distribution plots provides a excellent overview of model performance, but makes it difficult to extract precise numerical results for community reference and future benchmarking. Could the authors provide a supplementary file or table with the detailed, per-dataset numerical results (e.g., accuracy, MSE, Pearson R) for each model under the different evaluation protocols? This would greatly enhance the reproducibility and long-term utility of this valuable benchmark.

2.  **Ablation on Token Fusion Strategy:** The paper identifies the token fusion strategy (e.g., flatten-all vs. average) as a critical hidden implementation factor. Given its potential impact, have the authors conducted a systematic comparison of how different fusion strategies affect the performance of various foundation models (like EEGPT and CBraMod) across different types of downstream tasks (e.g., motor imagery vs. emotion recognition)? Such an ablation study would be highly insightful for the community to understand the interaction between model architecture, fusion strategy, and task characteristics.

---

> ### Author Response · Authors · 2025-11-24
> **Response to Reviewer Jg9y**
>
> # Response to Reviewer Jg9y
>
> We sincerely thank Reviewer Jg9y for the thoughtful and encouraging feedback. We greatly appreciate the recognition of our comprehensive benchmarking, the value of ST-EEGFormer as a transparent baseline, and the critical observations on reproducibility and generalizability. Below, we address your insightful comments point by point.
>
> ---
>
> ### **1. Scope of Downstream Tasks (Weakness 1)**
>
>  Thank you for this important observation. We have expanded the benchmark by adding **three additional datasets** that cover underrepresented and practically important EEG application domains:
>
> - **SEED-VIG** – A regression task for vigilance prediction.
> - **FACED** – A large-scale emotion recognition benchmark.
> - **TUEV** – A clinical EEG event classification dataset.
>
> These additions extend the total task coverage to **11 downstream datasets**, including both classification and regression, and offer broader representation across affective, cognitive, and clinical domains. This significantly strengthens the benchmark and better reflects the full utility spectrum of EEG foundation models.
>
> ---
>
> ### **2. Transparency of Numerical Results (Weakness 2 & Question 1)**
>
> We fully agree with the importance of providing complete results for transparency and reproducibility. We have now included all complete numerical results in an additional supplementary appendix. This extensive document details the performance metrics per dataset, per protocol, per subject, and per model (comprising over 300 pages). This commitment ensures full transparency of our findings and will significantly facilitate validation and reuse by the research community.
>
> ---
>
> ### **3. Ablation on Token Fusion Strategies (Question 2)**
>
> We agree and have conducted two sets of ablations to clarify these effects.
>
> **1) [CLS] vs. Average Token Fusion (Existing Ablation)**
>
> In the original submission, we compared average pooling over all tokens against using a [CLS] token for our proposed ST-EEGFormer:
>    - In Table E.4 (5 motor imagery datasets), using the [CLS] token leads to a performance decrease of ≈0.001–0.02 compared to average pooling.
>    - In Figure E.5 (P300), the [CLS] token is ≈0.02 lower than the average token.
>    - In Table E.5 (SSVEP), the [CLS] token outperforms average pooling by 0.04.
>    Overall, average pooling performs better on most intermediate tasks, which is why we adopt average token fusion as our default choice.
>
> **2) New Ablation: Fusion Strategy × Head Capacity**
>
>    In the revision, we further study the interaction between fusion and head design, comparing:
>    - A simple head**: average pooling over all tokens followed by a single linear layer (this is the head used in our final models).
>    - A more complex head: flattening all token embeddings followed by a two-layer MLP. For CBraMod and EEGPT, we kept their default complex head design.
>
>    As shown in Figure H.2, the more complex head generally yields higher classification performance, particularly under linear probing. This suggests that:
>    - Naive averaging can discard useful spatiotemporal information.
>    - There is room for further gains by designing more expressive classification heads on top of foundation models.
>
> In summary, our ablations confirm that token fusion and head architecture are non-trivial design choices. While we keep a simple, transparent head in this work for comparability and clarity, our results indicate that dedicated, more powerful heads for ST-EEGFormer are a promising direction for future research.
>
>
> ---
>
>
>
> We again thank Reviewer Jg9y for the thoughtful feedback and high confidence in the review. Your comments have led to measurable improvements in task coverage, experimental clarity, and architectural insight. We hope the revisions meet your expectations and further strengthen the value of our contribution.

---

### Official Review · Reviewer_EwsP · 2025-10-30

**Soundness:** 3
**Presentation:** 2
**Contribution:** 4
**Rating:** 8
**Confidence:** 5

**Summary:**

This paper highlights the importance of comprehensive evaluation—an aspect often overlooked in previous studies—when assessing EEG foundation models. The authors emphasise that fair comparison with classical methods, diverse train–test splitting strategies, and statistical testing of performance differences are essential to truly assess progress in this field. They systematically evaluated five popular EEG foundation models and uncovered meaningful relationships between model performance, downstream tasks, and evaluation protocols, revealing that the performance gains over classical approaches remain limited. Furthermore, they proposed a robust EEG foundation model that serves as a strong baseline and underscored the need for more diverse and large-scale EEG datasets to enable more reliable evaluations.

**Strengths:**

This paper is the first study in EEG Foundation literature with systematic comparisons and standardised the evaluation protocols. It addresses the fundamental question of whether pre-training large-scale foundation models on extensive EEG corpora truly benefits EEG signal decoding. It also serves as an important reminder to researchers in the field not to blindly upscale models—an approach that often leads to wasted resources—and encourages future studies to pursue more reliable and reproducible results without selective reporting.

Quality and Clarity:
- The comparative analysis is comprehensive, with evaluation protocols that cover most considerations relevant to real-world BCI applications. Fair comparisons are conducted using matched metrics, training strategies, and model sizes.
- Rationale of most strategic selections are clearly stated in the body or appendices. (e.g., selection of evaluation protocols and pre-training corpus of ST-EEGFormer)
- The Result section (Section 4) is well-organised, addressing important questions related to foundation models.

**Weaknesses:**

1.	A brief description of the train–validation–test splitting strategy (e.g., whether data were randomly shuffled or split by leaving entire trial sessions out) could be added to Section 3.1, as this choice may influence the performance of models.
2.	The scaling law of EEG foundation models has so far been examined only with respect to model size (i.e., the number of parameters). However, the volume of pre-training data is also a key factor contributing to the computational footprint and could influence the overall performance. Including an investigation of scaling behaviour with respect to dataset size would provide valuable insights and better inform the design of future EEG foundation models.
3.	This paper encourages the community to develop better EEG datasets, but it relies on vague descriptions such as “large-scale,” “diverse,” and “standardized.” These terms are not clearly defined, and the paper lacks elaboration on how greater diversity and standardisation in EEG datasets could specifically address the bottlenecks of EEG foundation models identified in the comparative analysis.
4.	Line 939 of Section C.9 highlights the importance of regression tasks; however, only one regression task is included in the evaluation.
5.	The joint visualisations of all six protocols in Figures 3 and 4 appear somewhat visually cluttered. Since each protocol or pair of protocols evaluates different aspects of the EEG decoding models, selectively presenting the most relevant ones would enhance clarity and allow readers to interpret the results more easily.

**Questions:**

1.	Only 5 representative EEG foundation models are selected. Would the conclusions generalise to the large body of EEG foundation models not tested in this evaluation study?
2.	Line 89 (section 2) mentions that many studies on EEG foundation model lack comparison to “compact neural network decoders.”  However, the well-cited foundation models -- for example, the one listed in Table C.1 --  are benchmarked against classical neural network models, including both general time-series models and EEG-specific models. Could you please clarify which types of baseline NN models are considered under-explored in existing EEG foundation model research?
3.	The difference between variants of STEEGformer -- STEEGformer-s, STEEGformer-b, and STEEGformer-l, is not clearly stated in the paper. Could you please explain the differences?
4.	The use of the phrases “significant” and “statistically significant” may be slightly misleading to readers in Section 4.3. This could be improved by replacing the non-statistical “significant” with other expressions such as “notable,” or “considerable.”
5.	How are the model sizes for EEG Foundation models in Figure 5 (a) calculated? Are the parameters of classification / regression heads included?

---

> ### Author Response · Authors · 2025-11-24
> **Response to Reviewer EwsP-part1**
>
> ## Response to Reviewer EwsP
>
> We sincerely thank the reviewer for the thoughtful comments. We deeply appreciate your recognition of the paper’s contributions, including its comprehensive evaluation, standardized protocol design, and the effort to bring statistical rigor and fairness to the assessment of EEG foundation models. Below, we address each of your points in turn and explain how the revised manuscript responds to your suggestions.
>
> ---
> ### **Weakness**
> #### **1. Train–Validation–Test Split Clarification**
>
> We agree that clarifying our split strategy is important. While Appendix D.1–D.2 provides per-dataset preprocessing and split details, Section 3.2 of the revised manuscript now states the protocol explicitly:
>
> - Classification tasks (general case): We use 5-fold cross-validation per subject across all protocols, except for BCI-IV-2A and ERN, where we retain the original competition train/test split.
> - SSVEP: We use 5-fold cross-validation with a one-session-out design to avoid any potential data leakage.
> - Regression tasks: For each subject/session, we split the time-ordered data into 80% training and 20% testing.
> - FACED and TUEV (added datasets): Given the hundreds of subjects, we follow the conventional benchmark setup and perform cross-subject zero-shot decoding, as in CBraMod.
>
> These choices balance comparability with prior work and rigorous control of leakage while enabling statistically reliable estimates across subjects and tasks.
>
>
> ---
>
> #### **2. Scaling Law vs. Pretraining Dataset Size**
>
> We acknowledge that we did not conduct a dataset-scaling study for pre-training, and we would like to clarify the reasons and implications:
>
>  **1) Why this was not included in the current study**
>   - **Computational constraints** As stated in Appendix E.7, our current pre-training (≈8M EEG segments) already required **>30,000 GPU-hours**. Extending this to multiple additional pre-training runs at different dataset sizes would exceed the available computational budget.
>   - **Prior work has conducted scaling analyses** Previous foundation model studies have already demonstrated that performance improves with more pre-training data for the same model family. For example, CBraMod reports improved performance on FACED dataset with increased training samples (Fig. 7), and LaBraM likewise shows gains with larger token corpora (Fig. 1). Since these trends are already known, we focused our study on a different dimension:  whether foundation models, even when large, actually outperform smaller neural or non-neural decoders across tasks.
>   Our findings show that even the strongest foundation models do not universally dominate—for example, large foundation models do not show statistical significant advantages on tasks such as inner speech and ERN. This highlights that EEG performance is highly task dependent, and not solely determined by model scale.
>   - **Beyond the scope of our research question** Our central goal is to evaluate whether foundation models actually deliver practical superiority in the current state of the field.
>
> **2) Future work**
>
>   We fully agree that scaling remains an important future direction. As more large-scale EEG corpora become available—and with greater computational resources.
>
> ---
>
> #### **3. Clarification of Dataset Terms**
>
> We appreciate the reviewer’s comment and agree that our original use of terms such as “large-scale,” “diverse,” and “standardized” was too vague. We clarify these concepts and their connection to our findings below.
>
> First, our comparative analysis shows substantial performance differences across downstream tasks (Figure G.13). For example, when looking only at average model rank on TUEV, fine-tuned foundation models perform best, whereas on FACED, classical models outperform foundation models. This suggests that conclusions about “state-of-the-art” EEG foundation models can change dramatically depending on which downstream datasets are chosen. Hence, the *composition* and *coverage* of benchmark datasets is a central bottleneck, not just model design.
>
> Concretely, our revised manuscript now emphasizes two community goals in Section 5.4:
>
> - **Develop and share large-scale EEG resources** that can be used *both* for pre-training and evaluation across multiple tasks.
> - **Establish common evaluation protocols and strong, consistent baselines** (including classical and compact neural decoders) so that future EEG foundation models are judged against a shared reference, rather than on isolated, cherry-picked tasks.
> ---
>
> (to be continued...)

---

> > ### Author Response · Authors · 2025-11-24
> > **Response to Reviewer EwsP-part2**
> >
> > ---
> >
> > #### **4. Regression Task Inclusion**
> >
> > We appreciate this observation. In response, we expanded our regression evaluation by adding a second regression benchmark—SEED-VIG (vigilance prediction)—in addition to the original auditory regression task. This new experiment is described in Section 4.4 and Appendix C.6.
> >
> > As mentioned in our overall response, there is also an **independent third-party regression evaluation** running in parallel to this work: our submission to the **NeurIPS 2025 EEG Foundation Challenge**, which uses the same underlying model (ST-EEGFormer) on a regression task. The detailed results of that challenge will become publicly accessible only after the review period, but they provide an additional blind, external check on the model’s regression performance. Together, the two in-paper regression benchmarks and this external evaluation offer a more robust view of how EEG foundation models behave in regression contexts.
> >
> > ---
> >
> > #### **5. Clarity in Visualizations**
> >
> > In the revised version:
> > - We replaced the previous radar plot in Figure 3 with a more straightforward heatmap + bar plot combination.
> > - The bar plot now summarizes the average rank each model achieves across all protocols.
> > - The heatmap provides the corresponding per-protocol rank details in a more readable format.
> > - For per-dataset performance, we refer readers to Appendix Figure G.13, which reports results at the dataset level.
> >
> > We believe these changes make the comparative performance of different models easier to interpret at a glance, while still providing detailed information in the appendix.
> >
> > ---
> >
> > ### **Reviewer Questions**
> >
> > **Q1: Would conclusions generalize beyond the 5 selected foundation models?**
> > Thank you for this insightful comment. To the best of our knowledge, our study currently represents the largest benchmarking effort for EEG foundation models to date, as detailed in our overall response. Our selection of five distinct foundation models is representative of the field's evolution, spanning from the pioneering work of BENDER (2021) to the most recent advancements, such as CBraMod (2025). Crucially, this selection incorporates models with diverse pre-training strategies, ensuring a robust evaluation of varying approaches. We believe this chosen set provides a comprehensive and up-to-date assessment. Furthermore, the evaluation protocols and models we aim to release are designed to serve as a strong, standardized material for future work, enabling robust testing of the generalization capabilities of ongoing EEG foundation model developments.
> >
> > **Q2: What is meant by 'compact neural network decoders'?**
> > Sorry for the misleading statement. What we would like to highlight is that previous foundation models seldom compared against classic non-neural network decoders. Most of the foundation models did a comparison with compact neural networks except for BENDR (one of the earliest
> > Transformer-based foundation models for EEG signals) in the original paper. We provide the following table describing the type of models compared in those foundation model works. We have corrected the main text and updated this information in Table C.1.
> >
> > | Model               | Foundation| NN | none-NN|
> > |-----|----| ---|---|
> > | BENDR (2021)             | ❌ |  ❌ | ❌  |
> > | BIOT (2023)               |  ❌   | ✅ | ❌|
> > | LaBraM (2024)             | ✅ | ✅ | ❌|
> > | EEGPT (2024)             | ✅ | ✅ | ❌ |
> > | CBraMod (2025)           | ✅ | ✅ | ❌ |
> > | **ST-EEGFormer (Ours)** |✅| ✅ | ✅|
> >
> >
> > **Q3: What are the differences between ST-EEGFormer variants?**
> > We explained this in Appendix Table E.1 that the three variants strictly follow the standard ViT-S/B/L configurations:
> >
> > - **ST-EEGFormer-s**: 32.7M parameters (ViT-S–style backbone)
> > - **ST-EEGFormer-b**: 110.9M parameters (ViT-B–style backbone)
> > - **ST-EEGFormer-l**: 328.4M parameters (ViT-L–style backbone)
> >
> > All variants share the same MAE pretraining setup and objectives; the only differences lie in the backbone capacity (depth/width), not in the training procedure or loss.
> >
> > **Q4: Usage of “significant” vs. “statistically significant”**
> > Thank you for this valuable observation. In Section 4.3, we revised all ambiguous uses of “significant” to say “notable” unless explicitly supported by statistical testing.
> >
> > **Q5: Are model sizes inclusive of classifier heads?**
> > Yes, the model sizes reported (e.g., in Figure 5a and Table C.2) include all trainable parameters, including classification or regression heads. This is now stated clearly in the figure caption as: Normalized accuracy versus total trainable parameters
> >
> > ---
> >
> > Thank you again for your thoughtful review and for supporting the acceptance of our work. Your suggestions have helped us significantly improve the clarity and rigor of the paper.

---

### Author Response · Authors · 2025-11-24
**Overall response-part1**

**Dear Area Chair and Reviewers,**

**Re: Response to Reviewers for ICLR 2026 Submission ID 9373, titled *‘Are EEG Foundation Models Worth It? Comparative Evaluation with Traditional Decoders in Diverse BCI Tasks’***

We sincerely thank the reviewers and the area chair for the constructive and thoughtful feedback. We greatly appreciate your time and patience—particularly as we have conducted dense additional experiments and analysis in response to your suggestions, many of which were computationally intensive.

We have provided detailed point-by-point responses to each reviewer, and revised the manuscript accordingly. Here, we would like to highlight and clarify several key contributions and clarifications related to the novelty, scope, and motivation of our work:

---

### **1. Motivation: Why This Study Now?**

We observed a fundamental ambiguity in the field: Nearly all EEG foundation model papers claim state-of-the-art (SOTA) performance across diverse BCI tasks, while many smaller or classical models simultaneously make the same SOTA claims on different downstream tasks. This inconsistency raises a simple yet unaddressed question:
>Do foundation models in EEG truly and consistently outperform classic neural and non-neural models with statistical significance in 2025?

To our knowledge, no prior published work systematically answers this.

---

### **2. Contribution I: A Broad and Statistically Rigorous Benchmarking Study**

We present the first benchmarking study that systematically compares EEG foundation models with both neural and non-neural baselines. We proposed novel **six evaluation protocols** to systematically study model performance, and benchmarked in total of **nine classification and two regression tasks**, with **statistical significance testing** throughout. While we acknowledge that two concurrent works (Wu et al., July 2025; Lee et al., July 2025) were released close to our submission, our study offers broader coverage across models and protocols, and uniquely includes classic non-neural decoders in the analysis. A comparative summary is below:

| Study               | Foundation Models | Neural NN Models | Non-NN Models | Evaluation Protocols | Downstream Tasks               | Statistical Testing |
|---------------------|:-----------------:|:----------------:|:-------------:|:---------------------:|:------------------------------:|:-------------------:|
| Lee et al., 2025    | 2                 | 2                | 0             | 1                     | 5 (classification)             | ❌                  |
| Wu et al., 2025     | 4                 | 4                | 0             | 3                     | 11 (cls), 1 (reg), 1 (retrieval) | ❌                |
| **This Study**      | 6 (incl. ours)    | 4                | 18            | 6                     | 9 (cls), 2 (regression)        | ✅                  |

> *Concurrent submissions (Lee et al., Wu et al.) were first submitted to arXiv within 3 months of our ICLR submission and should be considered parallel efforts. We believe our broader scope and rigorous analysis further distinguish our work.*

* Wu, Jiamin, et al. "Adabrain-bench: Benchmarking brain foundation models for brain-computer interface applications." arXiv preprint arXiv:2507.09882 (2025).
* Lee, Na, et al. "Are Large Brainwave Foundation Models Capable Yet? Insights from Fine-tuning." arXiv preprint arXiv:2507.01196 (2025).

---
(to be continued...)

---

> ### Author Response · Authors · 2025-11-24
> **Overall response-part2**
>
> ---
>
> ### **3. Contribution II: A Transparent, Simple, and Strong Foundation Model**
>
> We introduce ST-EEGFormer, a foundation model built on a standard Vision Transformer (ViT) architecture with plain masked autoencoding (MAE) pretraining. Our goal is not to introduce yet another novel architecture, but rather to establish a transparent, reproducible, and fully open-source baseline that the community can rely on. As the table below illustrates, existing models often involve complex pretraining strategies and are frequently closed-source or only partially released:
>
> | Model               | Pretraining Method                       | Model Availability                        |
> |--------------------|-------------------------------------------|-------------------------------------------|
> | BENDR              | MLM + Contrastive                         | ~3M ✅                                    |
> | BIOT               | Contrastive                               | 3.2M ✅                                   |
> | LaBraM             | MAE in spectral space                     | 5.8M ✅ / 46M ❌ / 369M ❌                 |
> | EEGPT              | Contrastive + MAE                         | 4.7M ❌ / 25M ✅                           |
> | CBraMod            | MAE + CNN + Criss-Cross Transformer       | 4M ✅                                     |
> | EEG2Rep            | MAE in representation space               | Not released ❌                           |
> | **ST-EEGFormer (Ours)** | Plain MAE (raw signals)               | 32.7M ✅ / 110.9M ✅ / 328.4M ✅           |
>
> Our contribution is to provide a **strong and scalable baseline**—simple in architecture, effective in performance, and fully open-source—to promote open science and reproducibility in EEG foundation model research.
>
> ### Transparency & Open-Science Commitment (During Review)
> We are strongly committed to open science and reproducibility. As tangible evidence of this commitment, we have **already provided the full ST-EEGFormer model and training scripts to the NeurIPS 2025 EEG Foundation Challenge organizers** as part of our official submission to this competition. After the double-blind review concludes, we will publicly release the complete repository—including code, pretrained weights, configurations to support transparent verification and community reuse.
>
> In parallel, we participated in the NeurIPS 2025 EEG Foundation Challenge on a regression task and observed strong performance on a large-scale leaderboard (>1,000 teams; >8,000 submissions). While orthogonal to this paper, the challenge offers an independent, blind validation of our model’s generalization. To fully comply with the double-blind policy, we used an **alternate public-facing alias** during the competition and are **withholding team identifiers and URLs** until the review period ends, after which we will link all related materials to this work.
>
>
> ---
>
> ### **4. Additional Experiments and Extensions**
>
> As part of our revision, we conducted substantial new experiments to expand coverage and address reviewer feedback:
>
> - **New Datasets (3 Added):**
>   - *SEED-VIG* – Vigilance prediction (regression)
>   - *FACED* – Emotion classification
>   - *TUEV* – EEG event classification
>
> - **Two Additional Experiments:**
>   - **Token Fusion & Head Ablation:** We varied fusion strategies and classifier heads across models to analyze performance sensitivity.
>   - **Downstream Data Scaling:** We studied how varying percentages of training data affect foundation vs. classic model performance.
>
> ---
>
> We hope these clarifications and additions address the concerns raised and clearly articulate the value and positioning of our work. Thank you again for your feedback and for helping us strengthen this study.
>
> Warm regards,
> *Authors of Submission 9373*

---

### Author Response · Authors · 2025-12-03
**Summary of Key Points for AC & PC Decision**

To the Area Chairs and the Program Chairs,

We sincerely appreciate your time and effort in overseeing this submission, especially under the current circumstances. We would like to briefly summarize two key points for clarification and final consideration:

1. **Lack of reviewer discussion**
   We regret that there was relatively little interaction during the discussion phase, despite the substantial additional experiments and analyses we conducted in response to the reviews. We remain confident that more open dialogue could have helped resolve several misunderstandings and better contextualize our contributions.

2. **Clarification regarding novelty and contribution**
   a) **On novelty concerns raised by two reviewers (ABR4 & gwxQ)**
   We believe it is inappropriate to judge our work’s novelty against *concurrent* papers (Wu et al., July 2025; Lee et al., July 2025) — as defined by ICLR’s concurrent-work policy. Even comparing to those works, our study distinguishes itself through:
   - broader model coverage (including the first systematic comparison with classical non-neural decoders),
   - expanded dataset and protocol scope,
   - rigorous statistical testing across all evaluations, and
   - the introduction of a fully open-source, scalable baseline (ST-EEGFormer) filling a current reproducibility gap,  as well as demonstrating critical implementation factors (e.g., token fusion, classifier design) that meaningfully impact foundation-model performance.

   b) **Our novelty is not in proposing another "novel" EEG foundation model architecture, but in establishing a fair, trustworthy, and reusable baseline for the community**
   Many prior EEG foundation models either release only partial resources (typically small variants), or not release at all, including those (EEG2Rep & Brant-2) suggested by the reviewer ABR4 — making direct comparison impossible.

   Our contribution is to provide a transparent and community-reusable baseline. As noted in our Overall Response (Part 2: Transparency & Open-Science Commitment (During Review)), we have already shared our model and training code with the NeurIPS 2025 EEG Foundation Challenge organizers as concrete evidence of our commitment to openness. This is complemented by a comprehensive benchmark that supports fair evaluation of future models, rather than continuing a cycle of non-reproducible SOTA claims.

In addition, we have responded point by point to each reviewer’s comments, with new experiments and analyses where appropriate; we kindly refer you to the discussion thread for detailed responses.

We hope these clarifications assist in your final assessment, and we are very grateful for your consideration.

---

### Meta-Review · Area_Chair_gYpy · 2026-01-05

**Summary:**

This paper asks whether EEG “foundation models” actually deliver practical gains over established EEG decoders, and answers it via a large, statistically rigorous benchmark spanning multiple paradigms, datasets, and *six* evaluation protocols (population, per-subject self/transfer, leave-one-subject-out zero-shot, fine-tune, and generalization drop). The authors compare several prominent EEG foundation models and their *linear-probe vs fine-tune* usage, against strong classic neural baselines and classic non-neural pipelines (e.g., CSP/FBCSP and Riemannian methods), and introduce a transparent ViT-style baseline, ST-EEGFormer, pre-trained with simple masked autoencoding on 8M+ raw EEG segments. Across various classification and regression tasks, they find that fine-tuned foundation models can be strong in data-rich, population settings, but in data-scarce/subject-specific regimes they often *do not significantly beat* compact neural networks and can be comparable to (or even lag) classic non-neural decoders; linear probing is consistently weak and task-dependent, and bigger models don’t yield a reliable “scaling law” payoff relative to their rising compute cost.

**Reviewer Concerns:**

The majority of reviewers find the paper comprehensive, highlighting its careful benchmarking and statistical rigor.

The main concern, raised by two reviewers (and echoed in part by a third), is novelty. One reviewer argues that the submission reads more like a benchmarking/engineering study than an ICLR-level research contribution. Reviewers also point to two recent papers, including an ICML 2025 paper [1], that investigate a similar question and reach broadly similar conclusions. Having read both, I agree with the shared takeaway that current EEG/LBM “foundation” approaches do not consistently and meaningfully outperform strong compact baselines. That said, the present submission is substantially broader in scope than [1], offering a more extensive benchmark across tasks and evaluation settings. In that sense, the results here can be viewed as complementary and as strengthening the evidence base behind the conclusions in [1].

Reviewers also raised several secondary issues, including expanding regression coverage, clarifying vague terminology, improving visualizations, clarifying fusion strategies, adding missing citations, and noting that the “scaling law” discussion largely considers model size rather than pretraining data. Many of these points were addressed by the authors during the discussion period.

[1] Lee, Na, et al. “Are Large Brainwave Foundation Models Capable Yet? Insights from Fine-tuning.” ICML 2025.

**Reviewer Scores:**

The reviews are mixed. Two reviewers recommend acceptance, one is slightly below the threshold, and one recommends rejection. The reviewer recommending rejection is primarily concerned about overlap with an ICML 2025 paper. However, I believe the current submission still adds value, both by expanding the evaluation to a broader set of tasks and protocols and by further reinforcing the central message of the ICML 2025 work in a more comprehensive setting.

I think the authors did a solid job addressing the remaining reviewer concerns. The paper is also well written and overall high quality. For these reasons, I recommend the paper to be accepted as a poster presentation.

---

### Decision · Program_Chairs · 2026-01-26

Accept (Poster)